# MoSSP: A Momentum-Based Single-Loop Stochastic Penalty Method for Nonconvex Constrained DC-Regularized Optimization

**Luxuan Li**[1] **Chunfeng Cui**[1] **Xiao Wang**[2]

## Abstract

In this paper, we study a structured class of nonconvex constrained stochastic problems with difference-of-convex (DC) regularization, where the feasible set is possibly nonconvex and the concave part of the DC regularizer is allowed to be nonsmooth. The fundamental challenge lies in maintaining feasibility for nonconvex constraints while achieving favorable oracle complexity. Although single-loop algorithms efficiently solve unconstrained DC optimization problems, their potential for constrained optimization with DC structure remains largely unexplored. To address this gap, we develop **MoSSP**, a **Mo**mentum-based **S**ingle-loop **S**tochastic **P**enalty method for such problems with provable complexity guarantees. The key idea is to apply a single stochastic proximal-gradient step to the Moreau envelope of the penalty plus the convex DC part, with the concave part's proximal mapping computed in parallel. We derive two algorithm variants: a Polyak-momentum version with $\mathcal{O}(\varepsilon^{-4})$ oracle complexity for finding stochastic $\varepsilon$-KKT points, and an improved $\mathcal{O}(\varepsilon^{-3})$ version incorporating recursive momentum. Experimental results demonstrate the effectiveness of the proposed algorithms.

## 1. Introduction

In this paper, we consider a class of nonconvex constrained stochastic difference-of-convex (DC)-regularized optimization problems

$$\min_{\boldsymbol{x} \in \mathbb{R}^n} \quad F(\boldsymbol{x}) = \{f(\boldsymbol{x}) = \mathbb{E}_{\xi}[\mathbf{f}(\boldsymbol{x}, \xi)]\} + h(\boldsymbol{x}) - g(\boldsymbol{x}),$$
$$\text{s.t.} \quad \boldsymbol{c}(\boldsymbol{x}) = \boldsymbol{0}, \tag{P}$$

[1]School of Mathematical Sciences, Beihang University, Beijing 100191, China [2]School of Computer Science and Engineering, Sun Yat-sen University, Guangzhou 510006, China. Correspondence to: Xiao Wang <wangx936@mail.sysu.edu.cn>.

*Proceedings of the 43rd International Conference on Machine Learning*, Seoul, South Korea. PMLR 306, 2026. Copyright 2026 by the author(s).

where $\xi$ is a random variable on a probability space $\Xi$, independent of $\boldsymbol{x}$. The function $f : \mathbb{R}^n \to \mathbb{R}$ and mapping $\boldsymbol{c} : \mathbb{R}^n \to \mathbb{R}^m$ are continuously differentiable, while $h : \mathbb{R}^n \to \mathbb{R} \cup \{+\infty\}$ and $g : \mathbb{R}^n \to \mathbb{R}$ are proper, closed, convex, and possibly nonsmooth functions. The feasible set is assumed to be nonempty. The proximal mappings of $h$ and $g$ are assumed to be available individually. Computing exact gradients of $f$ is often prohibitive because evaluating the expectation can be costly or the distribution of $\xi$ is not explicitly known. Therefore, we assume only access to stochastic gradients of $f$ at queried points.

Problem (P) captures a wide range of applications in machine learning and statistical learning, where $f$ is a data-fidelity loss and the DC structure appears in the regularizer $h - g$, which promotes desirable structures such as sparsity; see (Gong et al., 2013, Table 1) and (Xu et al., 2019; Wen et al., 2018). Nonconvex constraints arise naturally from structure, resource, or safety requirements; examples include energy budgets in DNN compression (Yang et al., 2019; Chen et al., 2018), safety constraints in reinforcement learning (Paternain et al., 2019; Zhang et al., 2025; 2026), and low-rank or sphere constraints (Roy et al., 2018; Witten et al., 2009). Crucially, we do not assume access to the proximal operator of the whole term $h - g$; for instance, neither the truncated $\ell_{1-2}$ regularizer (Ma et al., 2017) nor the truncated $\ell_1$ regularizer (Luo et al., 2015) admits a closed-form proximal mapping. Moreover, $g$ in Problem (P) is not necessarily differentiable. Prominent examples include the capped $\ell_1$ regularization model (Gong et al., 2013) and the $\ell_{1-2}$ regularization model (Yin et al., 2015) commonly used in compressed sensing.

### 1.1. Motivating Examples

We provide two motivating examples that can be formulated as instances of Problem (P).

**DNN training under energy budgets** (Yang et al., 2019). In energy-aware structured pruning, let $\boldsymbol{W} = \{\boldsymbol{w}_u\}_{u=1}^L$ and $\boldsymbol{S} = \{s_u\}_{u=1}^L$ denote the layer-wise weight tensors and sparsity-level variables. A typical formulation is

$$\min_{\boldsymbol{W}, \boldsymbol{S}} \quad f(\boldsymbol{W}) + \lambda R_{\mathrm{DC}}(\boldsymbol{W}),$$
$$\text{s.t.} \quad \phi(\boldsymbol{w}_u) \le s_u, \quad \psi(\boldsymbol{S}) \le E_{\mathrm{budget}}, \quad \forall u,$$

where $f(\boldsymbol{W}) := \mathbb{E}_\xi[\mathbf{f}(\boldsymbol{W}; \xi)]$ is the expected training loss, $\phi(\boldsymbol{w}_u)$ measures layer-wise sparsity, $\psi(\boldsymbol{S})$ models total energy consumption, and $R_{\mathrm{DC}}(\boldsymbol{W})$ is a DC-structured sparsity regularizer (e.g., capped-$\ell_1$ penalty). Both $\phi$ and $\psi$ can be nonlinear and nonconvex.

**Nonnegative sparse CCA** (Witten et al., 2009). For paired data $\boldsymbol{X} \in \mathbb{R}^{N \times p}$ and $\boldsymbol{Y} \in \mathbb{R}^{N \times q}$, sparse nonnegative canonical loading vectors can be obtained via

$$\min_{\boldsymbol{u}, \boldsymbol{v}} \quad -\boldsymbol{u}^\top \boldsymbol{X}^\top \boldsymbol{Y} \boldsymbol{v} + \lambda_1 \rho(\boldsymbol{u}) + \lambda_2 \rho(\boldsymbol{v}),$$
$$\text{s.t.} \quad \|\boldsymbol{u}\|_2^2 = 1, \quad \|\boldsymbol{v}\|_2^2 = 1, \quad \boldsymbol{u}, \boldsymbol{v} \geq \boldsymbol{0},$$

where $\rho(\cdot)$ is a DC-structured sparsity penalty (e.g., capped $\ell_1$). The problem is nonconvex due to the bilinear objective, unit-sphere constraints, and the nonconvex regularizer.

### 1.2. Related Work

For DC optimization problems, prior work has largely focused on unconstrained or convex-set constrained settings, where feasibility can be maintained by projection. The classical DC algorithm (DCA) (Tao & Souad, 1986) linearizes the concave part and solves a sequence of convex subproblems. Following this idea of constructing tractable convex majorants, proximal, Bregman, and stochastic variants have been developed with global convergence or complexity guarantees (Wen et al., 2018; Liu et al., 2019; Yang et al., 2025a; Liu & Takeda, 2022; Yang et al., 2025b; Nitanda & Suzuki, 2017; Xu et al., 2019). Handling general functional constraints, however, requires additional care. Recent efforts address convex inequality constraints in Problem (P) via convex approximations of the feasible region, obtaining favorable convergence guarantees under suitable constraint qualifications in the deterministic setting; see (Kanzow & Neder, 2026; Liu et al., 2025; Yu et al., 2021). The difficulty becomes more pronounced when nonconvex constraints are present, since feasibility is generally difficult to maintain. While there are algorithms tailored to specific nonconvex constraints, such as manifolds (Bergmann et al., 2024; Jiang et al., 2025) or conic sets (Xu et al., 2025), such analyses are inherently geometry-dependent and do not extend easily to general nonconvex constraints. More recently, Le Thi et al. (2024) studied DC composite optimization under nonconvex constraints via an exact penalty method, but non-asymptotic analysis in the stochastic setting remains unaddressed.

Note that most of the aforementioned methods are double-loop algorithms, in which a subproblem must be (approximately) solved in inner iterations. As a result, they can be relatively complex due to extensive hyperparameter tuning (e.g., the penalty parameter in (Algorithm 3, Le Thi et al., 2024)) and the need for precise termination criteria for subproblem solvers (e.g., Yu et al., 2021; Liu & Takeda, 2022). The growing scale of data and models motivates the need

for more efficient optimization methods. In this context, single-loop designs become attractive, especially in stochastic settings. For Problem (P) with $g \equiv 0$ or $h - g \equiv 0$, single-loop stochastic penalty algorithms, including augmented Lagrangian methods, already provide compelling oracle complexity guarantees for finding approximate KKT points (Alacaoglu & Wright, 2024; Shi et al., 2026; Liu & Xu, 2025; Lu et al., 2026). Nevertheless, the effectiveness of such single-loop approaches normally depends on tractable subproblems, a condition typically met when $h$ admits an efficiently computable proximal mapping. In Problem (P), however, the term $h - g$ is generally non-prox-friendly. Consequently, single-loop algorithms capable of handling such nonsmooth DC regularizations under general nonconvex constraints remain scarce.

To address this inherent nonsmoothness in the DC structure, some recent work has explored alternative smoothing techniques. Specifically, several studies (Moudafi, 2023; Sun & Sun, 2023; Hu et al., 2024; Chayti & Jaggi, 2025) apply the Moreau envelope to each convex component in the DC structure and then take their difference, yielding the Difference-of-Moreau-Envelopes (DME) for non-asymptotic convergence analysis. The DME approach preserves the global DC structure, and stationary points are recoverable via the inexpensive proximal operator of each original component. Among these works, the method of Hu et al. (2024) is, to the best of our knowledge, the first single-loop method for stochastic difference-of-weakly-convex (DWC) optimization based on the DME idea, and achieves $\mathcal{O}(\varepsilon^{-4})$ oracle complexity. More recently, Chayti & Jaggi (2025) introduced momentum into stochastic DC optimization by applying it to the concave component and obtained $\mathcal{O}(\varepsilon^{-4})$ oracle complexity. Despite these advances, most existing methods still rely on double-loop schemes to achieve state-of-the-art non-asymptotic rates (Nitanda & Suzuki, 2017; Xu et al., 2019), or are confined to unconstrained problems; see (Nitanda & Suzuki, 2017; Xu et al., 2019; Hu et al., 2024; Chayti & Jaggi, 2025). While Sun & Sun (2023) developed an augmented Lagrangian algorithm for linearly constrained DC problems in a deterministic setting, such constructions can become unstable under stochastic noise due to sensitive multiplier updates, thus hindering their practical adoption.[1]

In light of these limitations and to facilitate practical implementation, we develop an efficient single-loop, penalty-based method for solving Problem (P). Inspired by the DME technique, the proposed method smooths the penalized formulation and solves the resulting subproblems using stochastic proximal gradient updates. A refined analysis is then developed to establish favorable complexity guarantees for the proposed method. The key contributions of this paper are summarized as follows.

---

[1] A broader review is deferred to Appendix A.

▶ **Simple single-loop framework.** We propose **MoSSP**, a **mo**mentum-based **s**ingle-loop **s**tochastic **p**enalty framework for solving Problem (**P**). By fully leveraging the proximal operators of the components in the DC regularizer, MoSSP avoids solving inner subproblems and extensive hyperparameter tuning while flexibly integrating advanced variance reduction techniques.

▶ **Comparable complexity guarantee.** To the best of our knowledge, we establish the first complexity results for nonconvex DC-regularized optimization with nonlinear constraints. Under mild conditions, **MoSSP-P** (with Polyak momentum) achieves $\mathcal{O}(\varepsilon^{-4})$ oracle complexity for finding a stochastic $\varepsilon$-KKT point, while the recursive momentum variant, **MoSSP-R**, attains an improved $\mathcal{O}(\varepsilon^{-3})$ rate under the mean-squared smoothness assumption, matching the lower bound of unconstrained stochastic optimization (Arjevani et al., 2023) under the same assumption. A comparison with existing algorithms for solving (un)constrained DC(-regularized) problems is shown in Table 1.

▶ **Novel theoretical analysis.** We develop a DC-aware complexity analysis that explicitly characterizes stochastic errors within the criticality measure. By constructing a DC-specific potential function, our analysis achieves coordinated control of smoothing, momentum, and penalty parameters, thereby maintaining near-optimal complexity compared to the unconstrained setting.

## 2. Preliminaries

**Notations.** We use $\|\cdot\|$ for the Euclidean norm and $\langle\cdot,\cdot\rangle$ for the Euclidean inner product. The gradient of a differentiable function $f$ at $\boldsymbol{x}$ is denoted by $\nabla f(\boldsymbol{x})$. For the constraint mapping $\boldsymbol{c}:\mathbb{R}^n\to\mathbb{R}^m$, we use $\nabla\boldsymbol{c}(\boldsymbol{x})$ to denote the transpose of its Jacobian, i.e., $\nabla\boldsymbol{c}(\boldsymbol{x}):=J_{\boldsymbol{c}}(\boldsymbol{x})^\top\in\mathbb{R}^{n\times m}$. For any point $\boldsymbol{x}\in\mathbb{R}^n$ and any set $\mathcal{S}\subseteq\mathbb{R}^n$, $\mathrm{dist}(\boldsymbol{x},\mathcal{S}):=\inf_{\boldsymbol{y}\in\mathcal{S}}\|\boldsymbol{x}-\boldsymbol{y}\|$ denotes the point-to-set distance; for $\mathcal{X},\mathcal{Y}\subseteq\mathbb{R}^n$, $\mathrm{dist}(\mathcal{X},\mathcal{Y}):=\inf_{\boldsymbol{x}\in\mathcal{X},\boldsymbol{y}\in\mathcal{Y}}\|\boldsymbol{x}-\boldsymbol{y}\|$. For an extended-real-valued function $\varphi$, $\partial\varphi(\boldsymbol{x})$ denotes the general (limiting) subdifferential, which reduces to the convex subdifferential when $\varphi$ is convex. Let $\xi^{[k]}=\{\xi^0,\ldots,\xi^k\}$ be the collection of i.i.d. samples drawn up to iteration $k$. We use $\mathbb{E}[\cdot]$ for expectation and $\mathbb{E}[\cdot\mid\xi^{[k]}]$ for conditional expectation given the sample history.

### 2.1. Criticality in DC Optimization

Consider the unconstrained DC optimization problem:

$$\min_{\boldsymbol{x}\in\mathbb{R}^n}\Psi(\boldsymbol{x}):=\phi(\boldsymbol{x})-g(\boldsymbol{x}), \qquad (2.1)$$

where $\phi:\mathbb{R}^n\to\mathbb{R}\cup\{+\infty\}$ is $m_\phi$-weakly convex (possibly nonsmooth), and $g:\mathbb{R}^n\to\mathbb{R}\cup\{+\infty\}$ is a proper, closed, and convex function.

As is well known, the *Moreau envelope* provides favorable smoothing properties for weakly convex functions. For the component function $\phi$ and any $\mu\in(0,1/m_\phi)$, its Moreau envelope $\mathcal{M}_{\mu\phi}$ and the associated proximal mapping are well-defined as:

$$\mathcal{M}_{\mu\phi}(\boldsymbol{z}):=\min_{\boldsymbol{x}\in\mathbb{R}^n}\left\{\phi(\boldsymbol{x})+\frac{1}{2\mu}\|\boldsymbol{x}-\boldsymbol{z}\|^2\right\},$$

$$\mathrm{prox}_{\mu\phi}(\boldsymbol{z}):=\arg\min_{\boldsymbol{x}\in\mathbb{R}^n}\left\{\phi(\boldsymbol{x})+\frac{1}{2\mu}\|\boldsymbol{x}-\boldsymbol{z}\|^2\right\}, \quad (2.2)$$

respectively. Note that $\mathcal{M}_{\mu\phi}$ serves as a smooth surrogate of $\phi$ with gradient (Moreau, 1965)

$$\nabla\mathcal{M}_{\mu\phi}(\boldsymbol{z})$$
$$=\mu^{-1}(\boldsymbol{z}-\mathrm{prox}_{\mu\phi}(\boldsymbol{z}))\in\partial\phi(\mathrm{prox}_{\mu\phi}(\boldsymbol{z})). \qquad (2.3)$$

For the weakly convex optimization problem $\min\phi(\boldsymbol{x})$, given $\varepsilon>0$, one typically seeks an $\varepsilon$-stationary point of its Moreau envelope $\mathcal{M}_{\mu\phi}$, i.e., $\bar{\boldsymbol{x}}$ with $\|\nabla\mathcal{M}_{\mu\phi}(\bar{\boldsymbol{x}})\|\le\varepsilon$, as a relaxed convergence criterion (Davis & Drusvyatskiy, 2019). However, this approach cannot be directly extended to Problem (2.1), as the objective $\Psi$ may lack weak convexity. We thus introduce the notion of an $\varepsilon$-*critical point* for DC optimization (Sun & Sun, 2023), i.e., a point $\bar{\boldsymbol{x}}\in\mathbb{R}^n$ is an $\varepsilon$-*critical point* of $\Psi$ if there exist $\bar{\boldsymbol{y}}\in\mathbb{R}^n$ and $\bar{\boldsymbol{u}}\in\partial\phi(\bar{\boldsymbol{x}})-\partial g(\bar{\boldsymbol{y}})$ such that $\max\{\|\bar{\boldsymbol{u}}\|,\|\bar{\boldsymbol{x}}-\bar{\boldsymbol{y}}\|\}\le\varepsilon$. This two-point formulation serves as a natural stopping criterion for DCA-type methods in practice, since $\partial\phi$ and $\partial g$ are usually evaluated at different points at each iteration in standard DCA; for example, a subgradient $\xi^k\in\partial g(x^k)$ is used to compute $x^{k+1}$. Note that when $\varepsilon=0$, this definition recovers the classical DC criticality condition (Pang et al., 2017).

Moreover, the overall Moreau envelope of $\Psi$ in Problem (2.1) is computationally intractable. To this end, we apply the Moreau envelope to $\phi$ and $g$ individually and take their difference (DME) to define a smooth approximation of Problem (2.1) for any $\mu\in(0,1/m_\phi)$:

$$\min_{\boldsymbol{z}\in\mathbb{R}^n}\Psi_\mu(\boldsymbol{z}):=\mathcal{M}_{\mu\phi}(\boldsymbol{z})-\mathcal{M}_{\mu g}(\boldsymbol{z}). \qquad (2.4)$$

The smoothness of $\Psi_\mu(\boldsymbol{z})$ was established in Hiriart-Urruty (1991).

However, it is not immediately clear how the approximate solutions of Problems (2.4) and (2.1) relate to each other. We establish that any $\varepsilon$-stationary point of $\Psi_\mu$ can be converted into an $\varepsilon$-critical point of $\Psi$, with the detailed proof deferred to Appendix B.5. This correspondence plays a key role in our analysis. Complete properties of DME and the equivalence between solutions of these two problems are deferred to Appendices B.3 and B.4, respectively.

*Table 1.* Comparison of algorithms for (un)constrained DC(-regularized) optimization in stochastic and deterministic settings.

| Algorithm | Cons. Type | DC Struct. | Stoch. Assump. | Single-Loop | Iter. Comp. | Oracle Comp. |
|---|---|---|---|---|---|---|
| SPD (Nitanda & Suzuki, 2017) | – | ✓* | $L$-sm | – | $\mathcal{O}(\varepsilon^{-4})$ | – |
| SSDC-SPG (Xu et al., 2019) | – | ✓ | $\nu$-Hölder | – | $\mathcal{O}(\varepsilon^{-4/\nu})$ | – |
| SMAG (Hu et al., 2024) | – | ✓* | – | ✓ | $\mathcal{O}(\varepsilon^{-4})$ | $\mathcal{O}(\varepsilon^{-4})$ |
| Algo. 2-Polyak (Chayti & Jaggi, 2025) | – | ✓ | $L$-sm | ✓ | $\mathcal{O}(\varepsilon^{-4})$ | $\mathcal{O}(\varepsilon^{-4})$ |
| Algo. 2-Recursive (Chayti & Jaggi, 2025) | – | ✓ | Assump. 3.1 | ✓ | $\mathcal{O}(\varepsilon^{-4})$ | $\mathcal{O}(\varepsilon^{-4})$ |
| CLCDC-ALM (Sun & Sun, 2023) | linear | ✓ | – | – | $\tilde{\mathcal{O}}(\varepsilon^{-3})$ | – |
| iMBAdc (Liu et al., 2025) | cvx | ✓ | – | – | $\mathcal{O}(\varepsilon^{-2})$ | – |
| MLALM (Shi et al., 2026) | ncvx | – | Assump. 3.1 | ✓ | $\mathcal{O}(\varepsilon^{-3})$ | $\mathcal{O}(\varepsilon^{-3})$ |
| **MoSSP-P (Algo. 1)** | **ncvx** | ✓ | $L$-sm | ✓ | $\mathcal{O}(\varepsilon^{-4})$ | $\mathcal{O}(\varepsilon^{-4})$ |
| **MoSSP-R (Algo. 2)** | **ncvx** | ✓ | Assump. 3.1 | ✓ | $\mathcal{O}(\varepsilon^{-3})$ | $\mathcal{O}(\varepsilon^{-3})$ |

**Notes.** Abbreviations: Cons. Type = constraint type; DC Struct. = DC structure; cvx = convex; ncvx = nonconvex; $\tilde{\mathcal{O}}(\cdot)$ hides polylogarithmic factors. ✓* marks related but different unconstrained DC-type models: SPD (Nitanda & Suzuki, 2017) studies stochastic DC programs $\min_x g(x) - h(x)$ with differentiable convex components, whereas SMAG (Hu et al., 2024) studies stochastic DWC optimization under weak-convexity assumptions; neither covers the nonsmooth DC-regularized setting in Problem (P). Iter. Comp. counts the number of (outer) iterations; Oracle Comp. counts the total number of stochastic first-order oracle calls; Stoch. Assump. specifies the stochastic assumption, with deterministic methods (CLCDC-ALM (Sun & Sun, 2023), iMBAdc (Liu et al., 2025)) requiring none; $L$-sm = expected Lipschitz smoothness; Assump. 3.1 = mean-squared smoothness assumption; $\nu$-Hölder = gradient is $\nu$-Hölder continuous. For iMBAdc, the $\mathcal{O}(\varepsilon^{-2})$ bound counts only outer iterations; the total number of inner iterations is not quantified. For Algo. 2-Polyak/Recursive (Chayti & Jaggi, 2025), the concave component is assumed smooth, while the convex component is only required to have bounded subgradients; applying recursive momentum to the smooth concave part improves its per-sample oracle complexity to $\mathcal{O}(\varepsilon^{-3})$, but the convex-component queries remain $\mathcal{O}(\varepsilon^{-4})$ and therefore dominate the total complexity.

**Proposition 2.1.** *Given $\varepsilon > 0$, for any $0 < \mu < 1/m_\phi$, if $\bar{z}$ satisfies $\|\nabla \Psi_\mu(\bar{z})\| \leq \min\{1, \mu^{-1}\}\varepsilon$, then the point $\bar{x} := \text{prox}_{\mu\phi}(\bar{z})$ is an $\varepsilon$-critical point of $\Psi$ with the auxiliary point $\bar{y} := \text{prox}_{\mu g}(\bar{z})$.*

## 2.2. Approximate Solutions for Constrained DC-Regularized Optimization

As is standard in constrained nonconvex optimization, we seek points satisfying the KKT conditions of Problem (P). Due to the stochasticity in the objective, it is natural to evaluate the optimality residuals in expectation. We next define two types of $\varepsilon$-approximate stochastic KKT solutions for Problem (P), together with the associated criticality measures used in our analysis. A discussion of Definition 2.1 is provided in Appendix B.6.

**Definition 2.1.** *Given $\varepsilon > 0$, a point $\bar{x}$ is a stochastic $\varepsilon$-KKT point of Problem (P) if there exist $(\bar{u}, \bar{y}, \bar{\lambda}) \in \mathbb{R}^n \times \mathbb{R}^n \times \mathbb{R}^m$ such that almost surely,*

$$\bar{u} \in \nabla f(\bar{x}) + \nabla c(\bar{x})\bar{\lambda} + \partial h(\bar{x}) - \partial g(\bar{y}), \quad (2.5)$$

*and*

$$\begin{cases} \max\{\mathbb{E}[\|\bar{u}\|^2], \mathbb{E}[\|\bar{x} - \bar{y}\|^2]\} \leq \varepsilon^2 & \textbf{(criticality)}, \\ \mathbb{E}[\|c(\bar{x})\|^2] \leq \varepsilon^2 & \textbf{(feasibility)}. \end{cases} \quad (2.6)$$

*A point $\bar{x}$ is a stochastic $\varepsilon$-stationary point of Problem (P)*

*if there exist $(\bar{u}, \bar{y}, \bar{\lambda})$ satisfying (2.5) almost surely and*

$$\begin{cases} \max\{\mathbb{E}[\|\bar{u}\|^2], \mathbb{E}[\|\bar{x} - \bar{y}\|^2]\} \leq \varepsilon^2 & \textbf{(criticality)}, \\ \mathbb{E}[\|\nabla c(\bar{x})c(\bar{x})\|^2] \leq \varepsilon^2 & \textbf{(infeasible stationarity)}. \end{cases} \quad (2.7)$$

## 2.3. Main Assumptions

Throughout the paper, we make the following assumptions for Problem (P). We assume $F^* := \inf_{x \in \mathbb{R}^n} F(x) > -\infty$.

**Assumption 2.1.** *Function $f(\cdot)$ is $L_f$-smooth. Function $c$ is $L_c$-smooth. There exist $C, G > 0$ such that*

$$\|\nabla f(x)\| \leq G, \quad \sup_{v_h \in \partial h(x)} \|v_h\| \leq G, \quad \sup_{v_g \in \partial g(x)} \|v_g\| \leq G, \quad (2.8)$$

$$\|\nabla c(x)\| \leq G, \quad \|c(x)\| \leq C, \quad \forall x \in \mathbb{R}^n.$$

The boundedness condition is crucial for ensuring reliable convergence of the iterative sequence in stochastic constrained optimization. The inherent randomness of the process makes it difficult to guarantee that all iterates remain within a specific level set. The necessity of Assumption 2.1 is well established in (Berahas et al., 2021; Na et al., 2023b;a; Wang, 2025; Sun & Sun, 2023).

**Assumption 2.2.** *There exists a constant $\sigma > 0$ such that*

$$\mathbb{E}_\xi[\nabla \mathbf{f}(x, \xi)] = \nabla f(x),$$
$$\mathbb{E}_\xi[\|\nabla \mathbf{f}(x, \xi) - \nabla f(x)\|^2] \leq \sigma^2, \quad \forall x \in \mathbb{R}^n.$$

For nonconvex constraints, we impose the following non-singularity condition on the constraints to control feasibility of the generated solutions. Constraint qualifications are often required in solving constrained optimization problems; see (Shi et al., 2026; Lu et al., 2026; Liu & Xu, 2025; Curtis et al., 2024).

**Assumption 2.3.** For the iterate sequence $\{x^k\}_{k \in \mathbb{N}}$ generated by the algorithms, there exists a constant $\delta > 0$ such that

$$\|\nabla c(x^k) c(x^k)\| \geq \delta \|c(x^k)\|, \quad \forall k \geq 0.$$

## 3. Momentum-Based Single-Loop Stochastic Penalty Algorithms

### 3.1. Algorithmic Framework

We incorporate the nonconvex constraints in Problem (P) through a sequence of quadratically penalized DC objectives with nondecreasing penalty parameters $\{\rho_k\} \subset (0, \infty)$:

$$\min_{x \in \mathbb{R}^n} \left\{ F_{\rho_k}(x) := \underbrace{Q_{\rho_k}(x) + h(x)}_{\psi_{\rho_k}(x)} - g(x) \right\}, \quad (3.1)$$

where $Q_\rho(x) = f(x) + \frac{\rho}{2} \|c(x)\|^2$. To address the nonsmoothness of the penalty problems, we take the difference of the Moreau envelopes of each component to construct a smoothed surrogate function $F_{\rho,\mu}(z)$:

$$F_{\rho,\mu}(z) = \mathcal{M}_{\mu\psi_\rho}(z) - \mathcal{M}_{\mu g}(z), \quad \text{with } \rho = \rho_k. \quad (3.2)$$

Crucially, by Proposition 2.1, if $\bar{z}$ satisfies $\|\nabla F_{\rho,\mu}(\bar{z})\| \leq \min\{1, \mu^{-1}\} \varepsilon$, then any $\bar{x}$ with $\|\bar{x} - \text{prox}_{\mu g}(\bar{z})\| \leq \varepsilon$ is an $\varepsilon$-critical point of $F_\rho$. This gives the approximate criticality criterion targeted in (2.6) or (2.7). Therefore, we aim to efficiently find a point $\bar{z}$ by iteratively minimizing $F_{\rho,\mu}(z)$.

A standard strategy is to apply gradient descent to $F_{\rho,\mu}(z)$, which inspires our framework. Following the properties of the Moreau envelope, the gradient of $F_{\rho,\mu}$ at the point $z^k$ is given by

$$\begin{aligned} \nabla F_{\rho,\mu}(z^k) &= \mu^{-1}(z^k - \text{prox}_{\mu\psi_\rho}(z^k)) \\ &\quad - \mu^{-1}(z^k - \text{prox}_{\mu g}(z^k)) \quad (3.3) \\ &= \mu^{-1}\left(\text{prox}_{\mu g}(z^k) - \text{prox}_{\mu\psi_\rho}(z^k)\right). \end{aligned}$$

This involves the proximal operators of $g$ and $\psi_\rho$. However, computing the exact proximal operator of the composite function $\psi_\rho$ is typically intractable. To this end, we maintain a variable $x$ as an estimator of $\text{prox}_{\mu\psi_\rho}(z)$ and compute $\text{prox}_{\mu g}(z)$ in parallel. Specifically, at each iteration, we update $x^k$ via a stochastic proximal gradient descent step

$$\begin{aligned} x^{k+1} &= \text{prox}_{\mu_k h}\left(z^k - \mu_k \tilde{\nabla} Q_{\rho_k}(x^k)\right) \\ &= \underset{x \in \mathbb{R}^n}{\text{argmin}} \Big\{ \langle \tilde{\nabla} Q_{\rho_k}(x^k), x - x^k \rangle + h(x) \\ &\qquad + \frac{1}{2\mu_k} \|x - z^k\|^2 \Big\}, \end{aligned} \quad (3.4)$$

**Algorithm 1** MoSSP-P: **S**ingle-Loop **S**tochastic **P**enalty Algorithm with **P**olyak **Mo**mentum

---

**Input:** maximum number of iterations $K$, initial point $x^0 = z^0 \in \mathbb{R}^n$, a sequence $\{\alpha_k\} \subset (0, 1)$, positive parameters $\mu_k$, $\rho_k$, and $\beta$.
**for** $k = 0, 1, 2, \ldots K - 1$ **do**
  Sample $\xi^k$ from $\Xi$ and compute $S^k$ from (3.6).
  Compute $x^{k+1}$ using (3.8).
  Compute $z^{k+1}$ using (3.5).
**end for**
**Output:** $x^{R+1}$, where $R \in \{0, 1, \ldots, K - 1\}$ is uniformly and randomly chosen.

---

where $\tilde{\nabla} Q_{\rho_k}(x^k)$ is a stochastic estimator of $\nabla Q_{\rho_k}(x^k)$. Finally, $z^k$ is updated via a gradient-type step, where the gradient (3.3) is now partially estimated by replacing $\text{prox}_{\mu_k\psi_{\rho_k}}(z^k)$ with $x^{k+1}$ from (3.4):

$$z^{k+1} = z^k - \beta\left(\text{prox}_{\mu_k g}(z^k) - x^{k+1}\right), \quad (3.5)$$

where $\beta > 0$ is the stepsize. All parameter sequences are positive and will be specified later.

Our framework allows for flexible choices of $\tilde{\nabla} Q_{\rho_k}(x^k)$ in the $x$-update step (see (3.4)), enabling the integration of advanced momentum techniques to enhance both theoretical complexity and practical efficiency. We refer to this unified framework as **MoSSP** (**Mo**mentum-based **S**ingle-loop **S**tochastic **P**enalty).

### 3.2. MoSSP-P: MoSSP with Polyak Momentum

Polyak momentum, also known as the heavy-ball method and originally proposed by Polyak (1964), has been extensively studied in large-scale nonconvex optimization (Liu et al., 2020; Jelassi & Li, 2022; Gao et al., 2024). Motivated by this, at each iteration $k$, we draw an i.i.d. sample $\xi^k \sim \Xi$ and construct a Polyak momentum-based stochastic estimator for $\nabla Q_{\rho_k}(x^k)$:

$$S^k = s^k + \rho_k \nabla c(x^k) c(x^k), \quad (3.6)$$

where $s^k$ is updated by

$$s^k = \begin{cases} (1 - \alpha_{k-1}) s^{k-1} + \alpha_{k-1} \nabla \mathbf{f}(x^k, \xi^k), & k \geq 1, \\ \nabla \mathbf{f}(x^0, \xi^0), & k = 0. \end{cases} \quad (3.7)$$

The variable $x^{k+1}$ is then updated as:

$$x^{k+1} = \text{prox}_{\mu_k h}\left(z^k - \mu_k S^k\right), \quad (3.8)$$

where $\mu_k > 0$ for $k \geq 0$. The complete algorithm, referred to as **MoSSP-P**, is summarized in Algorithm 1.

**Oracle Complexity of MoSSP-P.** We now analyze the oracle complexity of MoSSP-P for finding a stochastic $\varepsilon$-KKT point and a stochastic $\varepsilon$-stationary point, respectively. The maximum number of iterations is limited to a fixed integer $K$.

Note that $Q_\rho(\boldsymbol{x})$ is smooth with Lipschitz constant $L_\rho$, where $L_\rho = \rho\tilde{L}$ and $\tilde{L} = \rho_0^{-1}L_f + G^2 + CL_c$, which is crucial for the subsequent analysis (see Lemma C.1).

We aim to bound the residuals in (2.6) and (2.7). Classical analysis of penalty methods in nonconvex constrained optimization often relies on the descent property of the penalty function. However, the concave component $-g$ in our penalty function $F_{\rho_k}$ breaks this descent property, rendering the standard potential function construction ineffective. To this end, we construct the following potential function:

$$\mathcal{L}_{\rho,\mu}(\boldsymbol{w}) = Q_\rho(\boldsymbol{x}) + h(\boldsymbol{x}) + \frac{1}{2\mu}\|\boldsymbol{x} - \boldsymbol{z}\|^2 - \mathcal{M}_{\mu g}(\boldsymbol{z}),$$

where $\boldsymbol{w} = (\boldsymbol{x}, \boldsymbol{z})$. The function $\mathcal{L}_{\rho,\mu}(\boldsymbol{w})$ is bounded from below (see Appendix C.2).

To quantify the criticality of iterates $\boldsymbol{w}^k$ generated by MoSSP-P, we define $\boldsymbol{u}^{k+1}$ as:

$$\boldsymbol{u}^{k+1} = \nabla Q_{\rho_k}(\boldsymbol{x}^{k+1}) - \boldsymbol{S}^k + \mu_k^{-1}(\text{prox}_{\mu_k g}(\boldsymbol{z}^k) - \boldsymbol{x}^{k+1}). \tag{3.9}$$

It can be proved that $\boldsymbol{u}^{k+1} \in \partial\psi_{\rho_k}(\boldsymbol{x}^{k+1}) - \partial g(\text{prox}_{\mu_k g}(\boldsymbol{z}^k))$ (see Appendix C.3).

Lemma C.3 gives a one-step bound on $\boldsymbol{u}^{k+1}$ relative to the iterate change (see Appendix C.3). The stochastic error in (C.19) implies that proper parameter settings are required to control variance and ensure favorable complexity. This differs from the deterministic framework for constrained DC optimization of Sun & Sun (2023). We next present an estimated convergence rate of MoSSP-P to guide parameter selection.

**Lemma 3.1.** *Suppose Assumptions 2.1 and 2.2 hold. Let $\{\boldsymbol{w}^k\}_{k\in\mathbb{N}}$ be the sequence generated by MoSSP-P with the parameters satisfying*

$$\begin{aligned}\rho_k &\equiv \mathcal{O}(K^l), \quad \mu_k \equiv \mathcal{O}(K^{-\tau}), \\ \alpha_k &\equiv \mathcal{O}(K^{-\tau}), \quad 0 < \beta \le 1,\end{aligned} \tag{3.10}$$

*where $0 < l \le \tau \le 2l < 1$ is independent of $K$. Then, it holds that*

$$\begin{aligned}&\max\{\mathbb{E}[\|\boldsymbol{u}^{R+1}\|^2], \mathbb{E}[\|\boldsymbol{x}^{R+1} - \text{prox}_{\mu_R g}(\boldsymbol{z}^R)\|^2]\} \\ &= \mathcal{O}\left(\max\begin{Bmatrix} K^{\tau-1}\left(\mathcal{L}_{\rho,\mu}(\boldsymbol{w}^0)+1\right), \\ K^{\tau-1}\mathbb{E}[\|\boldsymbol{e}^0\|^2], \quad K^{-\tau} \end{Bmatrix}\right), \\ &\mathbb{E}[\|\nabla\boldsymbol{c}(\boldsymbol{x}^{R+1})\boldsymbol{c}(\boldsymbol{x}^{R+1})\|^2] \\ &= \mathcal{O}\left(\max\begin{Bmatrix} K^{-2l+\tau-1}\left(\mathcal{L}_{\rho,\mu}(\boldsymbol{w}^0)+1\right), \\ K^{-2l+\tau-1}\mathbb{E}[\|\boldsymbol{e}^0\|^2], \quad K^{-2l} \end{Bmatrix}\right).\end{aligned} \tag{3.11}$$

Note that (3.11) indicates that the term $\frac{\rho}{2}\|\boldsymbol{c}(\boldsymbol{x}^0)\|^2$ in $\mathcal{L}_{\rho,\mu}(\boldsymbol{w}^0)$ may dominate the complexity due to $\rho = \rho_0 K^l$. Motivated by Xie & Wright (2021); Jin & Wang (2022), we employ an approximately feasible initialization to mitigate this overhead. Let $\boldsymbol{x}^0$ be an approximately feasible point satisfying $\|\boldsymbol{c}(\boldsymbol{x}^0)\|^2 = \mathcal{O}(K^{-l})$ for some $0 < l < 1$. In practice, such an initialization can be produced by the two-phase procedure proposed by Jin & Wang (2025) for weakly convex minimization with nonconvex constraints, where the first phase yields an approximately feasible point that warm-starts the second. The main complexity result is presented below, with detailed analysis deferred to Appendix C.4.

**Theorem 3.1.** *Suppose Assumptions 2.1-2.3 hold. Let $\{\boldsymbol{w}^k\}_{k\in\mathbb{N}}$ be the sequence generated by MoSSP-P with parameters satisfying*

$$\begin{aligned}\rho_k &\equiv \rho = \rho_0 K^{1/4}, \quad \mu_k \equiv \mu = \frac{\mu_0}{K^{1/2}\max\{L_f, \tilde{L}\}}, \\ \alpha_k &\equiv \alpha = \frac{\alpha_0\mu_0}{K^{1/2}}, \quad 0 < \beta \le 1,\end{aligned} \tag{3.12}$$

*where $\rho_0, \mu_0, \alpha_0 > 0$ are constants independent of $K$. If $\|\boldsymbol{c}(\boldsymbol{x}^0)\|^2 = \mathcal{O}(K^{-1/4})$, then it holds that*

$$\begin{aligned}&\max\{\mathbb{E}[\|\boldsymbol{u}^{R+1}\|^2], \mathbb{E}[\|\boldsymbol{x}^{R+1} - \text{prox}_{\mu g}(\boldsymbol{z}^R)\|^2]\} \\ &= \mathcal{O}(K^{-1/2}), \\ &\mathbb{E}[\|\boldsymbol{c}(\boldsymbol{x}^{R+1})\|^2] = \mathcal{O}(K^{-1/2}).\end{aligned} \tag{3.13}$$

*Consequently, given $\varepsilon > 0$, the oracle complexity of MoSSP-P to obtain a stochastic $\varepsilon$-KKT point $\boldsymbol{x}^{R+1}$ satisfying (2.6) is of order $\mathcal{O}(\varepsilon^{-4})$. In the absence of Assumption 2.3, we have $\mathbb{E}[\|\nabla\boldsymbol{c}(\boldsymbol{x}^{R+1})\boldsymbol{c}(\boldsymbol{x}^{R+1})\|^2] = \mathcal{O}(K^{-1/2})$, and the oracle complexity of MoSSP-P to obtain a stochastic $\varepsilon$-stationary point $\boldsymbol{x}^{R+1}$ satisfying (2.7) is of order $\mathcal{O}(\varepsilon^{-4})$.*

The result in Theorem 3.1 matches that of unconstrained stochastic first-order methods (SFOMs) for Problem (P) with $\boldsymbol{c} = \boldsymbol{0}$ and $g = 0$ (Ghadimi & Lan, 2013; Gao et al., 2024), as well as SFOMs for the unconstrained stochastic DC case (Chayti & Jaggi, 2025; Hu et al., 2024; Xu et al., 2019).

*Remark* 3.1. Existing analyses of penalty methods, including ALM analyses for nonconvex constrained problems, require penalty parameters to grow inversely with $\varepsilon$ to attain the best-known complexity. This worsens the problem conditioning, as the smoothness constant grows as $L_\rho = \Theta(\rho)$. Without momentum, one must compensate with a smaller stepsize or a larger batch size, potentially yielding complexity no better than $\mathcal{O}(\varepsilon^{-5})$ (Jin & Wang, 2022); in particular, a naive extension of unconstrained stochastic DC methods (e.g., (Hu et al., 2024; Xu et al., 2019)) to the penalty framework for Problem (P) risks such degradation. Under our penalty framework, momentum yields a genuine complexity improvement by allowing the penalty, stepsize, and momentum parameters to be jointly calibrated under $\mathcal{O}(1)$ batch

size, thereby achieving the rate stated above at no additional cost from the nonconvex constraints.

The oracle complexity of MoSSP-P without an approximately feasible initialization is presented as follows.

**Corollary 3.1.** *Under the assumptions of Theorem 3.1 with parameters set as $\rho = \mathcal{O}(K^{1/5})$, $\mu = \mathcal{O}(K^{-2/5})$, $\alpha = \mathcal{O}(K^{-2/5})$, and $0 < \beta \leq 1$, the oracle complexity of MoSSP-P to obtain a stochastic $\varepsilon$-KKT point or a stochastic $\varepsilon$-stationary point (without Assumption 2.3) $\boldsymbol{x}^{R+1}$ is of order $\mathcal{O}(\varepsilon^{-5})$.*

*Remark* 3.2. Corollary 3.1 highlights the role of approximate feasible initialization in improving the complexity order. Since the schedule $\rho = \mathcal{O}(K^l)$ is required to control $\mathbb{E}\left[\|\nabla \boldsymbol{c}(\boldsymbol{x}^k)\boldsymbol{c}(\boldsymbol{x}^k)\|^2\right]$, it inflates $\frac{\rho}{2}\|\boldsymbol{c}(\boldsymbol{x}^0)\|^2$ to $\mathcal{O}(K^l)$ when $\|\boldsymbol{c}(\boldsymbol{x}^0)\|^2 = \mathcal{O}(1)$, dominating the bound in (3.11). Such initialization confines this term to $\mathcal{O}(1)$, thereby improving the complexity order. In fact, our analysis can also be extended to an iteration-indexed penalty update (Alacaoglu & Wright, 2024) without this requirement.

### 3.3. MoSSP-R: MoSSP with Recursive Momentum

Alternatively, we incorporate another variance reduction technique, recursive momentum (STORM) (Cutkosky & Orabona, 2019), into the $\boldsymbol{x}$-update. STORM introduces a correction term based on a single sample to generate a variance-reduced gradient estimate, making it suitable for large-scale applications.

At iteration $k$, we construct a recursive momentum-based stochastic estimator for $\nabla Q_{\rho_k}(\boldsymbol{x}^k)$ as:

$$\boldsymbol{D}^k = \boldsymbol{d}^k + \rho_k \nabla \boldsymbol{c}(\boldsymbol{x}^k)\boldsymbol{c}(\boldsymbol{x}^k), \qquad (3.14)$$

where $\boldsymbol{d}^k$ is updated recursively by

$$\boldsymbol{d}^k = \begin{cases} \frac{1}{|\mathcal{B}_0|}\sum_{j \in \mathcal{B}_0} \nabla \mathbf{f}(\boldsymbol{x}^0, \xi_j^0), & k = 0, \\ \nabla \mathbf{f}(\boldsymbol{x}^k, \xi^k) + (1 - \alpha_{k-1}) & k \geq 1. \\ \quad \left(\boldsymbol{d}^{k-1} - \nabla \mathbf{f}(\boldsymbol{x}^{k-1}, \xi^k)\right), \end{cases}$$
$$(3.15)$$

We begin at $k = 0$ by independently drawing a sample set $\{\xi_j^0\}_{j \in \mathcal{B}_0}$ from the distribution $\Xi$, where the initial batch size is $|\mathcal{B}_0| = b_0$. For any $k \geq 1$, a single i.i.d. sample $\xi^k$ is drawn from $\Xi$. We then update $\boldsymbol{x}^{k+1}$ through

$$\boldsymbol{x}^{k+1} = \operatorname{prox}_{\mu_k h}\left(\boldsymbol{z}^k - \mu_k \boldsymbol{D}^k\right). \qquad (3.16)$$

The overall algorithm, termed **MoSSP-R**, is summarized in Algorithm 2.

**Oracle Complexity of MoSSP-R.** We now present complexity results for MoSSP-R for finding an approximate stationary point and an approximate KKT point, respectively, with detailed analysis deferred to Appendix C.5. In

---

**Algorithm 2 MoSSP-R**: **S**ingle-Loop **S**tochastic **P**enalty Algorithm with **R**ecursive **Mo**mentum

---

**Input:** maximum number of iterations $K$, initial point $\boldsymbol{x}^0 = \boldsymbol{z}^0 \in \mathbb{R}^n$, a sequence $\{\alpha_k\} \subset (0,1)$, positive parameters $\mu_k, \rho_k, \beta$.
**for** $k = 0, 1, 2, \ldots K - 1$ **do**
    Sample $\xi^k$ from $\Xi$ and compute $\boldsymbol{D}^k$ from (3.14).
    Compute $\boldsymbol{x}^{k+1}$ using (3.16).
    Compute $\boldsymbol{z}^{k+1}$ using (3.5).
**end for**
**Output:** $\boldsymbol{x}^{R+1}$, where $R \in \{0, 1, \ldots, K - 1\}$ is uniformly chosen.

---

MoSSP-R, we construct $\boldsymbol{u}^{k+1}$ as:

$$\boldsymbol{u}^{k+1} = \nabla Q_{\rho_k}(\boldsymbol{x}^{k+1}) - \boldsymbol{D}^k + \mu_k^{-1}(\operatorname{prox}_{\mu_k g}(\boldsymbol{z}^k) - \boldsymbol{x}^{k+1}). \qquad (3.17)$$

Before proceeding, we introduce an additional standard assumption, also known as the mean-squared smoothness assumption, widely used in variance reduction methods (Nguyen et al., 2017; Fang et al., 2018; Cutkosky & Orabona, 2019; Xu & Xu, 2023).

**Assumption 3.1.** For almost every $\xi \in \Xi$, $\mathbf{f}(\cdot, \xi)$ is differentiable and for all $\boldsymbol{x}, \boldsymbol{y} \in \mathbb{R}^n$,

$$\mathbb{E}_\xi\left[\|\nabla \mathbf{f}(\boldsymbol{x}; \xi) - \nabla \mathbf{f}(\boldsymbol{y}; \xi)\|^2\right] \leq L_f^2 \|\boldsymbol{x} - \boldsymbol{y}\|^2.$$

By Jensen's inequality, Assumption 3.1 implies that $f = \mathbb{E}_\xi[\mathbf{f}(\cdot; \xi)]$ is $L_f$-smooth, whereas the converse does not hold in general.

As noted by Xu & Xu (2023), employing a moderately large initial batch size in variance reduction can improve the complexity order at a negligible cost. Consequently, we establish the oracle complexity with an initial batch size $b_0 = \mathcal{O}(K^{1/3})$.

**Theorem 3.2.** *Suppose Assumptions 2.1-2.3 and 3.1 hold. Let $\{\boldsymbol{w}^k\}_{k \in \mathbb{N}}$ be the sequence generated by MoSSP-R with parameters satisfying*

$$\rho_k \equiv \rho = \rho_0 K^{1/3}, \quad \mu_k \equiv \mu = \frac{\mu_0}{K^{1/3}\max\{L_f, \tilde{L}\}},$$

$$\alpha_k \equiv \alpha = \frac{16\alpha_0 \mu_0^2}{K^{2/3}}, \quad 0 < \beta \leq 1, \qquad (3.18)$$

*where $\rho_0, \mu_0, \alpha_0 > 0$ are constants independent of $K$. If $\|\boldsymbol{c}(\boldsymbol{x}^0)\|^2 = \mathcal{O}(K^{-1/3})$ with the initial batch size $b_0 = \mathcal{O}(K^{1/3})$, then it holds that*

$$\max\{\mathbb{E}[\|\boldsymbol{u}^{R+1}\|^2], \mathbb{E}[\|\boldsymbol{x}^{R+1} - \operatorname{prox}_{\mu g}(\boldsymbol{z}^R)\|^2]\}$$
$$= \mathcal{O}(K^{-2/3}), \qquad (3.19)$$
$$\mathbb{E}\left[\|\boldsymbol{c}(\boldsymbol{x}^{R+1})\|^2\right] = \mathcal{O}(K^{-2/3}).$$

*Consequently, given $\varepsilon > 0$, the oracle complexity of MoSSP-R to obtain a stochastic $\varepsilon$-KKT point $\boldsymbol{x}^{R+1}$ satisfying (2.6) is of order $\mathcal{O}(\varepsilon^{-3})$. In the absence of Assumption 2.3, we have $\mathbb{E}\left[\|\nabla \boldsymbol{c}(\boldsymbol{x}^{R+1})\, \boldsymbol{c}(\boldsymbol{x}^{R+1})\|^2\right] = \mathcal{O}(K^{-2/3})$, and the oracle complexity of MoSSP-R to obtain a stochastic $\varepsilon$-stationary point $\boldsymbol{x}^{R+1}$ satisfying (2.7) is of order $\mathcal{O}(\varepsilon^{-3})$.*

The result in Theorem 3.2 matches the $\mathcal{O}(\varepsilon^{-3})$ lower bound for unconstrained SFOMs with variance reduction (Arjevani et al., 2023) and the best-known rate for nonconvex constrained stochastic optimization (Shi et al., 2026) under Assumption 3.1, where the DC structure is absent. Crucially, it demonstrates that our single-loop penalty framework and analysis can effectively handle the DC structure without increasing the complexity order with respect to $\varepsilon$.

The following corollary establishes the oracle complexity result for MoSSP-R without assuming initial approximate feasibility.

**Corollary 3.2.** *Under the assumptions of Theorem 3.2 with parameters set as $\rho = \mathcal{O}(K^{1/4})$, $\mu = \mathcal{O}(K^{-1/4})$, $\alpha = \mathcal{O}(K^{-1/2})$, $0 < \beta \leq 1$, the oracle complexity of MoSSP-R to obtain a stochastic $\varepsilon$-KKT point or a stochastic $\varepsilon$-stationary point (without Assumption 2.3) $\boldsymbol{x}^{R+1}$ is of order $\mathcal{O}(\varepsilon^{-4})$.*

# 4. Numerical Experiments

## 4.1. Experiment Setup

We evaluate MoSSP-P (Algorithm 1) and MoSSP-R (Algorithm 2) on an equality-constrained binary classification problem with DC regularization (Hong et al., 2023):

$$\min_{\boldsymbol{x}\in\mathbb{R}^n} \quad \frac{1}{N}\sum_{i=1}^{N} \log(1 + e^{-y_i(X_i^\top \boldsymbol{x})}) + \lambda\left(\|\boldsymbol{x}\|_1 - \|\boldsymbol{x}\|_2\right)$$
$$\text{s.t.} \quad \|\boldsymbol{x}\|_2^2 = 1, \tag{4.1}$$

where $X_i \in \mathbb{R}^n$ denotes the feature vector and $y_i \in \{-1, 1\}$ denotes the binary label for each $i \in [N]$, with $N$ denoting the total sample size. We compare our algorithms with two double-loop baselines: SPDC (Xu et al., 2019; Nitanda & Suzuki, 2017) for simple convex-constrained DC(-regularized) optimization and SALM (Sun & Sun, 2023) for linearly constrained DC-regularized optimization. Each baseline is equipped with Polyak momentum (P) or recursive momentum (R), yielding four variants: SPDC-P, SPDC-R, SALM-P, SALM-R. We evaluate on three LIBSVM datasets (Chang & Lin, 2011): a9a (32,561 samples, 123 features), phishing (11,055 samples, 68 features), and australian (690 samples, 14 features). The monitored metrics are objective value $F(\boldsymbol{x}^k)$ and constraint violation $\|\boldsymbol{x}^k\|_2^2 - 1|$. Figure 1 shows convergence trajectories on the a9a dataset (averaged over five runs), while Tables 2a and 2b report final values as mean $\pm$ std.

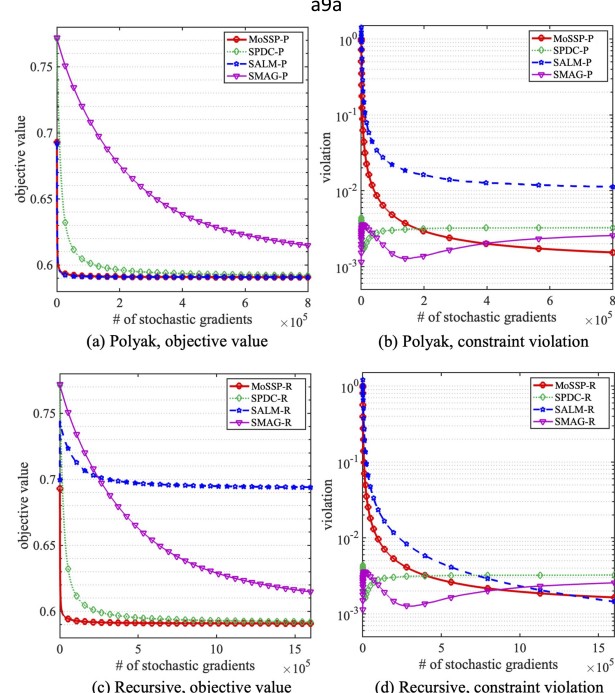

*Figure 1.* Comparison of MoSSP variants, SPDC, and SALM on a9a. (a), (b) Polyak momentum: objective value and constraint violation. (c), (d) Recursive momentum: objective value and constraint violation.

We also evaluate on Problem (D.1) with $M$ quadratic equality constraints; the results are reported in Appendix D.3. Complete results for the main experiments on all datasets, together with further details of the experimental setup, are provided in Appendices D.1 and D.2.

## 4.2. Results Analysis

Figure 1 shows that the MoSSP variants converge much faster than the baselines on the large-scale a9a dataset. Most notably, the MoSSP variants quickly reduce early-iterate constraint violations from $10^0$ to $10^{-3}$ with only a small number of gradient evaluations, demonstrating the efficiency of the single-loop framework. In contrast, SPDC and SALM are much slower at achieving feasibility, with violations remaining at $10^{-2}$ or higher. This speed advantage is consistently observed in both the Polyak momentum (Figure 1(a)–(b)) and recursive momentum (Figure 1(c)–(d)) variants, confirming that the gain comes from the single-loop design itself.

Tables 2a and 2b provide quantitative evidence. Both MoSSP-P and MoSSP-R achieve competitive objective values and final constraint violations across the tested datasets, showing that the proposed framework is robust to the choice of momentum technique. On the large-scale a9a and phishing datasets, the MoSSP variants rapidly reduce

*Table 2.* Final performance comparison on three LIBSVM datasets. Results are reported over five independent runs. Bold font denotes the best result.

*(a)* Mean $\pm$ std of objective value (Obj. Value) and constraint violation (Const. Viol.) for MoSSP-P and two baseline methods with Polyak momentum.

| DATASET | METRIC | MoSSP-P | SPDC-P | SALM-P |
|---|---|---|---|---|
| A9A | OBJ. VALUE | $\mathbf{0.5901 \pm 4.2 \times 10^{-5}}$ | $0.5917 \pm 1.9 \times 10^{-4}$ | $0.5917 \pm 1.17 \times 10^{-4}$ |
| | CONST. VIOL. | $\mathbf{1.53 \times 10^{-3} \pm 1.24 \times 10^{-5}}$ | $3.23 \times 10^{-2} \pm 1.65 \times 10^{-5}$ | $1.30 \times 10^{-2} \pm 2.52 \times 10^{-4}$ |
| PHISHING | OBJ. VALUE | $\mathbf{0.6045 \pm 1.6 \times 10^{-5}}$ | $0.6051 \pm 4.8 \times 10^{-5}$ | $0.6047 \pm 1.0 \times 10^{-6}$ |
| | CONST. VIOL. | $\mathbf{3.40 \times 10^{-3} \pm 9.2 \times 10^{-6}}$ | $3.66 \times 10^{-2} \pm 5.73 \times 10^{-5}$ | $1.21 \times 10^{-2} \pm 1.52 \times 10^{-4}$ |
| AUSTRALIAN | OBJ. VALUE | $\mathbf{0.6239 \pm 3.6 \times 10^{-5}}$ | $0.6265 \pm 6.74 \times 10^{-4}$ | $0.6241 \pm 1.3 \times 10^{-5}$ |
| | CONST. VIOL. | $7.14 \times 10^{-3} \pm 8.65 \times 10^{-5}$ | $\mathbf{1.95 \times 10^{-3} \pm 2.92 \times 10^{-5}}$ | $2.19 \times 10^{-2} \pm 5.11 \times 10^{-4}$ |

*(b)* Mean $\pm$ std of objective value (Obj. Value) and constraint violation (Const. Viol.) for MoSSP-R and two baseline methods with recursive momentum.

| DATASET | METRIC | MoSSP-R | SPDC-R | SALM-R |
|---|---|---|---|---|
| A9A | OBJ. VALUE | $\mathbf{0.5809 \pm 3.8 \times 10^{-4}}$ | $0.5921 \pm 2.14 \times 10^{-4}$ | $0.6303 \pm 4.23 \times 10^{-3}$ |
| | CONST. VIOL. | $\mathbf{1.03 \times 10^{-3} \pm 1.24 \times 10^{-5}}$ | $3.23 \times 10^{-2} \pm 7.35 \times 10^{-6}$ | $1.30 \times 10^{-2} \pm 1.17 \times 10^{-4}$ |
| PHISHING | OBJ. VALUE | $\mathbf{0.6045 \pm 1.1 \times 10^{-5}}$ | $0.6052 \pm 4.7 \times 10^{-5}$ | $0.6054 \pm 9.7 \times 10^{-5}$ |
| | CONST. VIOL. | $\mathbf{4.63 \times 10^{-3} \pm 7.12 \times 10^{-6}}$ | $1.10 \times 10^{-1} \pm 5.68 \times 10^{-4}$ | $7.42 \times 10^{-3} \pm 2.88 \times 10^{-4}$ |
| AUSTRALIAN | OBJ. VALUE | $\mathbf{0.6240 \pm 6.8 \times 10^{-5}}$ | $0.6265 \pm 6.74 \times 10^{-4}$ | $0.6243 \pm 1.5 \times 10^{-5}$ |
| | CONST. VIOL. | $6.88 \times 10^{-3} \pm 1.03 \times 10^{-4}$ | $\mathbf{1.94 \times 10^{-3} \pm 2.13 \times 10^{-5}}$ | $1.92 \times 10^{-2} \pm 4.86 \times 10^{-4}$ |

constraint violations and achieve competitive final feasibility. On the smaller `australian` dataset, SPDC attains a lower final constraint violation, as discussed in Appendix D.2. Overall, the proposed single-loop momentum framework achieves fast objective decrease and competitive feasibility control for nonconvex constrained DC-regularized optimization.

## 5. Conclusion and Discussion

This paper proposes **MoSSP**, a simple single-loop stochastic penalty framework for general nonconvex constrained DC-regularized optimization. We develop two momentum-based variants, MoSSP-P and MoSSP-R, and establish their oracle complexity orders for finding two types of stochastic approximate solutions for Problem (**P**). Our analysis shows that exploiting the DC structure does not worsen the complexity order, with or without the constraint qualification. To the best of our knowledge, this is the first systematic complexity study of single-loop stochastic methods for nonconvex, functionally constrained optimization involving DC structure.

Although we focus on equality constraints in Problem (**P**), the framework extends directly to inequality constraints:

$$\min_{\boldsymbol{x} \in \mathbb{R}^n} \quad F(\boldsymbol{x}) = f(\boldsymbol{x}) + h(\boldsymbol{x}) - g(\boldsymbol{x}),$$
$$\text{s.t.} \quad \boldsymbol{c}_{\mathcal{E}}(\boldsymbol{x}) = \boldsymbol{0}, \ \ \boldsymbol{c}_{\mathcal{I}}(\boldsymbol{x}) \leq \boldsymbol{0},$$

where $\boldsymbol{c}_{\mathcal{E}} : \mathbb{R}^n \to \mathbb{R}^{m_1}$ and $\boldsymbol{c}_{\mathcal{I}} : \mathbb{R}^n \to \mathbb{R}^{m_2}$ are smooth mappings, and $f, h, g$ are as before. The quadratic penalty

(3.1) becomes

$$Q_\rho(\boldsymbol{x}) = f(\boldsymbol{x}) + \tfrac{\rho}{2}\big(\|\boldsymbol{c}_{\mathcal{E}}(\boldsymbol{x})\|^2 + \|[\boldsymbol{c}_{\mathcal{I}}(\boldsymbol{x})]_+\|^2\big),$$

where $[\cdot]_+ := \max\{\cdot, 0\}$ is applied componentwise. Correspondingly, the gradient estimators in (3.6) and (3.14) are updated by replacing $\nabla \boldsymbol{c}(\boldsymbol{x}^k)\boldsymbol{c}(\boldsymbol{x}^k)$ with $\boldsymbol{J}(\boldsymbol{x}^k) := \nabla \boldsymbol{c}_{\mathcal{E}}(\boldsymbol{x}^k)\boldsymbol{c}_{\mathcal{E}}(\boldsymbol{x}^k) + \nabla \boldsymbol{c}_{\mathcal{I}}(\boldsymbol{x}^k)[\boldsymbol{c}_{\mathcal{I}}(\boldsymbol{x}^k)]_+$. This replacement preserves the smoothness of $Q_\rho$, and the error-bound condition in Assumption 2.3 can be replaced by

$$\big\|\boldsymbol{J}(\boldsymbol{x}^k)\big\| \geq \delta \big(\|\boldsymbol{c}_{\mathcal{E}}(\boldsymbol{x}^k)\|^2 + \|[\boldsymbol{c}_{\mathcal{I}}(\boldsymbol{x}^k)]_+\|^2\big)^{1/2}, \forall k \geq 0.$$

A natural direction is to extend our analysis framework to general DC optimization under nonlinear constraints, where the concave component admits the form $-\mathbb{E}_\xi[g(\boldsymbol{x}; \xi)]$ and $\text{prox}_{\mu g}(\cdot)$ is no longer available in closed form, as in problems such as PU learning and partial AUC maximization.

## Acknowledgements and Disclosure of Funding

Xiao Wang is supported by the National Natural Science Foundation of China (No. 12271278). Luxuan Li and Chunfeng Cui are supported by the National Natural Science Foundation of China (Nos. 12471282 and 12131004). The authors would like to thank Dr. Lei Yang for helpful discussions and insightful comments. The authors are grateful to the Area Chairs and the anonymous reviewers for their constructive comments.

## Impact Statement

This paper presents work whose goal is to advance the field of Machine Learning. There are many potential societal consequences of our work, none of which we feel must be specifically highlighted here.

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

# Contents of Appendices

# A. More Related Work

## A.1. Constrained DC and DC-Regularized Optimization

DC structures in learning problems commonly arise from two different sources. First, they may appear as an algebraic consequence of max-structured or minimax-type risk formulations, such as positive-unlabeled learning (Kiryo et al., 2017), partial AUC optimization (Yao et al., 2022), and related adversarial formulations, where the objective can often be rewritten or approximated as a DC function. Second, DC structure is deliberately introduced through sparsity-promoting nonconvex regularizers, such as capped-$\ell_1$, SCAD, MCP, or $\ell_1 - \ell_2$, which approximate $\ell_0$-type sparsity while retaining a tractable convex-concave decomposition; see, e.g., (Gong et al., 2013, Table 1).

In the deterministic setting, several methods have been developed for constrained DC and DC-regularized optimization problems. Zhou et al. (2024) proposed a proximal ADMM for structured DC programs whose objective is the difference of two possibly nonsmooth convex functions and whose constraints are given by a linear mapping into a closed convex set. Under the Kurdyka–Łojasiewicz property, they established convergence of the generated sequence to a critical point. Lu et al. (2022) developed penalty and augmented-Lagrangian methods for structured nonsmooth constrained DC programs with DC inequality constraints. Under a pointwise Slater condition, they showed that any feasible accumulation point generated by the penalty method is B-stationary for the original problem; the augmented-Lagrangian variant further yields KKT-type optimality conditions together with accumulation points of auxiliary multiplier sequences.

Several related deterministic schemes, including augmented-Lagrangian, sequential-convexification, and moving-balls-approximation (MBA) type methods, have also been studied for inequality-constrained DC or DC-regularized problems (Kanzow & Neder, 2026; Liu et al., 2025; Yu et al., 2021). Specifically, Kanzow & Neder (2026) proposed a safeguarded augmented Lagrangian method for DC optimization with linear equality constraints and convex inequality constraints. Under a modified Slater constraint qualification, they established subsequential convergence to generalized KKT points. Nevertheless, their theoretical analysis relies on the convexity of the constraints, making its extension to nonconvex feasible sets such as Problem (P) challenging. Yu et al. (2021) studied a line-search variant of MBA-type algorithms for DC-regularized problems with smooth inequality constraints. Under the MFCQ, they established convergence to stationary points when the constructed potential function satisfies the KL property. More recently, Liu et al. (2025) developed an inexact MBA method for DC-regularized optimization coupled with differentiable inequality constraints whose gradients are locally Lipschitz continuous. Under the MSCQ, the partial bounded multiplier property, boundedness of the iterates, and the KL property of the constructed potential function, they achieved full sequence convergence to strong stationary points. They also established an iteration complexity of $\mathcal{O}(\varepsilon^{-2})$ for finding an $\varepsilon$-KKT point under the MFCQ and smoothness assumptions on the constraints. However, the MBA-type methods above are designed for inequality-constrained problems and rely on feasibility/interiority-type assumptions. Hence, they do not directly apply to equality-constrained feasible sets, such as Problem (P).

## A.2. Stochastic Nonconvex Constrained Optimization

For stochastic optimization with nonconvex constraints, such as Problem (P) with $g = 0$, stochastic penalty methods, including augmented Lagrangian (AL) approaches, have been extensively studied (Kushner & Sanvicente, 1974; Wang et al., 2017; Xu, 2020). Stochastic sequential quadratic programming (SQP) methods represent another prominent class of algorithms for addressing equality-constrained stochastic problems. Recent advances in stochastic SQP encompass convergence guarantees (Berahas et al., 2021; Na et al., 2023b), analysis in expectation (Na et al., 2023a), almost-sure convergence (Curtis et al., 2025), and worst-case complexity bounds (Berahas et al., 2022; Na & Mahoney, 2025; Curtis et al., 2024). Notably, these methods differ in their computational structure: penalty methods require updating the penalty parameter in the outer loop and (approximately) minimizing the penalized subproblem in the inner loop, while SQP methods compute search directions by solving a sequence of quadratic programming subproblems.

Alternatively, single-loop penalty methods have gained traction for such problems, and many studies have focused on different settings of Problem (P). For Problem (P) with $h - g = 0$, Alacaoglu & Wright (2024) proposed a linearized quadratic penalty approach incorporating recursive momentum (Cutkosky & Orabona, 2019), achieving a sample complexity of $\tilde{\mathcal{O}}(\varepsilon^{-4})$ via adaptive penalty parameters. To identify a stronger $\varepsilon$-stationary point with constraint violation at most $\varepsilon$, Lu et al. (2026) introduced a linearized quadratic penalty method with truncated momentum. Under a regularity condition on constraint gradients, they achieved a nearly optimal complexity of $\tilde{\mathcal{O}}(\varepsilon^{-3})$ using recursive momentum, whereas Polyak momentum yields $\tilde{\mathcal{O}}(\varepsilon^{-4})$ with diminishing step sizes. Recently, Zuo et al. (2025) studied an adaptive single-loop stochastic penalty method for solving nonconvex optimization with equality and inequality constraints, proposing an adaptive penalty

parameter update scheme and establishing an $\mathcal{O}(\varepsilon^{-4})$ oracle complexity bound under a local CQ condition. For Problem (P) with $g = 0$, Jin & Wang (2022) proposed a single-loop stochastic primal-dual method based on the linearized AL function for nonconvex problems with numerous functional constraints. To find an $\varepsilon$-KKT point, their approach achieves an oracle complexity of $\mathcal{O}(\varepsilon^{-6})$ starting from an arbitrary point, and $\mathcal{O}(\varepsilon^{-5})$ with a feasible initial point. Subsequently, under certain constraint qualifications, Shi et al. (2026) developed a linearized AL method, establishing an oracle complexity of $\mathcal{O}(\varepsilon^{-4})$ (or a nearly optimal $\mathcal{O}(\varepsilon^{-3})$ with a nearly feasible initial point).

Within the exact penalty framework, Yang et al. (2026) proposed a single-loop stochastic algorithm based on a hinge penalty method for Problem (P) with weakly convex objective and constraint functions, achieving an $\mathcal{O}(\varepsilon^{-6})$ oracle complexity for finding a near-$\varepsilon$-KKT point under a regularity condition. By combining the SPIDER-type variance reduction technique (Fang et al., 2018), Liu & Xu (2025) presented a single-loop stochastic subgradient method for nonconvex nonsmooth stochastic optimization with weak inequality constraints in expectation. Under a (uniform) Slater-type constraint qualification, the method achieves an $\mathcal{O}(\varepsilon^{-4})$ sample complexity for evaluations of both the objective and constraint function subgradients, and $\mathcal{O}(\varepsilon^{-6})$ for evaluations of the constraint function values to produce an $(\varepsilon, \varepsilon)$-KKT point. For problems with deterministic equality constraints, Cui et al. (2025) proposed a method that achieves an $\mathcal{O}(\varepsilon^{-3})$ sample complexity to find an $\varepsilon$-KKT point.

### A.3. Unconstrained Stochastic DC Optimization

For the unconstrained case of Problem (P), while deterministic DC algorithms have been extensively studied (Le Thi & Pham Dinh, 2018), their stochastic counterparts (SDCA) with non-asymptotic convergence analysis remain relatively less explored, appearing only in a few works such as (Nitanda & Suzuki, 2017; Xu et al., 2019). However, these analyses require smoothness (Nitanda & Suzuki, 2017) or Hölder continuity assumptions on the gradients of DC components (Xu et al., 2019), which may be restrictive for many nonsmooth functions, such as the $\ell_{1-2}$ norm. The Moreau envelope provides a powerful smoothing technique for handling nonsmoothness via the proximal operator and has been widely used in stochastic weakly convex optimization (Davis & Drusvyatskiy, 2019). Subsequently, Sun & Sun (2023) applied Moreau envelope smoothing to both DC components, enabling non-asymptotic convergence analysis for unconstrained DC problems under a relaxed criticality criterion. Hu et al. (2024) further proposed a single-loop stochastic method for nonsmooth difference-of-weakly-convex problems with an $\mathcal{O}(\varepsilon^{-4})$ oracle complexity; however, their setting is unconstrained and their DWC analysis requires uniformly bounded second moments of stochastic subgradients for both components (see Assump. 4.6(iii) in Hu et al. (2024)). More recently, Chayti & Jaggi (2025) introduced momentum-based variance reduction for stochastic DC optimization, but their single-loop analysis assumes a smooth concave component and smooths only the convex component, which differs from our setting where the concave component in DC structure can be nonsmooth and the problem is further subject to nonconvex constraints.

## B. Preliminaries

For the convergence analysis, we first introduce several definitions and preliminary results for nonsmooth DC optimization and Moreau-envelope smoothing. We use the following definitions of general subgradient and subdifferential (Davis & Drusvyatskiy, 2019; Rockafellar & Wets, 2009).

**Definition B.1.** Consider a function $f : \mathbb{R}^n \to \mathbb{R} \cup \{+\infty\}$ and a point $\boldsymbol{x} \in \mathbb{R}^n$, with $f(\boldsymbol{x})$ being finite. A vector $\boldsymbol{v} \in \mathbb{R}^n$ is a general subgradient of $f$ at $\boldsymbol{x}$ if

$$f(\boldsymbol{y}) \geq f(\boldsymbol{x}) + \langle \boldsymbol{v}, \boldsymbol{y} - \boldsymbol{x} \rangle + o(\|\boldsymbol{y} - \boldsymbol{x}\|) \text{ as } \boldsymbol{y} \to \boldsymbol{x}.$$

The subdifferential $\partial f(\boldsymbol{x})$ is the set of subgradients of $f$ at $\boldsymbol{x}$. For a continuously differentiable function $f$, $\partial f(\boldsymbol{x}) = \{\nabla f(\boldsymbol{x})\}$; for convex functions, this coincides with the convex subdifferential. For notational simplicity, we abuse the notation $\partial f(\boldsymbol{x})$ to denote an arbitrary subgradient from the corresponding subdifferential when the context is clear.

A mapping $\mathcal{M} : \mathcal{D} \to \mathbb{R}^l$ is said to be $C$-Lipschitz continuous if $\|\mathcal{M}(\boldsymbol{x}) - \mathcal{M}(\boldsymbol{x}')\| \leq C\|\boldsymbol{x} - \boldsymbol{x}'\|$ for all $\boldsymbol{x}, \boldsymbol{x}' \in \mathcal{D}$. A function $f : \mathbb{R}^n \to \mathbb{R}$ is said to be $L_f$-smooth if it is continuously differentiable and has an $L_f$-Lipschitz continuous gradient, that is $\|\nabla f(\boldsymbol{x}) - \nabla f(\boldsymbol{x}')\| \leq L_f\|\boldsymbol{x} - \boldsymbol{x}'\|$ for all $\boldsymbol{x}, \boldsymbol{x}' \in \mathbb{R}^n$. It then satisfies the following inequality:

$$f(\boldsymbol{x}) \leq f(\boldsymbol{x}') + \langle \nabla f(\boldsymbol{x}'), \boldsymbol{x} - \boldsymbol{x}' \rangle + \frac{L_f}{2}\|\boldsymbol{x} - \boldsymbol{x}'\|^2, \quad \forall \boldsymbol{x}, \boldsymbol{x}' \in \mathbb{R}^n.$$

A function $\phi : \mathbb{R}^n \to \mathbb{R} \cup \{\infty\}$ is $m_\phi$-weakly convex if $\phi(\boldsymbol{x}) + \frac{m_\phi}{2}\|\boldsymbol{x}\|^2$ is convex.

### B.1. Properties of *Moreau Envelope*

Regarding the definition of the *Moreau envelope* of $\phi$ in (2.2), it is straightforward to verify that

$$\mathcal{M}_{\mu\phi}(\boldsymbol{z}) = \phi(\text{prox}_{\mu\phi}(\boldsymbol{z})) + \frac{1}{2\mu}\|\text{prox}_{\mu\phi}(\boldsymbol{z}) - \boldsymbol{z}\|^2.$$

For any $\mu \in (0, \frac{1}{m_\phi})$, $\text{prox}_{\mu\phi}(\boldsymbol{z})$ is unique due to the strong convexity of the minimization in (2.2) and is $\frac{1}{1-\mu m_\phi}$-Lipschitz continuous. It is well known that if $\phi$ is convex, then its Moreau envelope $\mathcal{M}_{\mu\phi}$ is also convex; see, e.g., (Rockafellar & Wets, 2009, Theorem 2.26). Moreover, $\mathcal{M}_{\mu\phi}$ maintains the global minimization structure of $\phi$, i.e.,

$$\min_{\boldsymbol{x}} \phi(\boldsymbol{x}) \leq \phi(\text{prox}_{\mu\phi}(\boldsymbol{z})) \leq \mathcal{M}_{\mu\phi}(\boldsymbol{z}) \leq \phi(\boldsymbol{z}) \quad \text{for all } \boldsymbol{z}. \tag{B.1}$$

### B.2. Criticality in DC Optimization

Consider the unconstrained DC optimization problem:

$$\min_{\boldsymbol{x}\in\mathbb{R}^n} \Psi(\boldsymbol{x}) := \phi(\boldsymbol{x}) - g(\boldsymbol{x}). \tag{B.2}$$

A point $\bar{\boldsymbol{x}} \in \mathbb{R}^n$ is called a *critical point* (de Oliveira, 2019) of Problem (B.2) if

$$\mathbf{0} \in \partial\phi(\bar{\boldsymbol{x}}) - \partial g(\bar{\boldsymbol{x}}), \text{ or equivalently, } \partial\phi(\bar{\boldsymbol{x}}) \cap \partial g(\bar{\boldsymbol{x}}) \neq \emptyset. \tag{B.3}$$

In particular, when $g$ is continuously differentiable or constant, (B.3) simplifies to the general stationarity condition $\mathbf{0} \in \partial\Psi(\bar{\boldsymbol{x}})$ (Rockafellar & Wets, 2009). Since it is generally hard to find an exact critical point in a finite number of iterations, iterative algorithms normally pursue an $\varepsilon$-*critical point* $\bar{\boldsymbol{x}}$ for a given $\varepsilon > 0$, satisfying

$$\text{dist}\left(\partial g(\bar{\boldsymbol{x}}), \partial\phi(\bar{\boldsymbol{x}})\right) \leq \varepsilon,$$

or equivalently, there exist $\bar{\boldsymbol{u}}_1 \in \partial\phi(\bar{\boldsymbol{x}})$ and $\bar{\boldsymbol{u}}_2 \in \partial g(\bar{\boldsymbol{x}})$ such that $\|\bar{\boldsymbol{u}}_1 - \bar{\boldsymbol{u}}_2\| \leq \varepsilon$. When $g$ is non-differentiable, however, finding such a point remains challenging (Xu et al., 2019). Instead, Xu et al. (2019) focuses on seeking a *nearly $\varepsilon$-stationary point* $\bar{\boldsymbol{x}}$; that is, there exists $\boldsymbol{x}^*$ such that $\|\bar{\boldsymbol{x}} - \boldsymbol{x}^*\| \leq \mathcal{O}(\varepsilon)$ and $\text{dist}(\partial g(\boldsymbol{x}^*), \partial\phi(\boldsymbol{x}^*)) \leq \varepsilon$.

In this paper, we introduce the notion of $\varepsilon$-*critical points* (Moudafi, 2023; Sun & Sun, 2023), defined as follows.

**Definition B.2** ($\varepsilon$-critical point). *Given $\varepsilon > 0$, a point $\bar{\boldsymbol{x}} \in \mathbb{R}^n$ is called an $\varepsilon$-critical point of Problem (B.2) if there exist $\bar{\boldsymbol{u}} \in \partial\phi(\bar{\boldsymbol{x}}) - \partial g(\bar{\boldsymbol{y}})$ and $\bar{\boldsymbol{y}} \in \mathbb{R}^n$ such that*

$$\max\{\|\bar{\boldsymbol{u}}\|, \|\bar{\boldsymbol{x}} - \bar{\boldsymbol{y}}\|\} \leq \varepsilon. \tag{B.4}$$

A weaker notion of Definition B.2 is used in (Hu et al., 2024; Yao et al., 2022) when proximal mappings of $\phi$ and $g$ can only be solved inexactly.

**Definition B.3** (Nearly $\varepsilon$-critical point (Yao et al., 2022)). *Given $\varepsilon > 0$, a point $\bar{\boldsymbol{x}} \in \mathbb{R}^n$ is called a nearly $\varepsilon$-critical point of Problem (B.2) if there exist $\bar{\boldsymbol{u}} \in \partial\phi(\bar{\boldsymbol{y}}_1) - \partial g(\bar{\boldsymbol{y}}_2)$ and $\bar{\boldsymbol{y}}_1, \bar{\boldsymbol{y}}_2 \in \mathbb{R}^n$ such that*

$$\max\{\|\bar{\boldsymbol{u}}\|, \|\bar{\boldsymbol{x}} - \bar{\boldsymbol{y}}_1\|, \|\bar{\boldsymbol{x}} - \bar{\boldsymbol{y}}_2\|\} \leq \varepsilon. \tag{B.5}$$

Notably, when $\bar{\boldsymbol{y}}_1 = \bar{\boldsymbol{x}}$, Definition B.3 reduces to Definition B.2, and when $\varepsilon = 0$, this definition recovers Equation (B.3).

### B.3. *Difference-of-Moreau-Envelopes* (DME) Smoothing

The DME smoothing approximation of $\Psi$ in Problem (B.2) for any $\mu \in (0, 1/m_\phi)$ is defined as

$$\Psi_\mu(\boldsymbol{z}) := \mathcal{M}_{\mu\phi}(\boldsymbol{z}) - \mathcal{M}_{\mu g}(\boldsymbol{z}). \tag{B.6}$$

The smoothness properties and approximation bounds of $\Psi_\mu$ are summarized below (Hiriart-Urruty, 1991).

**Proposition B.1.** *Consider Problem (B.2) and (B.6). For any $0 < \mu < \frac{1}{m_\phi}$, the following hold.*

(i) *$\Psi_\mu$ is continuously differentiable with gradient $\nabla\Psi_\mu(\boldsymbol{z}) = \mu^{-1}(\text{prox}_{\mu g}(\boldsymbol{z}) - \text{prox}_{\mu\phi}(\boldsymbol{z}))$.*

(ii) *$\nabla\Psi_\mu$ is $\left(\frac{2-\mu m_\phi}{\mu-\mu^2 m_\phi}\right)$-Lipschitz continuous.*

(iii) *For any $\boldsymbol{z} \in \mathbb{R}^n$, it holds that*

$$\Psi(\text{prox}_{\mu\phi}(\boldsymbol{z})) \leq \Psi_\mu(\boldsymbol{z}) \leq \Psi(\text{prox}_{\mu g}(\boldsymbol{z})).$$

### B.4. Solution Correspondence between Original Problem and its DME Surrogate

For completeness, we recall the theoretical relationship between the solutions of Problem (B.2) and its DME surrogate (B.6), as established in Hiriart-Urruty (1991) and Sun & Sun (2023).

Let $\mathcal{X}_\Psi := \arg\min_{\boldsymbol{x}} \Psi(\boldsymbol{x})$ and $\hat{\Psi} := \min_{\boldsymbol{x}} \Psi(\boldsymbol{x})$ denote its solution set and optimal value, respectively. For the DME surrogate, we similarly define its optimal value as $\hat{\Psi}_\mu := \inf_{\boldsymbol{z}} \Psi_\mu(\boldsymbol{z})$ and its solution set as $\mathcal{X}_{\Psi_\mu} := \arg\min_{\boldsymbol{z}} \Psi_\mu(\boldsymbol{z})$.

**Proposition B.2** (Stationary Point and Global Minimizer Correspondence (Hiriart-Urruty, 1991))**.** *Consider Problem (B.2) and its DME surrogate (B.6). Suppose that $\mathcal{X}_\Psi \neq \emptyset$. For any $\mu \in (0, 1/m_\phi)$, the following statements hold:*

*(i) **Stationary point correspondence.** If $\bar{\boldsymbol{z}}$ is a stationary point of $\Psi_\mu$ (i.e., $\nabla\Psi_\mu(\bar{\boldsymbol{z}}) = \boldsymbol{0}$), then $\bar{\boldsymbol{x}} := \mathrm{prox}_{\mu\phi}(\bar{\boldsymbol{z}})$ is a critical point of $\Psi$, satisfying*

$$\mathrm{prox}_{\mu\phi}(\bar{\boldsymbol{z}}) = \mathrm{prox}_{\mu g}(\bar{\boldsymbol{z}}), \quad \Psi(\bar{\boldsymbol{x}}) = \Psi_\mu(\bar{\boldsymbol{z}}). \tag{B.7}$$

*Conversely, if $\bar{\boldsymbol{x}}$ is a critical point of $\Psi$ and $\bar{\boldsymbol{\xi}} \in \partial\phi(\bar{\boldsymbol{x}}) \cap \partial g(\bar{\boldsymbol{x}})$, then $\bar{\boldsymbol{z}} := \bar{\boldsymbol{x}} + \mu\bar{\boldsymbol{\xi}}$ is a stationary point of $\Psi_\mu$ with $\mathrm{prox}_{\mu\phi}(\bar{\boldsymbol{z}}) = \mathrm{prox}_{\mu g}(\bar{\boldsymbol{z}}) = \bar{\boldsymbol{x}}$ and $\Psi(\bar{\boldsymbol{x}}) = \Psi_\mu(\bar{\boldsymbol{z}})$.*

*(ii) **Global minimizer correspondence.** The set $\mathcal{X}_{\Psi_\mu}$ is nonempty and $\hat{\Psi} = \hat{\Psi}_\mu$. Furthermore, if $\bar{\boldsymbol{z}} \in \mathcal{X}_{\Psi_\mu}$, then $\mathrm{prox}_{\mu\phi}(\bar{\boldsymbol{z}}) \in \mathcal{X}_\Psi$. Conversely, if $\bar{\boldsymbol{x}} \in \mathcal{X}_\Psi$ and $\bar{\boldsymbol{\xi}} \in \partial\phi(\bar{\boldsymbol{x}}) \cap \partial g(\bar{\boldsymbol{x}})$, then $\bar{\boldsymbol{z}} := \bar{\boldsymbol{x}} + \mu\bar{\boldsymbol{\xi}} \in \mathcal{X}_{\Psi_\mu}$.*

### B.5. Proof of Proposition 2.1

*Proof.* Let $\bar{\boldsymbol{z}}$ be a point satisfying $\|\nabla\Psi_\mu(\bar{\boldsymbol{z}})\| \leq \min\{1, \mu^{-1}\}\varepsilon$ and set $\bar{\boldsymbol{x}} := \mathrm{prox}_{\mu\phi}(\bar{\boldsymbol{z}})$, $\bar{\boldsymbol{y}} := \mathrm{prox}_{\mu g}(\bar{\boldsymbol{z}})$. Using (2.3), we have $\bar{\boldsymbol{\xi}}_\phi := \mu^{-1}(\bar{\boldsymbol{z}} - \bar{\boldsymbol{x}}) \in \partial\phi(\bar{\boldsymbol{x}})$ and $\bar{\boldsymbol{\xi}}_g := \mu^{-1}(\bar{\boldsymbol{z}} - \bar{\boldsymbol{y}}) \in \partial g(\bar{\boldsymbol{y}})$. Combining these with (2.3) yields $\bar{\boldsymbol{\xi}}_\phi - \bar{\boldsymbol{\xi}}_g = \mu^{-1}(\bar{\boldsymbol{y}} - \bar{\boldsymbol{x}}) = \nabla\Psi_\mu(\bar{\boldsymbol{z}})$. Setting $\bar{\boldsymbol{u}} := \bar{\boldsymbol{\xi}}_\phi - \bar{\boldsymbol{\xi}}_g \in \partial\phi(\bar{\boldsymbol{x}}) - \partial g(\bar{\boldsymbol{y}})$, it follows that

$$\max\{\|\bar{\boldsymbol{u}}\|, \|\bar{\boldsymbol{x}} - \bar{\boldsymbol{y}}\|\} = \max\{\|\nabla\Psi_\mu(\bar{\boldsymbol{z}})\|, \mu\|\nabla\Psi_\mu(\bar{\boldsymbol{z}})\|\} = \max\{1, \mu\}\|\nabla\Psi_\mu(\bar{\boldsymbol{z}})\| \leq \varepsilon,$$

which implies that $\bar{\boldsymbol{x}}$ is an $\varepsilon$-critical point of $\Psi$ in the sense of Definition B.2. $\qquad\square$

### B.6. Discussion on Definition 2.1

There are connections between $\varepsilon$-KKT points and $\varepsilon$-stationary points, but the two notions are not equivalent in general. For a given point $\boldsymbol{x}$, if the constraint violation $\|\boldsymbol{c}(\boldsymbol{x})\|$ is small, then the infeasible-stationarity measure $\|\nabla\boldsymbol{c}(\boldsymbol{x})\boldsymbol{c}(\boldsymbol{x})\|$ is also small. However, the reverse implication does not generally hold. In practical computations, an iterate may get trapped at a stationary point of the constraint-violation minimization problem $\min_{\boldsymbol{x}\in\mathbb{R}^n} \frac{1}{2}\|\boldsymbol{c}(\boldsymbol{x})\|^2$, satisfying $\nabla\boldsymbol{c}(\boldsymbol{x}^*)\boldsymbol{c}(\boldsymbol{x}^*) = 0$ while $\boldsymbol{c}(\boldsymbol{x}^*) \neq 0$, which is referred to as an infeasible stationary point. Therefore, to guarantee that small infeasible stationarity implies small constraint violation and hence approximate feasibility, a constraint qualification condition such as Assumption 2.3 is required. Under such a condition, an $\varepsilon$-stationary point further implies an $\mathcal{O}(\varepsilon)$-KKT point in expectation.

## C. Proofs of Complexity Analysis

### C.1. Proof Sketch

We provide a proof sketch of Theorems 3.1 and 3.2. We outline how the DME-based residual generated by MoSSP-P and MoSSP-R yields the stochastic approximate solutions in Definition 2.1. The proof proceeds in four steps.

STEP 1: FROM THE ALGORITHMIC RESIDUAL TO THE STOCHASTIC KKT INCLUSION.

Let us first set $\boldsymbol{y}^k := \mathrm{prox}_{\mu_k g}(\boldsymbol{z}^k)$. The purpose of this step is to identify the random variables required in Definition 2.1. For both MoSSP-P and MoSSP-R, the optimality condition of the $\boldsymbol{x}$-update implies, as shown in Appendix C.3, that

$$\boldsymbol{u}^{k+1} \in \nabla Q_{\rho_k}(\boldsymbol{x}^{k+1}) + \partial h(\boldsymbol{x}^{k+1}) - \partial g(\boldsymbol{y}^k).$$

Equivalently, by setting $\boldsymbol{\lambda}^{k+1} := \rho_k \boldsymbol{c}(\boldsymbol{x}^{k+1})$, we obtain

$$\boldsymbol{u}^{k+1} \in \nabla f(\boldsymbol{x}^{k+1}) + \nabla \boldsymbol{c}(\boldsymbol{x}^{k+1})\boldsymbol{\lambda}^{k+1} + \partial h(\boldsymbol{x}^{k+1}) - \partial g(\boldsymbol{y}^k).$$

Therefore, for the randomly chosen index $R$, the tuple

$$\bar{\boldsymbol{x}} := \boldsymbol{x}^{R+1}, \qquad \bar{\boldsymbol{y}} := \boldsymbol{y}^R, \qquad \bar{\boldsymbol{\lambda}} := \rho_R \boldsymbol{c}(\boldsymbol{x}^{R+1}), \qquad \bar{\boldsymbol{u}} := \boldsymbol{u}^{R+1},$$

satisfies the KKT inclusion (2.5) almost surely. Hence, it remains to bound

$$\|\boldsymbol{u}^{R+1}\|, \qquad \|\boldsymbol{x}^{R+1} - \boldsymbol{y}^R\|, \qquad \|\nabla \boldsymbol{c}(\boldsymbol{x}^{R+1})\boldsymbol{c}(\boldsymbol{x}^{R+1})\|,$$

and, under Assumption 2.3, the feasibility violation $\|\boldsymbol{c}(\boldsymbol{x}^{R+1})\|$.

STEP 2: ESTABLISHING DESCENT OF THE DC-STRUCTURED POTENTIAL FUNCTION.

The concave term $-g$ prevents the standard penalty objective from admitting a sufficient descent property. We therefore introduce the potential function

$$\mathcal{L}_{\rho,\mu}(\boldsymbol{w}) = Q_\rho(\boldsymbol{x}) + h(\boldsymbol{x}) + \frac{1}{2\mu}\|\boldsymbol{x} - \boldsymbol{z}\|^2 - \mathcal{M}_{\mu g}(\boldsymbol{z}), \qquad \boldsymbol{w} = (\boldsymbol{x}, \boldsymbol{z}),$$

which encodes the two proximal updates of the proposed algorithms. Under the mild parameter condition $\mu_k L_{\rho_k} \leq \frac{1}{4}$, Lemma C.2 yields

$$\mathcal{L}_{\rho_{k+1},\mu_{k+1}}(\boldsymbol{w}^{k+1}) \leq \mathcal{L}_{\rho_k,\mu_k}(\boldsymbol{w}^k) - \Omega(\mu_k^{-1})\|\boldsymbol{w}^{k+1} - \boldsymbol{w}^k\|^2 + \Delta_{k+1} + \frac{\rho_{k+1} - \rho_k}{2}C^2 + \mu_k\|\boldsymbol{e}^k\|^2.$$

Here $\boldsymbol{e}^k$ is the stochastic gradient estimation error, and $\Delta_{k+1}$ accounts for the change of the smoothing parameter $\mu_k$. Thus, after summing over $k$, the iterate variation

$$\sum_{k=0}^{K-1} \mu_k^{-1}\|\boldsymbol{w}^{k+1} - \boldsymbol{w}^k\|^2$$

is controlled by the initial potential difference, the parameter variation terms, and the cumulative stochastic error. In the constant-parameter setting used in the main theorems, the variation terms $\Delta_{k+1}$ and $\rho_{k+1} - \rho_k$ vanish, leaving only the initial potential difference and the stochastic error. This iterate variation bound is then used to control the criticality measure in the next steps.

STEP 3: FROM ITERATE VARIATION TO RESIDUAL BOUNDS.

The one-step residual estimate in Lemma C.3 shows that there exists a constant $\kappa_1 > 0$ independent of $K$ such that

$$\max\{\|\boldsymbol{u}^{k+1}\|^2, \|\boldsymbol{x}^{k+1} - \boldsymbol{y}^k\|^2\} \leq \kappa_1 \left(\|\boldsymbol{e}^k\|^2 + \left(L_{\rho_k}^2 + \mu_k^{-2}\right)\|\boldsymbol{w}^{k+1} - \boldsymbol{w}^k\|^2\right).$$

Thus the descent estimate from Step 2 reduces the averaged DME-induced criticality residual $\max\{\|\boldsymbol{u}^{k+1}\|^2, \|\boldsymbol{x}^{k+1} - \boldsymbol{y}^k\|^2\}$ to the iterate-variation term and the cumulative stochastic error.

The infeasible stationarity measure is then controlled via Lemma C.4, which gives, for some constant $\kappa_2 > 0$ independent of $K$, with $\underline{\rho}_K := \min_{0 \leq k \leq K-1} \rho_k$,

$$\frac{1}{K}\sum_{k=0}^{K-1}\|\nabla \boldsymbol{c}(\boldsymbol{x}^{k+1})\boldsymbol{c}(\boldsymbol{x}^{k+1})\|^2 \leq \frac{\kappa_2}{\underline{\rho}_K^2}\left(\frac{1}{K}\sum_{k=0}^{K-1}\|\boldsymbol{u}^{k+1}\|^2 + 1\right).$$

This estimate makes explicit the role of the penalty parameter: a larger $\rho_k$ improves feasibility control through the factor $\underline{\rho}_K^{-2}$, but it simultaneously increases the smoothness constant $L_{\rho_k}$ in the criticality bound above. Thus, the penalty, smoothing, and momentum parameters must be carefully balanced. Under Assumption 2.3, the infeasible stationarity bound further implies feasibility.

STEP 4: STOCHASTIC ERROR CONTROL AND PARAMETER SELECTION.

The two algorithmic variants differ mainly in how the cumulative stochastic error identified in Step 3 is controlled.

(i) For MoSSP-P, the Polyak error recursion contains an $\alpha_k^{-1}$-weighted variation term ($\alpha_k^{-1}L_f^2\|\boldsymbol{w}^{k+1} - \boldsymbol{w}^k\|^2$ in (C.32)), which necessitates the parameter coupling $\alpha = \Theta(\mu)$. Consequently, the dominant criticality bound takes the form

$$\mathcal{R}_P := \max\left\{\mathbb{E}[\|\boldsymbol{u}^{R+1}\|^2], \mathbb{E}[\|\boldsymbol{x}^{R+1} - \text{prox}_{\mu g}(\boldsymbol{z}^R)\|^2]\right\} \lesssim \frac{1}{\mu K} + \mu.$$

Balancing the two terms gives $\mu = \Theta(K^{-1/2})$, $\alpha = \Theta(K^{-1/2})$, and hence $\mathcal{R}_P = \mathcal{O}(K^{-1/2})$. With $\rho = \Theta(K^{1/4})$, $\|c(x^0)\|^2 = \mathcal{O}(K^{-1/4})$, the feasibility residual is also of order $\mathcal{O}(K^{-1/2})$ under Assumption 2.3. Thus $K = \mathcal{O}(\varepsilon^{-4})$ suffices, yielding the $\mathcal{O}(\varepsilon^{-4})$ oracle complexity of MoSSP-P.

(ii) For MoSSP-R, the recursive momentum estimator does not introduce the $\alpha_k^{-1}$-weighted variation term (see (C.44)). This allows for the weaker coupling $\alpha = \Theta(\mu^2)$, and the dominant criticality bound becomes

$$\mathcal{R}_R := \max\left\{\mathbb{E}[\|u^{R+1}\|^2], \mathbb{E}[\|x^{R+1} - \text{prox}_{\mu g}(z^R)\|^2]\right\} \lesssim \frac{1}{\mu K} + \mu^2.$$

Balancing the two terms gives $\mu = \Theta(K^{-1/3})$, $\alpha = \Theta(K^{-2/3})$. Together with the initial batch size $b_0 = \Theta(K^{1/3})$, which controls the initial estimation error, we obtain $\mathcal{R}_R = \mathcal{O}(K^{-2/3})$. With $\rho = \Theta(K^{1/3})$ and $\|c(x^0)\|^2 = \mathcal{O}(K^{-1/3})$, the feasibility residual is also of order $\mathcal{O}(K^{-2/3})$ under Assumption 2.3. Hence the setting $K = \mathcal{O}(\varepsilon^{-3})$ ensures an $\varepsilon$-approximate solution. Since $b_0 = \mathcal{O}(K^{1/3})$ is dominated by the total number of per-iteration oracle calls, MoSSP-R achieves $\mathcal{O}(\varepsilon^{-3})$ oracle complexity.

## C.2. Constructed Potential Function

We define the potential function

$$\mathcal{L}_{\rho,\mu}(w) := Q_\rho(x) + h(x) + \frac{1}{2\mu}\|x - z\|^2 - \mathcal{M}_{\mu g}(z), \tag{C.1}$$

where $w = (x, z)$. Recall that $F^*$ is the finite infimum introduced in Problem (P). Since $Q_\rho(x) = f(x) + \frac{\rho}{2}\|c(x)\|^2 \geq f(x)$, we obtain the uniform lower bound:

$$\inf_{x \in \mathbb{R}^n} \{Q_\rho(x) + h(x) - g(x)\} \geq \inf_{x \in \mathbb{R}^n} F(x) = F^* > -\infty, \quad \forall \rho > 0. \tag{C.2}$$

Unlike the convex case (where $g = 0$), the concave component $-g$ prevents the potential sequence $\{\mathcal{L}_{\rho_k,\mu_k}(w^k)\}_{k\in\mathbb{N}}$ from being trivially lower-bounded. However, since $g$ is convex and $\sup_{v_g \in \partial g(x)} \|v_g\| \leq G$ for all $x$ (Assumption 2.1), $g$ is $G$-Lipschitz continuous, and we can bound the potential function for any $\mu > 0$ and $\rho > 0$ as follows:

$$\begin{aligned}
\mathcal{L}_{\rho,\mu}(w) &\geq Q_\rho(x) + h(x) + \frac{1}{2\mu}\|x - z\|^2 - g(z) \\
&\geq Q_\rho(x) + h(x) - g(x) + \frac{1}{2\mu}\|x - z\|^2 - G\|x - z\| \\
&\geq Q_\rho(x) + h(x) - g(x) - \frac{G^2\mu}{2} \\
&\geq F^* - \frac{G^2\mu}{2},
\end{aligned} \tag{C.3}$$

where the first inequality follows from $\mathcal{M}_{\mu g}(z) \leq g(z)$ in (B.1), the second utilizes the Lipschitz continuity of $g$, and the third applies Young's inequality. Under (C.7), the sequence $\{\mu_k\}$ is non-increasing, hence there exists a constant $\bar{\mu}$ independent of $K$ satisfying $\mu_k \leq \bar{\mu}$ for all $k \in \mathbb{N}$. Then (C.3) yields the uniform lower bound

$$\mathcal{L}_{\rho_k,\mu_k}(w^k) \geq F^* - \frac{G^2\bar{\mu}}{2} =: \mathcal{L}^*, \quad \forall k \in \mathbb{N}, \tag{C.4}$$

which we use throughout the proofs.

## C.3. Common Auxiliary Lemmas

For notational convenience, define the stochastic gradient error

$$e^k := \begin{cases} S^k - \nabla Q_{\rho_k}(x^k), & \text{for MoSSP-P,} \\ D^k - \nabla Q_{\rho_k}(x^k), & \text{for MoSSP-R.} \end{cases} \tag{C.5}$$

Also set
$$\boldsymbol{y}^k := \text{prox}_{\mu_k g}(\boldsymbol{z}^k).$$

We first verify that the residual $\boldsymbol{u}^{k+1}$ defined in (3.9) or (3.17) satisfies the residual inclusion
$$\boldsymbol{u}^{k+1} \in \partial\psi_{\rho_k}(\boldsymbol{x}^{k+1}) - \partial g(\text{prox}_{\mu_k g}(\boldsymbol{z}^k)). \tag{C.6}$$

Indeed, both (3.8) and (3.16) can be written in the unified form
$$\boldsymbol{x}^{k+1} = \text{prox}_{\mu_k h}\big(\boldsymbol{z}^k - \mu_k \tilde{\nabla} Q_{\rho_k}(\boldsymbol{x}^k)\big).$$

The optimality condition of this proximal update gives
$$\boldsymbol{0} \in \tilde{\nabla} Q_{\rho_k}(\boldsymbol{x}^k) + \partial h(\boldsymbol{x}^{k+1}) + \mu_k^{-1}(\boldsymbol{x}^{k+1} - \boldsymbol{z}^k).$$

Hence, there exists
$$\boldsymbol{v}_h^{k+1} := \mu_k^{-1}(\boldsymbol{z}^k - \boldsymbol{x}^{k+1}) - \tilde{\nabla} Q_{\rho_k}(\boldsymbol{x}^k) \in \partial h(\boldsymbol{x}^{k+1}).$$

Moreover, by the definition of $\boldsymbol{y}^k = \text{prox}_{\mu_k g}(\boldsymbol{z}^k)$, we have
$$\boldsymbol{v}_g^k := \mu_k^{-1}(\boldsymbol{z}^k - \boldsymbol{y}^k) \in \partial g(\boldsymbol{y}^k).$$

Using the unified definition
$$\boldsymbol{u}^{k+1} := \nabla Q_{\rho_k}(\boldsymbol{x}^{k+1}) - \tilde{\nabla} Q_{\rho_k}(\boldsymbol{x}^k) + \mu_k^{-1}(\boldsymbol{y}^k - \boldsymbol{x}^{k+1}),$$

which coincides with (3.9) for MoSSP-P and with (3.17) for MoSSP-R, we obtain
$$\boldsymbol{u}^{k+1} = \nabla Q_{\rho_k}(\boldsymbol{x}^{k+1}) + \boldsymbol{v}_h^{k+1} - \boldsymbol{v}_g^k \in \nabla Q_{\rho_k}(\boldsymbol{x}^{k+1}) + \partial h(\boldsymbol{x}^{k+1}) - \partial g(\boldsymbol{y}^k).$$

Since $Q_{\rho_k}$ is smooth and $\psi_{\rho_k} = Q_{\rho_k} + h$, we have
$$\nabla Q_{\rho_k}(\boldsymbol{x}^{k+1}) + \partial h(\boldsymbol{x}^{k+1}) = \partial\psi_{\rho_k}(\boldsymbol{x}^{k+1}).$$

This proves (C.6). Equivalently, because
$$\nabla Q_{\rho_k}(\boldsymbol{x}^{k+1}) = \nabla f(\boldsymbol{x}^{k+1}) + \rho_k \nabla \boldsymbol{c}(\boldsymbol{x}^{k+1})\boldsymbol{c}(\boldsymbol{x}^{k+1}),$$

the choice
$$\boldsymbol{\lambda}^{k+1} := \rho_k \boldsymbol{c}(\boldsymbol{x}^{k+1})$$

yields
$$\boldsymbol{u}^{k+1} \in \nabla f(\boldsymbol{x}^{k+1}) + \nabla \boldsymbol{c}(\boldsymbol{x}^{k+1})\boldsymbol{\lambda}^{k+1} + \partial h(\boldsymbol{x}^{k+1}) - \partial g(\boldsymbol{y}^k).$$

Thus the residual $\boldsymbol{u}^{k+1}$, together with $\boldsymbol{y}^k$ and $\boldsymbol{\lambda}^{k+1}$, provides the KKT-type inclusion used throughout the proof. $\qquad\square$

Under Assumption 2.1, we can guarantee the smoothness of $Q_\rho(\boldsymbol{x})$, which is essential for our analysis.

**Lemma C.1.** *Suppose that Assumption 2.1 holds. Then, for any $k \geq 1$ and $\rho \geq \rho_0$, the function $Q_\rho(\boldsymbol{x})$ is $L_\rho$-smooth on $\mathbb{R}^n$, where $L_\rho = \rho\tilde{L}$ with $\tilde{L} = \rho_0^{-1}L_f + G^2 + CL_c$.*

The following lemma shows that the potential function value sequence $\{\mathcal{L}_{\rho_k,\mu_k}(\boldsymbol{w}^k)\}_{k\in\mathbb{N}}$ in (C.1) is non-increasing up to a stochastic gradient error term. To ensure this descent, we assume the parameters $\mu_k$, $\rho_k$ and $\beta$ satisfy
$$\mu_k L_{\rho_k} \leq \frac{1}{4}, \quad \mu_{k+1} \leq \mu_k, \quad \rho_k \leq \rho_{k+1}, \quad 0 < \beta \leq 1, \quad \forall k \geq 0. \tag{C.7}$$

**Lemma C.2.** *Suppose that Assumption 2.1 holds, and the parameters $\mu_k$, $\rho_k$ and $\beta$ satisfy (C.7). Then, for any $k \geq 0$, it holds that*
$$\mathcal{L}_{\rho_{k+1},\mu_{k+1}}(\boldsymbol{w}^{k+1}) \leq \mathcal{L}_{\rho_k,\mu_k}(\boldsymbol{w}^k) - \frac{(2\mu_k)^{-1} - L_{\rho_k}}{2}\|\boldsymbol{w}^{k+1} - \boldsymbol{w}^k\|^2 + \frac{\rho_{k+1} - \rho_k}{2}C^2 + \Delta_{k+1} + \mu_k\|\boldsymbol{e}^k\|^2, \tag{C.8}$$

*where $\Delta_{k+1} := \frac{|\mu_k - \mu_{k+1}|}{2\mu_{k+1}^2}(C^2 + \|\boldsymbol{x}^{k+1} - \boldsymbol{z}^{k+1}\|^2)$. Moreover, it holds that*
$$\|\boldsymbol{w}^{k+1} - \boldsymbol{w}^k\|^2 \leq \frac{4\mu_k}{1 - 2\mu_k L_{\rho_k}}\Big(\mathcal{L}_{\rho_k,\mu_k}(\boldsymbol{w}^k) - \mathcal{L}_{\rho_{k+1},\mu_{k+1}}(\boldsymbol{w}^{k+1}) + \Delta_{k+1} + \frac{\rho_{k+1} - \rho_k}{2}C^2 + \mu_k\|\boldsymbol{e}^k\|^2\Big). \tag{C.9}$$

*Proof.* Since $\mu_k L_{\rho_k} \leq \frac{1}{4}$, it follows that $\frac{(2\mu_k)^{-1} - L_{\rho_k}}{2} = \frac{1}{4\mu_k} - \frac{L_{\rho_k}}{2} = \frac{1 - 2\mu_k L_{\rho_k}}{4\mu_k} \geq \frac{1}{8\mu_k} > 0$. To prove (C.8), we bound $\mathcal{L}_{\rho_{k+1}, \mu_{k+1}}(\boldsymbol{w}^{k+1}) - \mathcal{L}_{\rho_k, \mu_k}(\boldsymbol{w}^k)$ by decomposing it as

$$\mathcal{L}_{\rho_{k+1}, \mu_{k+1}}(\boldsymbol{w}^{k+1}) - \mathcal{L}_{\rho_k, \mu_k}(\boldsymbol{w}^k) = [\mathcal{L}_{\rho_{k+1}, \mu_{k+1}}(\boldsymbol{w}^{k+1}) - \mathcal{L}_{\rho_k, \mu_k}(\boldsymbol{w}^{k+1})] + [\mathcal{L}_{\rho_k, \mu_k}(\boldsymbol{w}^{k+1}) - \mathcal{L}_{\rho_k, \mu_k}(\boldsymbol{w}^k)]. \quad \text{(C.10)}$$

For the first term in (C.10), we have

$$\mathcal{L}_{\rho_{k+1}, \mu_{k+1}}(\boldsymbol{w}^{k+1}) - \mathcal{L}_{\rho_k, \mu_k}(\boldsymbol{w}^{k+1}) \leq \frac{\rho_{k+1} - \rho_k}{2} C^2 + \Delta_{k+1}, \quad \text{(C.11)}$$

where we use $\|\boldsymbol{c}(\boldsymbol{x}^{k+1})\| \leq C$ from Assumption 2.1.

For the second term of (C.10), we decompose it as

$$\begin{aligned}
&\mathcal{L}_{\rho_k, \mu_k}(\boldsymbol{w}^{k+1}) - \mathcal{L}_{\rho_k, \mu_k}(\boldsymbol{w}^k) \\
&= [\mathcal{L}_{\rho_k, \mu_k}(\boldsymbol{x}^{k+1}, \boldsymbol{z}^{k+1}) - \mathcal{L}_{\rho_k, \mu_k}(\boldsymbol{x}^{k+1}, \boldsymbol{z}^k)] + [\mathcal{L}_{\rho_k, \mu_k}(\boldsymbol{x}^{k+1}, \boldsymbol{z}^k) - \mathcal{L}_{\rho_k, \mu_k}(\boldsymbol{x}^k, \boldsymbol{z}^k)].
\end{aligned} \quad \text{(C.12)}$$

The first part of (C.12) is

$$\begin{aligned}
&\mathcal{L}_{\rho_k, \mu_k}(\boldsymbol{x}^{k+1}, \boldsymbol{z}^{k+1}) - \mathcal{L}_{\rho_k, \mu_k}(\boldsymbol{x}^{k+1}, \boldsymbol{z}^k) \\
&= \frac{1}{2\mu_k}(\|\boldsymbol{x}^{k+1} - \boldsymbol{z}^{k+1}\|^2 - \|\boldsymbol{x}^{k+1} - \boldsymbol{z}^k\|^2) + \mathcal{M}_{\mu_k g}(\boldsymbol{z}^k) - \mathcal{M}_{\mu_k g}(\boldsymbol{z}^{k+1}) \\
&\leq \frac{1}{2\mu_k}(\|\boldsymbol{x}^{k+1} - \boldsymbol{z}^{k+1}\|^2 - \|\boldsymbol{x}^{k+1} - \boldsymbol{z}^k\|^2) - \frac{1}{\mu_k}\langle \boldsymbol{z}^k - \text{prox}_{\mu_k g}(\boldsymbol{z}^k), \boldsymbol{z}^{k+1} - \boldsymbol{z}^k \rangle.
\end{aligned} \quad \text{(C.13)}$$

Using the identity $\|\boldsymbol{a}\|^2 - \|\boldsymbol{b}\|^2 = 2\langle \boldsymbol{a} - \boldsymbol{b}, \boldsymbol{a} \rangle - \|\boldsymbol{a} - \boldsymbol{b}\|^2$, with $\boldsymbol{a} = \boldsymbol{x}^{k+1} - \boldsymbol{z}^{k+1}$ and $\boldsymbol{b} = \boldsymbol{x}^{k+1} - \boldsymbol{z}^k$, we obtain

$$\|\boldsymbol{x}^{k+1} - \boldsymbol{z}^{k+1}\|^2 - \|\boldsymbol{x}^{k+1} - \boldsymbol{z}^k\|^2 = 2\langle \boldsymbol{z}^k - \boldsymbol{z}^{k+1}, \boldsymbol{x}^{k+1} - \boldsymbol{z}^{k+1} \rangle - \|\boldsymbol{z}^{k+1} - \boldsymbol{z}^k\|^2.$$

Substituting this into (C.13) and combining the optimality condition of the subproblem (3.5),

$$\text{prox}_{\mu_k g}(\boldsymbol{z}^k) - \boldsymbol{x}^{k+1} = \beta^{-1}(\boldsymbol{z}^k - \boldsymbol{z}^{k+1}), \quad \text{(C.14)}$$

we obtain

$$\mathcal{L}_{\rho_k, \mu_k}(\boldsymbol{x}^{k+1}, \boldsymbol{z}^{k+1}) - \mathcal{L}_{\rho_k, \mu_k}(\boldsymbol{x}^{k+1}, \boldsymbol{z}^k) \leq -\frac{1}{\mu_k}(\beta^{-1} - \frac{1}{2})\|\boldsymbol{z}^{k+1} - \boldsymbol{z}^k\|^2. \quad \text{(C.15)}$$

For the second part of (C.12), by the optimality condition of the subproblem (3.8) or (3.16), together with its strong convexity, one has

$$h(\boldsymbol{x}^{k+1}) + \langle \boldsymbol{e}^k + \nabla Q_{\rho_k}(\boldsymbol{x}^k), \boldsymbol{x}^{k+1} - \boldsymbol{x}^k \rangle + \frac{1}{2\mu_k}\|\boldsymbol{x}^{k+1} - \boldsymbol{z}^k\|^2 \leq h(\boldsymbol{x}^k) + \frac{1}{2\mu_k}\|\boldsymbol{x}^k - \boldsymbol{z}^k\|^2 - \frac{1}{2\mu_k}\|\boldsymbol{x}^{k+1} - \boldsymbol{x}^k\|^2.$$

Using the $L_{\rho_k}$-smoothness of $Q_{\rho_k}$ and the fact $\langle \boldsymbol{a}, \boldsymbol{b} \rangle \leq \frac{1}{4\mu_k}\|\boldsymbol{a}\|^2 + \mu_k\|\boldsymbol{b}\|^2$, one has

$$\mathcal{L}_{\rho_k, \mu_k}(\boldsymbol{x}^{k+1}, \boldsymbol{z}^k) - \mathcal{L}_{\rho_k, \mu_k}(\boldsymbol{x}^k, \boldsymbol{z}^k) \leq -\frac{(2\mu_k)^{-1} - L_{\rho_k}}{2}\|\boldsymbol{x}^{k+1} - \boldsymbol{x}^k\|^2 + \mu_k\|\boldsymbol{e}^k\|^2. \quad \text{(C.16)}$$

Then, combining (C.15) and (C.16) in (C.12), we obtain

$$\mathcal{L}_{\rho_k, \mu_k}(\boldsymbol{w}^{k+1}) - \mathcal{L}_{\rho_k, \mu_k}(\boldsymbol{w}^k) \leq -\min\{\frac{(2\mu_k)^{-1} - L_{\rho_k}}{2}, \frac{1}{\mu_k}(\beta^{-1} - \frac{1}{2})\}\|\boldsymbol{w}^{k+1} - \boldsymbol{w}^k\|^2 + \mu_k\|\boldsymbol{e}^k\|^2. \quad \text{(C.17)}$$

Since (C.7) ensures $\frac{(2\mu_k)^{-1} - L_{\rho_k}}{2} \leq \frac{1}{\mu_k}(\beta^{-1} - \frac{1}{2})$, one has $\min\{\frac{(2\mu_k)^{-1} - L_{\rho_k}}{2}, \frac{1}{\mu_k}(\beta^{-1} - \frac{1}{2})\} = \frac{(2\mu_k)^{-1} - L_{\rho_k}}{2}$. Then, substituting (C.11) and (C.17) into (C.10) yields (C.8). Since the coefficient of $\|\boldsymbol{w}^{k+1} - \boldsymbol{w}^k\|^2$ in (C.8) is positive, rearranging this inequality gives (C.9), completing the proof. $\square$

We next derive the one-step bound on $\boldsymbol{u}$ and the associated criticality measure during iteration.

**Lemma C.3.** *Suppose that Assumption 2.1 holds. Then, for any $k \geq 0$, with $\boldsymbol{u}^{k+1}$ defined by (3.9) or (3.17), it holds that*

$$\|\boldsymbol{u}^{k+1}\|^2 \leq 3\|\boldsymbol{e}^k\|^2 + 3\big(L_{\rho_k}^2 + (\mu_k\beta)^{-2}\big)\|\boldsymbol{w}^{k+1} - \boldsymbol{w}^k\|^2, \tag{C.18}$$

*and*

$$\max\{\|\boldsymbol{u}^{k+1}\|^2, \|\boldsymbol{x}^{k+1} - \mathrm{prox}_{\mu_k g}(\boldsymbol{z}^k)\|^2\} \leq 3\|\boldsymbol{e}^k\|^2 + \big(3\big(L_{\rho_k}^2 + (\mu_k\beta)^{-2}\big) + \beta^{-2}\big)\|\boldsymbol{w}^{k+1} - \boldsymbol{w}^k\|^2. \tag{C.19}$$

*Proof.* Recalling (C.14) and substituting it into (3.9) or (3.17), one obtains

$$\boldsymbol{u}^{k+1} = -\boldsymbol{e}^k + \nabla Q_{\rho_k}(\boldsymbol{x}^{k+1}) - \nabla Q_{\rho_k}(\boldsymbol{x}^k) + \frac{1}{\mu_k\beta}(\boldsymbol{z}^k - \boldsymbol{z}^{k+1}).$$

Consequently, it holds that

$$\begin{aligned}
\|\boldsymbol{u}^{k+1}\|^2 &\leq 3\|\boldsymbol{e}^k\|^2 + 3\|\nabla Q_{\rho_k}(\boldsymbol{x}^{k+1}) - \nabla Q_{\rho_k}(\boldsymbol{x}^k)\|^2 + 3(\mu_k\beta)^{-2}\|\boldsymbol{z}^{k+1} - \boldsymbol{z}^k\|^2 \\
&\leq 3\|\boldsymbol{e}^k\|^2 + 3L_{\rho_k}^2\|\boldsymbol{x}^{k+1} - \boldsymbol{x}^k\|^2 + 3(\mu_k\beta)^{-2}\|\boldsymbol{z}^{k+1} - \boldsymbol{z}^k\|^2 \\
&\leq 3\|\boldsymbol{e}^k\|^2 + 3\big(L_{\rho_k}^2 + (\mu_k\beta)^{-2}\big)\|\boldsymbol{w}^{k+1} - \boldsymbol{w}^k\|^2,
\end{aligned} \tag{C.20}$$

where the first inequality follows from $\|\boldsymbol{a} + \boldsymbol{b} + \boldsymbol{c}\|^2 \leq 3(\|\boldsymbol{a}\|^2 + \|\boldsymbol{b}\|^2 + \|\boldsymbol{c}\|^2)$ and the second inequality uses the $L_{\rho_k}$-smoothness of $Q_{\rho_k}$, establishing (C.18).

For the left-hand side of (C.19), one has, by (C.14),

$$\max\{\|\boldsymbol{u}^{k+1}\|^2, \|\boldsymbol{x}^{k+1} - \mathrm{prox}_{\mu_k g}(\boldsymbol{z}^k)\|^2\} \leq \|\boldsymbol{u}^{k+1}\|^2 + \beta^{-2}\|\boldsymbol{z}^{k+1} - \boldsymbol{z}^k\|^2.$$

Combining this with (C.20) gives (C.19). $\qquad\square$

In general, finding a feasible solution for nonconvex constrained optimization problems is challenging. It is thus necessary to characterize infeasible stationarity measures $\|\nabla\boldsymbol{c}(\boldsymbol{x})\boldsymbol{c}(\boldsymbol{x})\|^2$. Under a constraint qualification such as Assumption 2.3, we can further bound constraint violations $\|\boldsymbol{c}(\boldsymbol{x})\|^2$. The following lemma establishes key bounds relating $\boldsymbol{u}$ to these two measures.

**Lemma C.4.** *Suppose that the conditions of Lemma C.2 hold. Let $\underline{\rho}_K := \min_{0 \leq k \leq K-1} \rho_k$. Then, for any $K \geq 1$, it holds that*

$$\frac{1}{K}\sum_{k=0}^{K-1}\|\nabla\boldsymbol{c}(\boldsymbol{x}^{k+1})\boldsymbol{c}(\boldsymbol{x}^{k+1})\|^2 \leq \frac{2}{\underline{\rho}_K^2 K}\sum_{k=0}^{K-1}\|\boldsymbol{u}^{k+1}\|^2 + \frac{18G^2}{\underline{\rho}_K^2}. \tag{C.21}$$

*Moreover, under Assumption 2.3, it holds that*

$$\frac{1}{K}\sum_{k=0}^{K-1}\|\boldsymbol{c}(\boldsymbol{x}^{k+1})\|^2 \leq \frac{2}{\underline{\rho}_K^2\delta^2 K}\sum_{k=0}^{K-1}\|\boldsymbol{u}^{k+1}\|^2 + \frac{18G^2}{\delta^2\underline{\rho}_K^2}. \tag{C.22}$$

*Proof.* By (C.6), there exist $\boldsymbol{v}_h^{k+1} \in \partial h(\boldsymbol{x}^{k+1})$ and $\nabla\mathcal{M}_{\mu_k g}(\boldsymbol{z}^k) \in \partial g(\mathrm{prox}_{\mu_k g}(\boldsymbol{z}^k))$ such that $\boldsymbol{u}^{k+1} = \nabla Q_{\rho_k}(\boldsymbol{x}^{k+1}) + \boldsymbol{v}_h^{k+1} - \nabla\mathcal{M}_{\mu_k g}(\boldsymbol{z}^k)$. This yields

$$\begin{aligned}
&\frac{1}{\rho_k}\|\rho_k\nabla\boldsymbol{c}(\boldsymbol{x}^{k+1})\boldsymbol{c}(\boldsymbol{x}^{k+1})\| \\
&= \frac{1}{\rho_k}\|\nabla f(\boldsymbol{x}^{k+1}) + \rho_k\nabla\boldsymbol{c}(\boldsymbol{x}^{k+1})\boldsymbol{c}(\boldsymbol{x}^{k+1}) - \nabla f(\boldsymbol{x}^{k+1})\| \\
&= \frac{1}{\rho_k}\|\nabla Q_{\rho_k}(\boldsymbol{x}^{k+1}) + \boldsymbol{v}_h^{k+1} - \nabla\mathcal{M}_{\mu_k g}(\boldsymbol{z}^k) - \nabla f(\boldsymbol{x}^{k+1}) - \boldsymbol{v}_h^{k+1} + \nabla\mathcal{M}_{\mu_k g}(\boldsymbol{z}^k)\| \\
&\leq \frac{1}{\rho_k}\big(\|\boldsymbol{u}^{k+1}\| + \|\nabla f(\boldsymbol{x}^{k+1})\| + \|\boldsymbol{v}_h^{k+1}\| + \|\nabla\mathcal{M}_{\mu_k g}(\boldsymbol{z}^k)\|\big) \\
&\leq \frac{\|\boldsymbol{u}^{k+1}\|}{\rho_k} + \frac{3G}{\rho_k},
\end{aligned} \tag{C.23}$$

where the last inequality uses $\|\nabla f(\boldsymbol{x}^{k+1})\|, \|\boldsymbol{v}_h^{k+1}\|, \|\nabla \mathcal{M}_{\mu_k g}(\boldsymbol{z}^k)\| \leq G$ from Assumption 2.1. Squaring both sides of (C.23) gives

$$\|\nabla \boldsymbol{c}(\boldsymbol{x}^{k+1}) \boldsymbol{c}(\boldsymbol{x}^{k+1})\|^2 \leq \frac{2\|\boldsymbol{u}^{k+1}\|^2}{\rho_k^2} + \frac{18G^2}{\rho_k^2}. \tag{C.24}$$

Summing (C.24) over $k = 0, \ldots, K-1$ and dividing by $K$, we obtain

$$\frac{1}{K} \sum_{k=0}^{K-1} \|\nabla \boldsymbol{c}(\boldsymbol{x}^{k+1}) \boldsymbol{c}(\boldsymbol{x}^{k+1})\|^2 \leq \frac{2}{K} \sum_{k=0}^{K-1} \frac{\|\boldsymbol{u}^{k+1}\|^2}{\rho_k^2} + \frac{18G^2}{K} \sum_{k=0}^{K-1} \frac{1}{\rho_k^2},$$

where using $\rho_k \geq \underline{\rho}_K$ for $0 \leq k \leq K-1$ establishes (C.21).

Under Assumption 2.3, it is straightforward to obtain that

$$\frac{1}{K} \sum_{k=0}^{K-1} \|\boldsymbol{c}(\boldsymbol{x}^{k+1})\|^2 \leq \frac{1}{\delta^2 K} \sum_{k=0}^{K-1} \|\nabla \boldsymbol{c}(\boldsymbol{x}^{k+1}) \boldsymbol{c}(\boldsymbol{x}^{k+1})\|^2,$$

where substituting (C.21) into the above inequality yields (C.22), completing the proof. $\qquad\square$

The following lemma establishes an expected averaged bound on $\mathbb{E}[\|\boldsymbol{u}^{k+1}\|^2]$.

**Lemma C.5.** *Suppose that the conditions of Lemma C.2 hold. Then, for $K \geq 1$, it holds that*

$$\frac{1}{K} \sum_{k=0}^{K-1} \mathbb{E}[\|\boldsymbol{u}^{k+1}\|^2] \leq \frac{24(\frac{1}{16} + \beta^{-2})}{\mu_K K} (\mathcal{L}_{\rho_0,\mu_0}(\boldsymbol{w}^0) - \mathcal{L}^*)$$

$$+ \frac{3 + 24(\frac{1}{16} + \beta^{-2})}{K} \sum_{k=0}^{K-1} \mathbb{E}[\|\boldsymbol{e}^k\|^2]$$

$$+ \frac{24(\frac{1}{16} + \beta^{-2})}{\mu_K K} \sum_{k=0}^{K-1} \left( \frac{\rho_{k+1} - \rho_k}{2} C^2 + \mathbb{E}[\Delta_{k+1}] \right), \tag{C.25}$$

*where $\mathcal{L}^*$ is as defined in (C.4).*

*Proof.* Taking expectations in (C.18) and averaging over $k = 0, \ldots, K-1$, one obtains

$$\frac{1}{K} \sum_{k=0}^{K-1} \mathbb{E}[\|\boldsymbol{u}^{k+1}\|^2] \leq \frac{3}{K} \sum_{k=0}^{K-1} (L_{\rho_k}^2 + (\mu_k \beta)^{-2}) \mathbb{E}[\|\boldsymbol{w}^{k+1} - \boldsymbol{w}^k\|^2] + \frac{3}{K} \sum_{k=0}^{K-1} \mathbb{E}[\|\boldsymbol{e}^k\|^2]. \tag{C.26}$$

Next, substituting (C.9) into the first term of (C.26) gives

$$\frac{3}{K} \sum_{k=0}^{K-1} (L_{\rho_k}^2 + (\mu_k \beta)^{-2}) \mathbb{E}[\|\boldsymbol{w}^{k+1} - \boldsymbol{w}^k\|^2]$$

$$\leq \frac{12}{K} \sum_{k=0}^{K-1} \frac{\mu_k^2 L_{\rho_k}^2 + \beta^{-2}}{\mu_k(1 - 2\mu_k L_{\rho_k})} \left( \mathbb{E}[\mathcal{L}_{\rho_k,\mu_k}(\boldsymbol{w}^k) - \mathcal{L}_{\rho_{k+1},\mu_{k+1}}(\boldsymbol{w}^{k+1})] \right.$$

$$\left. + \frac{\rho_{k+1} - \rho_k}{2} C^2 + \mathbb{E}[\Delta_{k+1}] + \mu_k \mathbb{E}[\|\boldsymbol{e}^k\|^2] \right). \tag{C.27}$$

Given $\mu_k \geq \mu_K$ and $\mu_k L_{\rho_k} \leq \frac{1}{4}$ from (C.7), it holds that

$$\frac{\mu_k^2 L_{\rho_k}^2 + \beta^{-2}}{\mu_k(1 - 2\mu_k L_{\rho_k})} \leq \frac{2}{\mu_K}\left(\frac{1}{16} + \beta^{-2}\right), \tag{C.28}$$

where we use the inequality $\frac{v^2 + \beta^{-2}}{1 - 2v} \le 2(\frac{1}{16} + \beta^{-2})$ with $v = \mu_k L_{\rho_k}$. Then, substituting this into (C.27) yields

$$
\frac{3}{K} \sum_{k=0}^{K-1} (L_{\rho_k}^2 + (\beta\mu_k)^{-2}) \mathbb{E}[\|\boldsymbol{w}^{k+1} - \boldsymbol{w}^k\|^2]
$$

$$
\le \frac{24(\frac{1}{16} + \beta^{-2})}{\mu_K K} (\mathcal{L}_{\rho_0, \mu_0}(\boldsymbol{w}^0) - \mathcal{L}^*) + \frac{24(\frac{1}{16} + \beta^{-2})}{\mu_K K} \sum_{k=0}^{K-1} \left( \frac{\rho_{k+1} - \rho_k}{2} C^2 + \mathbb{E}[\Delta_{k+1}] \right)
$$

$$
+ \frac{24(\frac{1}{16} + \beta^{-2})}{K} \sum_{k=0}^{K-1} \mathbb{E}[\|\boldsymbol{e}^k\|^2].
$$

Combining this with the second term of (C.26), we obtain the desired inequality, completing the proof. $\qquad\square$

We now proceed to estimate the quantities in (2.6) and (2.7) for the output point $\boldsymbol{x}^{R+1}$. The result follows directly from Lemma C.4 and Lemma C.5, and the proof is omitted for brevity.

**Lemma C.6.** *Suppose that the conditions of Lemma C.2 hold. Let $\underline{\rho}_K := \min_{0 \le k \le K-1} \rho_k$. Then, it holds that*

$$
\max\{\mathbb{E}[\|\boldsymbol{u}^{R+1}\|^2], \mathbb{E}[\|\boldsymbol{x}^{R+1} - \mathrm{prox}_{\mu_R g}(\boldsymbol{z}^R)\|^2]\}
$$

$$
\le \frac{24(\frac{1}{16} + \beta^{-2} + \frac{1}{3}(\mu_0/\beta)^2)}{\mu_K K} (\mathcal{L}_{\rho_0, \mu_0}(\boldsymbol{w}^0) - \mathcal{L}^*) + \frac{24(\frac{1}{16} + \beta^{-2} + \frac{1}{3}(\mu_0/\beta)^2)}{\mu_K K} \sum_{k=0}^{K-1} \left( \frac{\rho_{k+1} - \rho_k}{2} C^2 + \mathbb{E}[\Delta_{k+1}] \right)
$$

$$
+ \frac{24(\frac{1}{16} + \beta^{-2} + \frac{1}{3}(\mu_0/\beta)^2) + 3}{K} \sum_{k=0}^{K-1} \mathbb{E}[\|\boldsymbol{e}^k\|^2], \tag{C.29}
$$

*and*

$$
\mathbb{E}[\|\nabla\boldsymbol{c}(\boldsymbol{x}^{R+1})\boldsymbol{c}(\boldsymbol{x}^{R+1})\|^2]
$$

$$
\le \frac{48(\frac{1}{16} + \beta^{-2})}{\underline{\rho}_K^2 \mu_K K} \left( \mathcal{L}_{\rho_0, \mu_0}(\boldsymbol{w}^0) - \mathcal{L}^* + \sum_{k=0}^{K-1} \left( \frac{\rho_{k+1} - \rho_k}{2} C^2 + \mathbb{E}[\Delta_{k+1}] \right) \right)
$$

$$
+ \frac{18G^2}{\underline{\rho}_K^2} + \frac{6 + 48(\frac{1}{16} + \beta^{-2})}{\underline{\rho}_K^2 K} \sum_{k=0}^{K-1} \mathbb{E}[\|\boldsymbol{e}^k\|^2]. \tag{C.30}
$$

*Moreover, under Assumption 2.3, it holds that*

$$
\mathbb{E}[\|\boldsymbol{c}(\boldsymbol{x}^{R+1})\|^2]
$$

$$
\le \frac{48(\frac{1}{16} + \beta^{-2})}{\delta^2 \underline{\rho}_K^2 \mu_K K} \left( \mathcal{L}_{\rho_0, \mu_0}(\boldsymbol{w}^0) - \mathcal{L}^* + \sum_{k=0}^{K-1} \left( \frac{\rho_{k+1} - \rho_k}{2} C^2 + \mathbb{E}[\Delta_{k+1}] \right) \right)
$$

$$
+ \frac{18G^2}{\delta^2 \underline{\rho}_K^2} + \frac{6 + 48(\frac{1}{16} + \beta^{-2})}{\delta^2 \underline{\rho}_K^2 K} \sum_{k=0}^{K-1} \mathbb{E}[\|\boldsymbol{e}^k\|^2]. \tag{C.31}
$$

### C.4. Oracle Complexity Analysis of MoSSP-P

We begin by establishing a recursive relationship between $\mathbb{E}[\|\boldsymbol{e}^{k+1}\|^2]$ and $\mathbb{E}[\|\boldsymbol{e}^k\|^2]$. The following lemma is adapted from (Lemma 5.2, Gao et al., 2024). For completeness, we provide the proof.

**Lemma C.7.** *Suppose that Assumptions 2.1-2.2 hold. Then, for any $k \ge 0$ and $0 < \alpha_k < 1$, it holds that*

$$
\mathbb{E}[\|\boldsymbol{e}^{k+1}\|^2] \le (1 - \alpha_k)\mathbb{E}[\|\boldsymbol{e}^k\|^2] + \frac{1}{\alpha_k} L_f^2 \mathbb{E}[\|\boldsymbol{w}^{k+1} - \boldsymbol{w}^k\|^2] + \alpha_k^2 \sigma^2. \tag{C.32}
$$

*Proof.* Let $\xi^{[k]} := \{\xi^0, \ldots, \xi^k\}$ denote the collection of i.i.d. samples drawn up to iteration $k$ in MoSSP-P, and let $\mathbb{E}[\cdot \mid \xi^{[k]}]$

denote the corresponding conditional expectation. Recalling (C.5) and (3.6), one has

$$
\begin{aligned}
&\mathbb{E}[\|\boldsymbol{e}^{k+1}\|^2|\xi^{[k]}] \\
&= \mathbb{E}[\|\boldsymbol{S}^{k+1} - \nabla Q_{\rho_{k+1}}(\boldsymbol{x}^{k+1})\|^2|\xi^{[k]}] \\
&= \mathbb{E}[\|\boldsymbol{s}^{k+1} - \nabla f(\boldsymbol{x}^{k+1})\|^2|\xi^{[k]}] \\
&= \mathbb{E}[\|(1-\alpha_k)(\boldsymbol{s}^k - \nabla f(\boldsymbol{x}^{k+1})) + \alpha_k(\nabla \mathbf{f}(\boldsymbol{x}^{k+1}, \xi^{k+1}) - \nabla f(\boldsymbol{x}^{k+1}))\|^2|\xi^{[k]}] \\
&= (1-\alpha_k)^2 \mathbb{E}[\|\boldsymbol{s}^k - \nabla f(\boldsymbol{x}^{k+1})\|^2|\xi^{[k]}] + \alpha_k^2 \mathbb{E}[\|\nabla \mathbf{f}(\boldsymbol{x}^{k+1}, \xi^{k+1}) - \nabla f(\boldsymbol{x}^{k+1})\|^2|\xi^{[k]}] \\
&\quad + 2(1-\alpha_k)\alpha_k \mathbb{E}[\langle \boldsymbol{s}^k - \nabla f(\boldsymbol{x}^{k+1}), \nabla \mathbf{f}(\boldsymbol{x}^{k+1}, \xi^{k+1}) - \nabla f(\boldsymbol{x}^{k+1})\rangle|\xi^{[k]}].
\end{aligned}
\tag{C.33}
$$

Notice that it follows from Assumption 2.2 that

$$
\begin{aligned}
&\mathbb{E}[\langle \boldsymbol{s}^k - \nabla f(\boldsymbol{x}^{k+1}), \nabla \mathbf{f}(\boldsymbol{x}^{k+1}, \xi^{k+1}) - \nabla f(\boldsymbol{x}^{k+1})\rangle|\xi^{[k]}] \\
&= \langle \boldsymbol{s}^k - \nabla f(\boldsymbol{x}^{k+1}), \mathbb{E}[\nabla \mathbf{f}(\boldsymbol{x}^{k+1}, \xi^{k+1}) - \nabla f(\boldsymbol{x}^{k+1})|\xi^{[k]}]\rangle = 0.
\end{aligned}
\tag{C.34}
$$

Then, substituting this into (C.33) yields

$$
\begin{aligned}
&\mathbb{E}[\|\boldsymbol{e}^{k+1}\|^2|\xi^{[k]}] \\
&= (1-\alpha_k)^2 \mathbb{E}[\|\boldsymbol{s}^k - \nabla f(\boldsymbol{x}^k) + \nabla f(\boldsymbol{x}^k) - \nabla f(\boldsymbol{x}^{k+1})\|^2|\xi^{[k]}] \\
&\quad + \alpha_k^2 \mathbb{E}[\|\nabla \mathbf{f}(\boldsymbol{x}^{k+1}, \xi^{k+1}) - \nabla f(\boldsymbol{x}^{k+1})\|^2|\xi^{[k]}] \\
&\leq (1-\alpha_k)^2 \left(\frac{1}{1-\alpha_k}\mathbb{E}[\|\boldsymbol{e}^k\|^2|\xi^{[k]}] + \frac{1}{\alpha_k}\|\nabla f(\boldsymbol{x}^k) - \nabla f(\boldsymbol{x}^{k+1})\|^2\right) + \alpha_k^2 \sigma^2 \\
&= (1-\alpha_k)\mathbb{E}[\|\boldsymbol{e}^k\|^2|\xi^{[k]}] + \frac{(1-\alpha_k)^2}{\alpha_k}\|\nabla f(\boldsymbol{x}^k) - \nabla f(\boldsymbol{x}^{k+1})\|^2 + \alpha_k^2 \sigma^2 \\
&\leq (1-\alpha_k)\mathbb{E}[\|\boldsymbol{e}^k\|^2|\xi^{[k]}] + \frac{1}{\alpha_k}L_f^2\|\boldsymbol{w}^{k+1} - \boldsymbol{w}^k\|^2 + \alpha_k^2 \sigma^2,
\end{aligned}
\tag{C.35}
$$

where the second inequality uses Young's inequality $\|a+b\|^2 \leq (1+\gamma)\|a\|^2 + (1+\gamma^{-1})\|b\|^2$ with $\gamma = \frac{\alpha_k}{1-\alpha_k} > 0$ and the bounded variance condition in Assumption 2.2, and the third inequality uses the $L_f$-smoothness in Assumption 2.1 along with $(1-\alpha_k)^2 \leq 1$ for $0 < \alpha_k < 1$. Taking the full expectation on both sides of this inequality yields (C.32), completing the proof. $\square$

As can be seen from Lemma C.6, the convergence rate is affected by the accumulated error $\frac{1}{K}\sum_{k=0}^{K-1}\mathbb{E}[\|\boldsymbol{e}^k\|^2]$. To establish an upper bound for this accumulated error, we impose the following parameter setting

$$
\frac{1}{\alpha_k} \leq \frac{1}{2\mu_k}, \quad L_{\rho_k} + \frac{2}{\alpha_k}L_f^2 \leq \frac{3}{4}\frac{1}{2\mu_k}.
\tag{C.36}
$$

**Lemma C.8.** *Suppose that Assumptions 2.1-2.2 hold, and the parameters satisfy (C.7) and (C.36). Then, it holds that for $K \geq 1$,*

$$
\begin{aligned}
&\frac{1}{K}\sum_{k=0}^{K-1}\mu_k\mathbb{E}[\|\boldsymbol{e}^k\|^2] \\
&\leq \frac{3(\mathcal{L}_{\rho_0,\mu_0}(\boldsymbol{w}^0) - \mathcal{L}^*)}{K} + \frac{2\mathbb{E}[\|\boldsymbol{e}^0\|^2]}{K} + \frac{3}{K}\sum_{k=0}^{K-1}\left(\frac{C^2}{2}(\rho_{k+1}-\rho_k) + \mathbb{E}[\Delta_{k+1}]\right) + \frac{2}{K}\sum_{k=0}^{K-1}\alpha_k^2\sigma^2.
\end{aligned}
\tag{C.37}
$$

*Furthermore, if we set $\mu_k \equiv \mu$, $\alpha_k \equiv \alpha$, $\rho_k \equiv \rho$ for $k \geq 1$, it holds that for any $K \geq 1$,*

$$
\frac{1}{K}\sum_{k=0}^{K-1}\mathbb{E}[\|\boldsymbol{e}^k\|^2] \leq \frac{3(\mathcal{L}_{\rho,\mu}(\boldsymbol{w}^0) - \mathcal{L}^*)}{\mu K} + \frac{2\mathbb{E}[\|\boldsymbol{e}^0\|^2]}{\mu K} + \frac{2\alpha^2\sigma^2}{\mu}.
\tag{C.38}
$$

*Proof.* Recalling (C.8), we take full expectation on both sides of the inequality and substitute it into (C.32) to yield

$$
\begin{aligned}
&\mathbb{E}[\mathcal{L}_{\rho_{k+1},\mu_{k+1}}(\boldsymbol{w}^{k+1}) + \|\boldsymbol{e}^{k+1}\|^2] \\
&\leq \mathbb{E}[\mathcal{L}_{\rho_k,\mu_k}(\boldsymbol{w}^k) + \|\boldsymbol{e}^k\|^2] + \frac{1}{2}\left(-\frac{1}{2\mu_k} + L_{\rho_k} + \frac{2}{\alpha_k}L_f^2\right)\mathbb{E}[\|\boldsymbol{w}^{k+1} - \boldsymbol{w}^k\|^2] \\
&\quad + (\mu_k - \alpha_k)\mathbb{E}[\|\boldsymbol{e}^k\|^2] + \frac{\rho_{k+1} - \rho_k}{2}C^2 + \mathbb{E}[\Delta_{k+1}] + \alpha_k^2\sigma^2 \\
&\leq \mathbb{E}[\mathcal{L}_{\rho_k,\mu_k}(\boldsymbol{w}^k)] + (1 - \alpha_k + 2\mu_k)\|\boldsymbol{e}^k\|^2] - \mu_k\mathbb{E}[\|\boldsymbol{e}^k\|^2] + \frac{\rho_{k+1} - \rho_k}{2}C^2 \\
&\quad + \frac{1}{2}\left(-\frac{1}{2\mu_k} + L_{\rho_k} + \frac{2}{\alpha_k}L_f^2\right)\mathbb{E}[\|\boldsymbol{w}^{k+1} - \boldsymbol{w}^k\|^2] + \mathbb{E}[\Delta_{k+1}] + \alpha_k^2\sigma^2 \\
&\leq \mathbb{E}[\mathcal{L}_{\rho_k,\mu_k}(\boldsymbol{w}^k) + \|\boldsymbol{e}^k\|^2] - \frac{\mathbb{E}[\|\boldsymbol{w}^{k+1} - \boldsymbol{w}^k\|^2]}{16\mu_k} - \mu_k\mathbb{E}[\|\boldsymbol{e}^k\|^2] + \frac{\rho_{k+1} - \rho_k}{2}C^2 + \mathbb{E}[\Delta_{k+1}] + \alpha_k^2\sigma^2,
\end{aligned}
\tag{C.39}
$$

where the last inequality follows from $1 - \alpha_k + 2\mu_k \leq 1$ and that $-\frac{1}{2}(2\mu_k)^{-1} + \frac{1}{2}L_{\rho_k} + \alpha_k^{-1}L_f^2 \leq -\frac{1}{16\mu_k}$ under (C.36). Taking expectation in (C.9) and using $\mu_k L_{\rho_k} \leq \frac{1}{4}$, we obtain

$$
\begin{aligned}
&\frac{1}{8}\frac{1}{2\mu_k}\mathbb{E}[\|\boldsymbol{w}^{k+1} - \boldsymbol{w}^k\|^2] \\
&\leq \frac{1}{2}(\mathbb{E}[\mathcal{L}_{\rho_k,\mu_k}(\boldsymbol{w}^k)] - \mathbb{E}[\mathcal{L}_{\rho_{k+1},\mu_{k+1}}(\boldsymbol{w}^{k+1})] + \frac{\rho_{k+1} - \rho_k}{2}C^2 + \mathbb{E}[\Delta_{k+1}] + \mu_k\mathbb{E}[\|\boldsymbol{e}^k\|^2]).
\end{aligned}
$$

Substituting this into (C.39) and rearranging the inequality yields

$$
\begin{aligned}
\mu_k\mathbb{E}[\|\boldsymbol{e}^k\|^2] &\leq 3\big(\mathbb{E}[\mathcal{L}_{\rho_k,\mu_k}(\boldsymbol{w}^k)] - \mathbb{E}[\mathcal{L}_{\rho_{k+1},\mu_{k+1}}(\boldsymbol{w}^{k+1})] + \frac{\rho_{k+1} - \rho_k}{2}C^2 + \mathbb{E}[\Delta_{k+1}]\big) \\
&\quad + 2(\mathbb{E}[\|\boldsymbol{e}^k\|^2] - \mathbb{E}[\|\boldsymbol{e}^{k+1}\|^2]) + 2\alpha_k^2\sigma^2.
\end{aligned}
\tag{C.40}
$$

Summing over $k = 0, \ldots, K - 1$ and dividing by $K$ yields (C.37). When $\mu_k \equiv \mu$, $\alpha_k \equiv \alpha$, $\rho_k \equiv \rho$, one has that $\rho_{k+1} - \rho_k = 0$ and $\Delta_{k+1} = 0$, implying (C.38) and completing the proof. $\square$

We now characterize the measurements in (2.7) for MoSSP-P with constant parameters $\rho_k \equiv \rho$, $\mu_k \equiv \mu$, $\alpha_k \equiv \alpha$.

**Lemma C.9.** *Suppose that the conditions of Lemma C.8 hold. Then, it holds that*

$$
\begin{aligned}
&\max\{\mathbb{E}[\|\boldsymbol{u}^{R+1}\|^2], \mathbb{E}[\|\boldsymbol{x}^{R+1} - \mathrm{prox}_{\mu g}(\boldsymbol{z}^R)\|^2]\} \\
&\leq \frac{1}{K}\frac{96(\frac{1}{16} + \beta^{-2} + \frac{1}{3}(\mu/\beta)^2) + 9}{\mu}(\mathcal{L}_{\rho,\mu}(\boldsymbol{w}^0) - \mathcal{L}^*) + \frac{(48(\frac{1}{16} + \beta^{-2} + \frac{1}{3}(\mu/\beta)^2) + 6)\mathbb{E}[\|\boldsymbol{e}^0\|^2]}{\mu K} \\
&\quad + \frac{(48(\frac{1}{16} + \beta^{-2} + \frac{1}{3}(\mu/\beta)^2) + 6)\alpha^2\sigma^2}{\mu},
\end{aligned}
\tag{C.41}
$$

*and*

$$
\begin{aligned}
&\mathbb{E}[\|\nabla\boldsymbol{c}(\boldsymbol{x}^{R+1})\boldsymbol{c}(\boldsymbol{x}^{R+1})\|^2] \\
&\leq \frac{192(\frac{1}{16} + \beta^{-2}) + 18}{\mu\rho^2 K}(\mathcal{L}_{\rho,\mu}(\boldsymbol{w}^0) - \mathcal{L}^*) + \frac{(12 + 96(\frac{1}{16} + \beta^{-2}))\mathbb{E}[\|\boldsymbol{e}^0\|^2]}{\mu\rho^2 K} \\
&\quad + \frac{(12 + 96(\frac{1}{16} + \beta^{-2}))\alpha^2\sigma^2}{\mu\rho^2} + \frac{18G^2}{\rho^2}.
\end{aligned}
\tag{C.42}
$$

*Proof.* Substituting (C.38) into (C.29) and noticing that all parameters are constant, we derive the desired bound in (C.41). Then, substituting (C.38) into (C.25) yields

$$
\begin{aligned}
&\frac{1}{K}\sum_{k=0}^{K-1}\mathbb{E}[\|\boldsymbol{u}^{k+1}\|^2] \\
&\leq \frac{9 + 96(\frac{1}{16} + \beta^{-2})}{\mu K}(\mathcal{L}_{\rho,\mu}(\boldsymbol{w}^0) - \mathcal{L}^*) + \frac{6 + 48(\frac{1}{16} + \beta^{-2})}{\mu K}\mathbb{E}[\|\boldsymbol{e}^0\|^2] + \frac{6 + 48(\frac{1}{16} + \beta^{-2})}{\mu}\alpha^2\sigma^2.
\end{aligned}
$$

Then, combining this bound with (C.21) from Lemma C.4 directly gives (C.42), completing the proof. □

### C.4.1. PROOF OF LEMMA 3.1

Selecting appropriate parameters is crucial as the final oracle complexity depends on these choices. To satisfy (C.7) and (C.36), we adopt the following parameter setting:

$$\rho_k \equiv \rho = \rho_0 K^l, \ \mu_k \equiv \mu = \frac{\mu_0}{K^\tau \max\{L_f, \tilde{L}\}}, \ \alpha_k \equiv \alpha = \frac{\alpha_0 \mu_0}{K^\tau}, 0 < \beta \le 1, \tag{C.43}$$

where $0 < l \le \tau \le 2l < 1, 0 < \mu_0 \le \min\{\frac{1}{4\rho_0}, \frac{1}{4L_f}\}, \alpha_0 = \frac{2\gamma}{\max\{L_f, \tilde{L}\}}$, and $\gamma \ge \max\{1, 8L_f^2\}$ are constants independent of $K$. Then, substituting (C.43) into Lemma C.9 directly yields (3.11). To reach the rate of $\mathcal{O}(K^{-1/2})$ under approximate feasible initialization, we set $\tau = 2l = \frac{1}{2}$. We are now ready to prove Theorem 3.1.

### C.4.2. PROOF OF THEOREM 3.1

*Proof.* First, under Assumption 2.3, combining (C.22) with (C.42) yields that the upper bound on $\mathbb{E}[\|c(x^{R+1})\|^2]$ is in the same order as $\mathbb{E}[\|\nabla c(x^{R+1})c(x^{R+1})\|^2]$.

Second, by the approximate feasibility condition $\|c(x^0)\|^2 = \mathcal{O}(K^{-l})$, we have $\mathcal{L}_{\rho,\mu}(w^0) + 1 = \mathcal{O}(1)$ and the two upper bounds in (3.11) are simplified to

$$\mathcal{O}\big(\max\{K^{\tau-1}, K^{-\tau}\}\big) \quad \text{and} \quad \mathcal{O}\big(\max\{K^{-2l+\tau-1}, K^{-2l}\}\big).$$

If we choose $l = \frac{1}{4}$ and $\tau = \frac{1}{2}$, both reduce to $\mathcal{O}(K^{-1/2})$. Hence, to obtain a stochastic $\varepsilon$-stationary point, $K$ is of order $\mathcal{O}(\varepsilon^{-4})$, and the associated oracle complexity is $\mathcal{O}(\varepsilon^{-4})$. Under Assumption 2.3, the complexity to reach a stochastic $\varepsilon$-KKT point is $\mathcal{O}(\varepsilon^{-4})$. □

### C.4.3. PROOF OF COROLLARY 3.1

Without assuming initial approximate feasibility, we have $\mathcal{L}_{\rho,\mu}(w^0) + 1 = \mathcal{O}(K^l)$ under $\|c(x^0)\|^2 = \mathcal{O}(1)$ and $\rho = \mathcal{O}(K^l)$. Moreover, the one-sample Polyak initialization and Assumption 2.2 give $\mathbb{E}[\|e^0\|^2] = \mathcal{O}(1)$. Combining these bounds with (3.11), we obtain

$$\max\{\mathbb{E}[\|u^{R+1}\|^2], \mathbb{E}[\|x^{R+1} - \text{prox}_{\mu g}(z^R)\|^2]\} = \mathcal{O}\big(\max\{K^{l+\tau-1}, K^{\tau-1}, K^{-\tau}\}\big),$$

and

$$\mathbb{E}[\|\nabla c(x^{R+1})c(x^{R+1})\|^2] = \mathcal{O}\big(\max\{K^{-l+\tau-1}, K^{-2l+\tau-1}, K^{-2l}\}\big).$$

Choosing $l = \frac{1}{5}$ and $\tau = \frac{2}{5}$ gives $\rho = \mathcal{O}(K^{1/5}), \mu = \mathcal{O}(K^{-2/5})$, and $\alpha = \mathcal{O}(K^{-2/5})$. The two bounds reduce to

$$\max\{\mathbb{E}[\|u^{R+1}\|^2], \mathbb{E}[\|x^{R+1} - \text{prox}_{\mu g}(z^R)\|^2]\} = \mathcal{O}(K^{-2/5}),$$

and

$$\mathbb{E}[\|\nabla c(x^{R+1})c(x^{R+1})\|^2] = \mathcal{O}(K^{-2/5}).$$

Hence, to make the squared residuals no larger than $\varepsilon^2$, it suffices to take $K = \mathcal{O}(\varepsilon^{-5})$. Since MoSSP-P uses one stochastic first-order oracle call per iteration, the total oracle complexity for finding a stochastic $\varepsilon$-stationary point is $\mathcal{O}(\varepsilon^{-5})$. Under Assumption 2.3, the same bound on $\mathbb{E}[\|\nabla c(x^{R+1})c(x^{R+1})\|^2]$ yields $\mathbb{E}[\|c(x^{R+1})\|^2] = \mathcal{O}(K^{-2/5})$, and the same oracle complexity holds for finding a stochastic $\varepsilon$-KKT point. The corresponding certificate is $\bar{y} = \text{prox}_{\mu g}(z^R), \bar{\lambda} = \rho c(x^{R+1})$, and $\bar{u} = u^{R+1}$; the inclusion in (2.5) follows almost surely from (C.6).

### C.5. Oracle Complexity Analysis of MoSSP-R

We now establish a recursive relationship between $\mathbb{E}[\|e^{k+1}\|^2]$ and $\mathbb{E}[\|e^k\|^2]$ in the following lemma. The proof follows from Xu & Xu (2023). For completeness, we provide the proof.

**Lemma C.10.** *Suppose that Assumptions 2.1, 2.2 and 3.1 hold. Then, for any $k \ge 0$, it holds that*

$$\mathbb{E}\left[\|e^{k+1}\|^2\right] \le (1 - \alpha_k)\mathbb{E}\left[\|e^k\|^2\right] + 2L_f^2 \mathbb{E}\left[\|w^{k+1} - w^k\|^2\right] + 2\alpha_k^2 \sigma^2. \tag{C.44}$$

*Proof.* Let $\xi^{[k]} := \{\xi^0, \ldots, \xi^k\}$ denote the collection of i.i.d. samples drawn up to iteration $k$ in MoSSP-R, and let $\mathbb{E}[\cdot \mid \xi^{[k]}]$ denote the corresponding conditional expectation. Recalling (C.5) and (3.14), it holds that

$$
\begin{aligned}
&\mathbb{E}[\|\boldsymbol{e}^{k+1}\|^2 | \xi^{[k]}] \\
&= \mathbb{E}[\|\boldsymbol{D}^{k+1} - \nabla Q_{\rho_{k+1}}(\boldsymbol{x}^{k+1})\|^2 | \xi^{[k]}] \\
&= \mathbb{E}[\|\boldsymbol{d}^{k+1} - \nabla f(\boldsymbol{x}^{k+1})\|^2 | \xi^{[k]}] \\
&= \mathbb{E}[\|\nabla \mathbf{f}(\boldsymbol{x}^{k+1}, \xi^{k+1}) - \nabla f(\boldsymbol{x}^{k+1}) + (1 - \alpha_k)(\boldsymbol{d}^k - \nabla \mathbf{f}(\boldsymbol{x}^k, \xi^{k+1}))\|^2 | \xi^{[k]}] \\
&= \mathbb{E}[\|(1 - \alpha_k)\boldsymbol{e}^k + \nabla \mathbf{f}(\boldsymbol{x}^{k+1}, \xi^{k+1}) - \nabla f(\boldsymbol{x}^{k+1}) + (1 - \alpha_k)(\nabla f(\boldsymbol{x}^k) - \nabla \mathbf{f}(\boldsymbol{x}^k, \xi^{k+1}))\|^2 | \xi^{[k]}] \\
&= (1 - \alpha_k)^2 \mathbb{E}[\|\boldsymbol{e}^k\|^2 | \xi^{[k]}] + \mathbb{E}\big[\|\nabla \mathbf{f}(\boldsymbol{x}^{k+1}, \xi^{k+1}) - \nabla f(\boldsymbol{x}^{k+1}) + (1 - \alpha_k)(\nabla f(\boldsymbol{x}^k) - \nabla \mathbf{f}(\boldsymbol{x}^k, \xi^{k+1}))\|^2 | \xi^{[k]}\big], \quad \text{(C.45)}
\end{aligned}
$$

where (C.45) follows from Assumption 2.2, which ensures that

$$
\begin{aligned}
\mathbb{E}[\langle \nabla \mathbf{f}(\boldsymbol{x}^{k+1}, \xi^{k+1}) - \nabla f(\boldsymbol{x}^{k+1}), \boldsymbol{e}^k \rangle \mid \xi^{[k]}] &= 0, \\
\mathbb{E}[\langle \nabla \mathbf{f}(\boldsymbol{x}^k, \xi^{k+1}) - \nabla f(\boldsymbol{x}^k), \boldsymbol{e}^k \rangle \mid \xi^{[k]}] &= 0.
\end{aligned}
$$

Let us focus on the second term (C.45) by rewriting this term as

$$
\begin{aligned}
&\mathbb{E}[\|\nabla \mathbf{f}(\boldsymbol{x}^{k+1}, \xi^{k+1}) - \nabla f(\boldsymbol{x}^{k+1}) + (1 - \alpha_k)(\nabla f(\boldsymbol{x}^k) - \nabla \mathbf{f}(\boldsymbol{x}^k, \xi^{k+1}))\|^2 | \xi^{[k]}] \\
&= \mathbb{E}[\|\alpha_k(\nabla \mathbf{f}(\boldsymbol{x}^{k+1}, \xi^{k+1}) - \nabla f(\boldsymbol{x}^{k+1})) + (1 - \alpha_k)(\nabla f(\boldsymbol{x}^k) - \nabla f(\boldsymbol{x}^{k+1}) \\
&\qquad\qquad + \nabla \mathbf{f}(\boldsymbol{x}^{k+1}, \xi^{k+1}) - \nabla \mathbf{f}(\boldsymbol{x}^k, \xi^{k+1}))\|^2 | \xi^{[k]}] \\
&\leq 2\alpha_k^2 \mathbb{E}[\|\nabla \mathbf{f}(\boldsymbol{x}^{k+1}, \xi^{k+1}) - \nabla f(\boldsymbol{x}^{k+1})\|^2 | \xi^{[k]}] \\
&\qquad + 2(1 - \alpha_k)^2 \mathbb{E}[\|\nabla f(\boldsymbol{x}^k) - \nabla f(\boldsymbol{x}^{k+1}) + \nabla \mathbf{f}(\boldsymbol{x}^{k+1}, \xi^{k+1}) - \nabla \mathbf{f}(\boldsymbol{x}^k, \xi^{k+1})\|^2 | \xi^{[k]}] \\
&\leq 2\alpha_k^2 \sigma^2 + 2(1 - \alpha_k)^2 \mathbb{E}[\|\nabla \mathbf{f}(\boldsymbol{x}^k, \xi^{k+1}) - \nabla \mathbf{f}(\boldsymbol{x}^{k+1}, \xi^{k+1})\|^2 | \xi^{[k]}] \quad\quad\quad\quad\quad \text{(C.46)} \\
&\leq 2\alpha_k^2 \sigma^2 + 2L_f^2 \mathbb{E}[\|\boldsymbol{w}^{k+1} - \boldsymbol{w}^k\|^2 | \xi^{[k]}], \quad\quad\quad\quad\quad\quad\quad\quad\quad\quad\quad\quad\quad\quad\quad \text{(C.47)}
\end{aligned}
$$

where (C.46) uses $\mathbb{E}\big[\nabla \mathbf{f}(\boldsymbol{x}^{k+1}, \xi^{k+1}) - \nabla \mathbf{f}(\boldsymbol{x}^k, \xi^{k+1}) | \xi^{[k]}\big] = \nabla f(\boldsymbol{x}^{k+1}) - \nabla f(\boldsymbol{x}^k)$ and (C.47) uses Assumption 3.1 and $0 < \alpha_k \leq 1$. Then, substituting (C.47) into (C.45) and taking full expectation on both sides of this inequality, we obtain (C.44) and complete the proof. $\qquad\square$

For MoSSP-R, we similarly use the recursive structure of $\mathbb{E}[\|\boldsymbol{e}^k\|^2]$ from Lemma C.10 to establish a weighted cumulative error bound. We assume the following additional parameter conditions on $\alpha_k$ and $\mu_k$:

$$
0 < 32\mu_k^2 L_f^2 \leq \alpha_k \leq 1, \quad \forall\, k \geq 0. \tag{C.48}
$$

**Lemma C.11.** *Suppose the assumptions of Lemma C.10 hold and the parameters satisfy both (C.7) and (C.48). Then, for any $K \geq 1$, it holds that*

$$
\begin{aligned}
&\frac{1}{K} \sum_{k=0}^{K-1} \alpha_k \mathbb{E}[\|\boldsymbol{e}^k\|^2] \\
&\leq \frac{32\mu_0 L_f^2 (\mathcal{L}_{\rho_0, \mu_0}(\boldsymbol{w}^0) - \mathcal{L}^*)}{K} + \frac{2\mathbb{E}[\|\boldsymbol{e}^0\|^2]}{K} + \frac{32\mu_0 L_f^2}{K} \sum_{k=0}^{K-1} \left( \frac{\rho_{k+1} - \rho_k}{2} C^2 + \mathbb{E}[\Delta_{k+1}] \right) + \frac{1}{K} \sum_{k=0}^{K-1} 4\alpha_k^2 \sigma^2. \quad \text{(C.49)}
\end{aligned}
$$

*Furthermore, if we set $\mu_k \equiv \mu$, $\alpha_k \equiv \alpha$, and $\rho_k \equiv \rho$ for $k \geq 0$, it holds that*

$$
\frac{1}{K} \sum_{k=0}^{K-1} \mathbb{E}[\|\boldsymbol{e}^k\|^2] \leq \frac{32\mu L_f^2 (\mathcal{L}_{\rho, \mu}(\boldsymbol{w}^0) - \mathcal{L}^*)}{\alpha K} + \frac{2\mathbb{E}[\|\boldsymbol{e}^0\|^2]}{\alpha K} + 4\alpha \sigma^2. \tag{C.50}
$$

*Proof.* From (C.44), we obtain

$$
\alpha_k \mathbb{E}[\|\boldsymbol{e}^k\|^2] \leq \mathbb{E}[\|\boldsymbol{e}^k\|^2] - \mathbb{E}[\|\boldsymbol{e}^{k+1}\|^2] + 2\alpha_k^2 \sigma^2 + 2L_f^2 \mathbb{E}[\|\boldsymbol{w}^{k+1} - \boldsymbol{w}^k\|^2]. \tag{C.51}
$$

On the other hand, taking expectation on both sides of (C.9) yields

$$
\mathbb{E}[\|\boldsymbol{w}^{k+1} - \boldsymbol{w}^k\|^2]
$$
$$
\leq 8\mu_k \mathbb{E}[\mathcal{L}_{\rho_k,\mu_k}(\boldsymbol{w}^k) - \mathcal{L}_{\rho_{k+1},\mu_{k+1}}(\boldsymbol{w}^{k+1}) + \frac{\rho_{k+1} - \rho_k}{2}C^2 + \Delta_{k+1} + \mu_k\|\boldsymbol{e}^k\|^2], \tag{C.52}
$$

where the inequality follows from $1 - 2\mu_k L_\rho \geq \frac{1}{2}$, which is implied by $\mu_k L_\rho \leq \frac{1}{4}$.

Then, substituting (C.52) into (C.51), using $\mu_k \leq \mu_0$ for the non-error terms, summing over $k = 0, \ldots, K-1$, and dividing by $K$, we obtain

$$
\frac{1}{K}\sum_{k=0}^{K-1}(\alpha_k - 16\mu_k^2 L_f^2)\mathbb{E}[\|\boldsymbol{e}^k\|^2]
$$
$$
\leq \frac{1}{K}\sum_{k=0}^{K-1}(\mathbb{E}[\|\boldsymbol{e}^k\|^2] - \mathbb{E}[\|\boldsymbol{e}^{k+1}\|^2]) + \frac{1}{K}\sum_{k=0}^{K-1}2\alpha_k^2\sigma^2
$$
$$
+ \frac{16\mu_0 L_f^2}{K}\sum_{k=0}^{K-1}\left(\mathcal{L}_{\rho_k,\mu_k}(\boldsymbol{w}^k) - \mathcal{L}_{\rho_{k+1},\mu_{k+1}}(\boldsymbol{w}^{k+1}) + \frac{\rho_{k+1} - \rho_k}{2}C^2 + \mathbb{E}[\Delta_{k+1}]\right). \tag{C.53}
$$

Since $\alpha_k \geq 32\mu_k^2 L_f^2$, we have $\alpha_k - 16\mu_k^2 L_f^2 \geq \frac{\alpha_k}{2}$, yielding

$$
\frac{1}{K}\sum_{k=0}^{K-1}\frac{\alpha_k}{2}\mathbb{E}[\|\boldsymbol{e}^k\|^2]
$$
$$
\leq \frac{\mathbb{E}[\|\boldsymbol{e}^0\|^2]}{K} + \frac{16\mu_0 L_f^2(\mathcal{L}_{\rho_0,\mu_0}(\boldsymbol{w}^0) - \mathcal{L}^*)}{K} + \frac{16\mu_0 L_f^2}{K}\sum_{k=0}^{K-1}\left(\frac{\rho_{k+1} - \rho_k}{2}C^2 + \mathbb{E}[\Delta_{k+1}]\right) + \frac{1}{K}\sum_{k=0}^{K-1}2\alpha_k^2\sigma^2,
$$

where rearranging the inequality yields (C.49) and completes the proof. □

We now characterize the measurements in (2.7) for MoSSP-R with constant parameters $\rho_k \equiv \rho$, $\mu_k \equiv \mu$, $\alpha_k \equiv \alpha$ for any $k \geq 0$.

**Lemma C.12.** *Suppose that the conditions of Lemma C.11 hold. Then, it holds that*

$$
\max\{\mathbb{E}[\|\boldsymbol{u}^{R+1}\|^2], \mathbb{E}[\|\boldsymbol{x}^{R+1} - \mathrm{prox}_{\mu g}(\boldsymbol{z}^R)\|^2]\}
$$
$$
\leq \frac{1}{K}\left(\frac{(96 + 768(\frac{1}{16} + \beta^{-2} + \frac{1}{3}(\mu/\beta)^2))\mu L_f^2}{\alpha} + \frac{24(\frac{1}{16} + \beta^{-2} + \frac{1}{3}(\mu/\beta)^2)}{\mu}\right)(\mathcal{L}_{\rho,\mu}(\boldsymbol{w}^0) - \mathcal{L}^*)
$$
$$
+ \frac{(48(\frac{1}{16} + \beta^{-2} + \frac{1}{3}(\mu/\beta)^2) + 6)\mathbb{E}[\|\boldsymbol{e}^0\|^2]}{\alpha K} + (96(\frac{1}{16} + \beta^{-2} + \frac{1}{3}(\mu/\beta)^2) + 12)\alpha\sigma^2, \tag{C.54}
$$

*and*

$$
\mathbb{E}[\|\nabla \boldsymbol{c}(\boldsymbol{x}^{R+1})\boldsymbol{c}(\boldsymbol{x}^{R+1})\|^2]
$$
$$
\leq \frac{2}{\rho^2 K}\left(\frac{(144 + 768\beta^{-2})\mu L_f^2}{\alpha} + \frac{24(\frac{1}{16} + \beta^{-2})}{\mu}\right)(\mathcal{L}_{\rho,\mu}(\boldsymbol{w}^0) - \mathcal{L}^*) + \frac{(12 + 96(\frac{1}{16} + \beta^{-2}))\mathbb{E}[\|\boldsymbol{e}^0\|^2]}{\alpha\rho^2 K}
$$
$$
+ \frac{(24 + 192(\frac{1}{16} + \beta^{-2}))\alpha\sigma^2}{\rho^2} + \frac{18G^2}{\rho^2}. \tag{C.55}
$$

*Proof.* For (C.54), we substitute (C.50) into (C.29) to obtain the desired results.

For (C.55), under the constant parameter setting, we substitute (C.50) into (C.25), obtaining

$$\frac{1}{K} \sum_{k=0}^{K-1} \mathbb{E}[\|\boldsymbol{u}^{k+1}\|^2]$$

$$\leq \frac{1}{K} \left( \frac{(96 + 768(\frac{1}{16} + \beta^{-2}))\mu L_f^2}{\alpha} + \frac{24(\frac{1}{16} + \beta^{-2})}{\mu} \right) (\mathcal{L}_{\rho,\mu}(\boldsymbol{w}^0) - \mathcal{L}^*)$$

$$+ \frac{2(3 + 24(\frac{1}{16} + \beta^{-2}))\mathbb{E}[\|\boldsymbol{e}^0\|^2]}{\alpha K} + (12 + 96(\frac{1}{16} + \beta^{-2}))\alpha\sigma^2.$$

Combining this inequality with (C.21) directly yields (C.55), completing the proof. $\square$

To ensure the parameter conditions (C.7) and (C.48), we set the parameters as follows

$$
\begin{aligned}
\rho_k &\equiv \rho = \rho_0 K^l, \\
\mu_k &\equiv \mu = \frac{\mu_0}{K^l \max\{L_f, \tilde{L}\}}, \\
\alpha_k &\equiv \alpha = \frac{16\alpha_0\mu_0^2}{K^\tau}, \quad 0 < \beta \leq 1,
\end{aligned}
\tag{C.56}
$$

where $\rho_0 > 0$, $0 < \mu_0 \leq \min\{\frac{1}{4\rho_0}, \frac{\max\{L_f,\tilde{L}\}}{4\sqrt{2}L_f}\}$, $0 < \tau \leq 2l < 1$ and $\alpha_0 \in [\frac{2L_f^2}{\max\{L_f,\tilde{L}\}^2}, \frac{1}{16\mu_0^2}]$ are given constants independent of $K$. Then, Lemma C.12 yields the following convergence rate of MoSSP-R for finding a *stochastic $\varepsilon$-stationary point*:

$$
\begin{cases}
\max\{\mathbb{E}[\|\boldsymbol{u}^{R+1}\|^2], \mathbb{E}[\|\boldsymbol{x}^{R+1} - \mathrm{prox}_{\mu g}(\boldsymbol{z}^R)\|^2]\} \\
\quad = \mathcal{O}\big(\max\{K^{l-1}\left(\mathcal{L}_{\rho,\mu}(\boldsymbol{w}^0) + 1\right), K^{\tau-1}\mathbb{E}[\|\boldsymbol{e}^0\|^2], K^{-\tau}\}\big), \\
\mathbb{E}[\|\nabla\boldsymbol{c}(\boldsymbol{x}^{R+1})\boldsymbol{c}(\boldsymbol{x}^{R+1})\|^2] \\
\quad = \mathcal{O}\big(\max\{K^{-l-1}\left(\mathcal{L}_{\rho,\mu}(\boldsymbol{w}^0) + 1\right), K^{-2l+\tau-1}\mathbb{E}[\|\boldsymbol{e}^0\|^2], K^{-\tau}\}\big).
\end{cases}
\tag{C.57}
$$

As can be seen in (C.57), $\mathbb{E}[\|\boldsymbol{e}^0\|^2]$ affects the order in (C.57). Therefore, with an initial batch size $b_0 = \mathcal{O}(K^l)$, one has $\mathbb{E}[\|\boldsymbol{e}^0\|^2] = \mathcal{O}(K^{-l})$. Similar to the proof of Appendix C.4.2, under Assumption 2.3, combining (C.22) with (C.55) yields that the upper bound on $\mathbb{E}[\|\boldsymbol{c}(\boldsymbol{x}^{R+1})\|^2]$ is in the same order as $\mathbb{E}[\|\nabla\boldsymbol{c}(\boldsymbol{x}^{R+1})\boldsymbol{c}(\boldsymbol{x}^{R+1})\|^2]$.

We are now ready to prove Theorem 3.2.

### C.5.1. PROOF OF THEOREM 3.2

*Proof.* Consider the parameter choices in (C.56). When $\|\boldsymbol{c}(\boldsymbol{x}^0)\|^2 = \mathcal{O}(K^{-l})$ and the initial batch size is chosen as $b_0 = \mathcal{O}(K^l)$, we have $\mathcal{L}_{\rho,\mu}(\boldsymbol{w}^0) = \mathcal{O}(1)$ and $\mathbb{E}[\|\boldsymbol{e}^0\|^2] = \mathcal{O}(K^{-l})$. Combining it with (C.57), we observe that the two bounds reduce to

$$
\begin{cases}
\max\left\{\mathbb{E}[\|\boldsymbol{u}^{R+1}\|^2], \mathbb{E}[\|\boldsymbol{x}^{R+1} - \mathrm{prox}_{\mu g}(\boldsymbol{z}^R)\|^2]\right\} = \mathcal{O}\big(\max\{K^{l-1}, K^{\tau-l-1}, K^{-\tau}\}\big), \\
\mathbb{E}[\|\nabla\boldsymbol{c}(\boldsymbol{x}^{R+1})\boldsymbol{c}(\boldsymbol{x}^{R+1})\|^2] = \mathcal{O}\big(\max\{K^{-l-1}, K^{-\tau}\}\big).
\end{cases}
\tag{C.58}
$$

To achieve the optimal rate $\mathcal{O}(K^{-2/3})$, we set $\tau = 2l = \frac{2}{3}$. Hence, to obtain a stochastic $\varepsilon$-stationary point, $K$ should be of order $\mathcal{O}(\varepsilon^{-3})$. Noting that each iteration calls two stochastic gradients $\nabla\mathbf{f}(\boldsymbol{x}^k, \xi^k)$ and $\nabla\mathbf{f}(\boldsymbol{x}^{k-1}, \xi^k)$, and the initial batch requires $b_0 = \mathcal{O}(K^{1/3}) = \mathcal{O}(\varepsilon^{-1})$ gradient evaluations, the total oracle complexity is

$$b_0 + 2K = \mathcal{O}(\varepsilon^{-1}) + 2\mathcal{O}(\varepsilon^{-3}) = \mathcal{O}(\varepsilon^{-3}).$$

Under Assumption 2.3, the established bound on $\mathbb{E}[\|\boldsymbol{c}(\boldsymbol{x}^{R+1})\|^2]$ ensures that the oracle complexity to find a stochastic $\varepsilon$-KKT point is also $\mathcal{O}(\varepsilon^{-3})$. $\square$

C.5.2. PROOF OF COROLLARY 3.2

Without assuming initial approximate feasibility, we have $\mathcal{L}_{\rho,\mu}(\boldsymbol{w}^0) + 1 = \mathcal{O}(K^l)$. With a constant initial batch size $b_0 = \mathcal{O}(1)$, Assumption 2.2 gives $\mathbb{E}[\|\boldsymbol{e}^0\|^2] = \mathcal{O}(1)$. Invoking Lemma C.12 and combining it with (C.57), we observe that the two bounds admit

$$\mathcal{O}\big(\max\{K^{2l-1}, K^{\tau-1}, K^{-\tau}\}\big) \qquad \text{and} \qquad \mathcal{O}\big(\max\{K^{-1}, K^{\tau-1}, K^{-\tau}\}\big).$$

If we choose $\tau = 2l = \frac{1}{2}$, both reduce to $\mathcal{O}(K^{-1/2})$. Hence, to obtain a stochastic $\varepsilon$-stationary point, $K$ should be of order $\mathcal{O}(\varepsilon^{-4})$ and the total oracle complexity to find an $\varepsilon$-stationary point is of order $\mathcal{O}(\varepsilon^{-4})$. Under Assumption 2.3, the same complexity holds for finding a stochastic $\varepsilon$-KKT point, establishing the result in Corollary 3.2.

# D. Experimental Results

## D.1. Implementation Details

All experiments were conducted in MATLAB R2018b on a MacBook Pro (4-core processor, 16 GB RAM) under macOS 15.3.2.

**Baselines**. We compare our algorithms with two double-loop baselines: SPDC (Xu et al., 2019; Nitanda & Suzuki, 2017) for simple convex-constrained DC(-regularized) optimization and SALM (Sun & Sun, 2023) for linearly constrained DC-regularized optimization:

(i) **SPDC (Xu et al., 2019; Nitanda & Suzuki, 2017):** This is a double-loop algorithm designed for solving DC problems by linearizing the concave component and adding a quadratic proximal term to construct a strongly convex subproblem at each outer iteration $k$. To handle the constraint $\|\mathbf{x}\|_2^2 = 1$, we apply the same quadratic penalty approach for nonconvex constraints in our comparison. To ensure strong convexity of the subproblem, we adopt an appropriate proximal parameter. The inner subproblem is solved iteratively using the stochastic subgradient descent method (SPG) described in (Xu et al., 2019).

(ii) **SALM (Sun & Sun, 2023):** This double-loop method is designed for deterministic, linearly constrained composite DC optimization. The method handles constraints by constructing the Augmented Lagrangian Method (ALM), linearizing the AL function at each outer iteration, and solving the inner loop using a proximal gradient method. We adopt a similar strategy and adapt it to nonlinear constraints by applying the same linearization approach. Specifically, the concave part is linearized at $\boldsymbol{x}^k$, and the inner solver is used to update the solution. For the stochastic part, we use the same gradient estimators as in our algorithm.

**Hyperparameter Settings**. We initialize all algorithms from a feasible point, generated by normalizing a random vector. Given the data scale $N$, we use a batch size of 32 for the a9a and phishing datasets, and a batch size of 16 for the australian dataset. The momentum parameter $\alpha$ is set to 0.905 for Polyak momentum and 0.9 for recursive momentum across all baselines. The maximum number of iterations is set to $K = 25,000$ for all experiments. For MoSSP-P, the smoothing and penalty parameters are set according to the theoretical rates $\mu_k = \mathcal{O}(K^{-1/2})$ and $\rho_k = \mathcal{O}(K^{1/4})$; for MoSSP-R, they are set as $\mu_k = \mathcal{O}(K^{-1/3})$ and $\rho_k = \mathcal{O}(K^{1/3})$, with the initial batch size chosen according to $b_0 = \mathcal{O}(K^{1/3})$. For SPDC and SALM, we fix $\mu_k$ as a constant independent of $K$, tuned from $\{0.05, 0.2, 0.5, 1\}$. For SALM, $\rho$ is tuned from $\{0.01, 0.1, 0.5, 1\}$, while for SPDC, it is tuned from $\{1, 5, 10, 20\}$. The step size $\beta$ in both MoSSP variants is set to $\beta = 1$. For SALM, the dual update step size is set equal to the penalty parameter $\rho$, while the step sizes for the SPDC subgradient update and the inner-loop update of SPDC are tuned from $\{0.001, 0.01, 0.05, 0.1\}$. The regularization parameter $\lambda$ is validated over the set $\{0.005, 0.05, 0.1\}$, and the optimal value is selected. The number of inner iterations is chosen from the range $5 - 10$.

We perform five independent runs for each dataset and algorithm combination. For fair time comparison, we first run MoSSP-P (Algorithm 1) for $K = 25,000$ iterations and record its total CPU time; all Polyak-momentum baselines are run under the same CPU-time budget, and their trajectories are plotted against the cumulative number of stochastic gradients. The same procedure applies to MoSSP-R (Algorithm 2) and recursive momentum-based baselines. Figures 2 and 3 show convergence trajectories on the phishing and australian datasets (averaged over five runs).

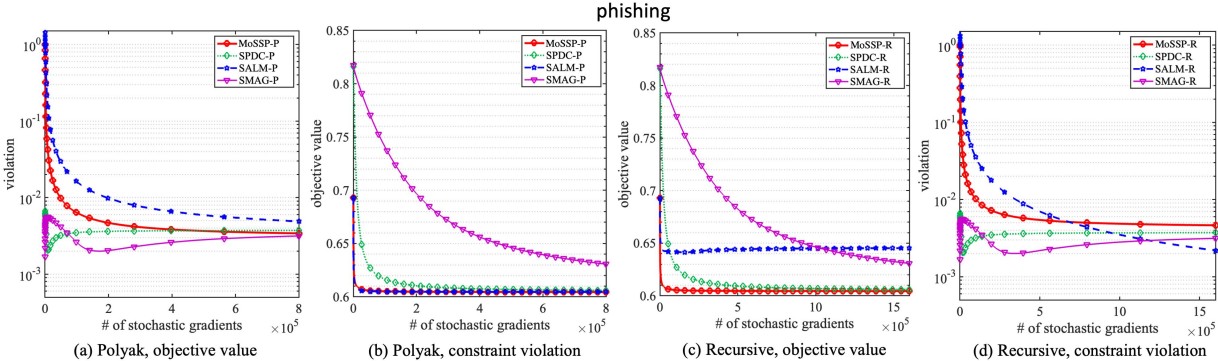

*Figure 2.* Comparison of MoSSP variants, SPDC, and SALM for solving the constrained binary classification problem (4.1) on the phishing dataset. (a) Objective value (Polyak). (b) Constraint violation (Polyak). (c) Objective value (Recursive). (d) Constraint violation (Recursive). Results are averaged over five independent runs.

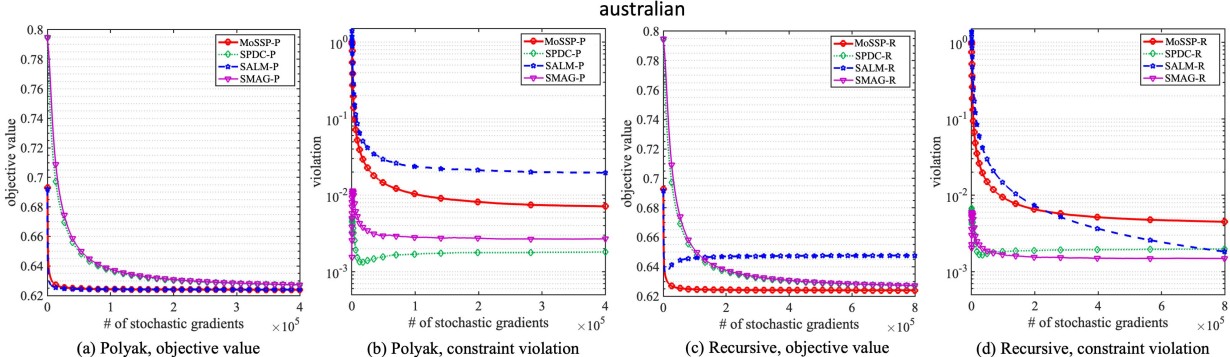

*Figure 3.* Comparison of MoSSP variants, SPDC, and SALM for solving the constrained binary classification problem (4.1) on the australian dataset. (a) Objective value (Polyak). (b) Constraint violation (Polyak). (c) Objective value (Recursive). (d) Constraint violation (Recursive). Results are averaged over five independent runs.

### D.2. Additional Experimental Results

As shown in Figure 2, on the large-scale dataset phishing, both MoSSP variants converge faster in objective value and attain the lowest or highly competitive final results. Notably, the MoSSP variants rapidly reduce constraint violations from an initial level of $10^0$ to approximately $10^{-3}$, demonstrating effective feasibility control. In contrast, SPDC and SALM are slower at achieving feasibility, with violations remaining around $10^{-2}$ or higher throughout most of the optimization.

On the smaller-scale australian dataset (Figure 3), the relative performance shifts. While the MoSSP variants still achieve faster objective descent, SPDC attains superior feasibility with violations stabilizing around the $10^{-3}$ level, outperforming the MoSSP variants in constraint satisfaction. This suggests that the simpler landscape and lower stochastic noise enable subgradient-based constraint handling to be more effective, whereas the MoSSP design prioritizes rapid objective reduction over strict feasibility in such settings. This scale-dependent behavior warrants further investigation.

### D.3. Experimental Results on Multiple Quadratic Equality Constraints

We test the proposed method on a quadratically constrained DC-regularized logistic regression problem (Jin & Wang, 2022; Shi et al., 2026):

$$\min_{\boldsymbol{x} \in \mathbb{R}^n} \quad \frac{1}{N} \sum_{i=1}^{N} \log\big(1 + \exp(-y_i X_i^\top \boldsymbol{x})\big) + \lambda\big(\|\boldsymbol{x}\|_1 - \|\boldsymbol{x}\|_2\big)$$

$$\text{s.t.} \quad c_j(\boldsymbol{x}) := \frac{1}{2} \sum_{\ell=1}^{n} q_{j,\ell}\, x_\ell^2 + \boldsymbol{a}_j^\top \boldsymbol{x} - b_j = 0, \quad j = 1, \dots, M. \tag{D.1}$$

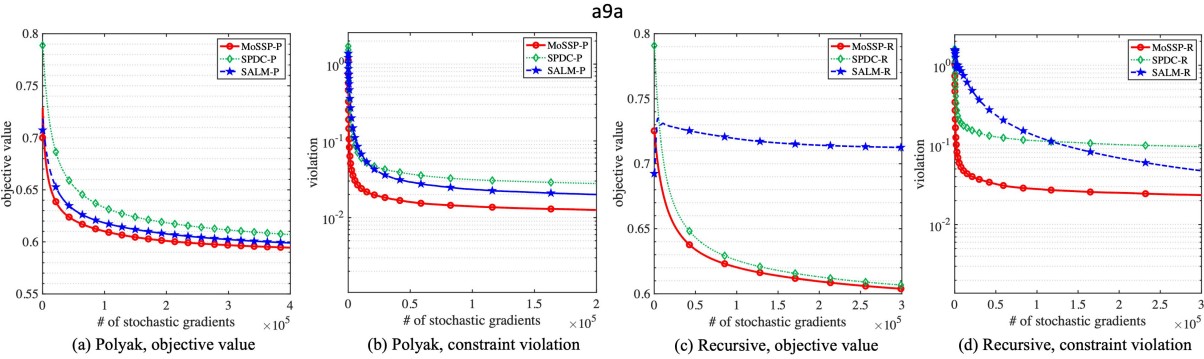

*Figure 4.* Comparison of MoSSP variants, SPDC, and SALM for solving the problem with multiple quadratic equality constraints (D.1) on the `a9a` dataset. (a) Objective value (Polyak). (b) Constraint violation (Polyak). (c) Objective value (Recursive). (d) Constraint violation (Recursive). Results are averaged over five independent runs.

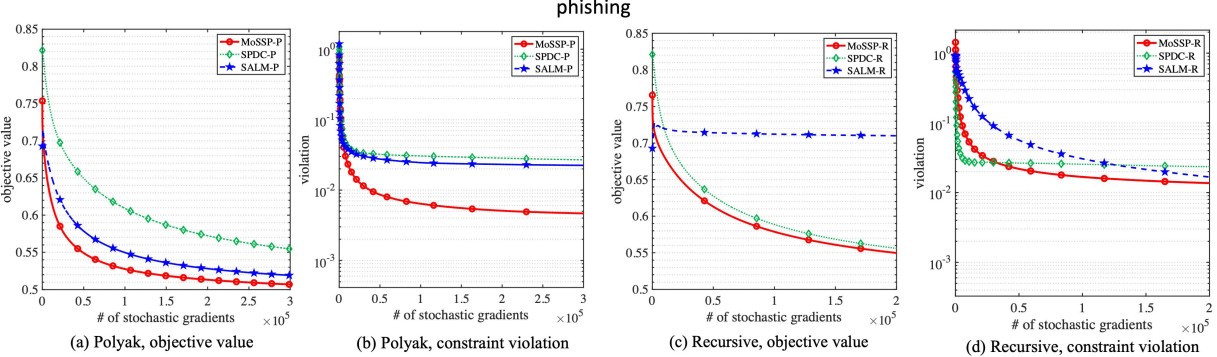

*Figure 5.* Comparison of MoSSP variants, SPDC, and SALM for solving the problem with multiple quadratic equality constraints (D.1) on the `phishing` dataset. (a) Objective value (Polyak). (b) Constraint violation (Polyak). (c) Objective value (Recursive). (d) Constraint violation (Recursive). Results are averaged over five independent runs.

Each constraint can be equivalently written as $c_j(\boldsymbol{x}) = \frac{1}{2}\boldsymbol{x}^\top Q_j \boldsymbol{x} + \boldsymbol{a}_j^\top \boldsymbol{x} - b_j$, where $Q_j = \mathrm{Diag}(q_{j,1}, \ldots, q_{j,n})$. Since all diagonal entries $q_{j,\ell}$ are positive, each $Q_j$ is positive definite. We test on two LIBSVM datasets used in the main experiments: `a9a` with $N = 32{,}561$ and $n = 123$, and `phishing` with $N = 11{,}055$ and $n = 68$. Each sample vector is normalized to unit $\ell_2$ norm. We set $M = 20$. For each constraint $j$, the coefficients $q_{j,\ell}$ are sampled uniformly from $[0.5, 1]/n$, and $\boldsymbol{a}_j \sim \mathcal{N}(\boldsymbol{0}, \boldsymbol{I}/n)$. A random unit vector $\boldsymbol{x}_\star$ is generated first, and $b_j$ is set to

$$b_j = \tfrac{1}{2} \sum_{\ell=1}^{n} q_{j,\ell} x_{\star,\ell}^2 + \boldsymbol{a}_j^\top \boldsymbol{x}_\star,$$

so that $\boldsymbol{x}_\star$ is feasible for all constraints.

**Hyperparameter Settings**. Given the data scale $N$, we use a batch size of 32 for both `a9a` and `phishing` datasets. The momentum parameter is set to $\alpha = 0.905$ for Polyak momentum and $\alpha = 0.9$ for recursive momentum across all methods. The maximum number of iterations is set to $K = 20{,}000$, and the regularization parameter is fixed as $\lambda = 0.01$. For MoSSP-P, we use $\beta = 1$, $\mu_k = \mathcal{O}(K^{-1/2})$, and $\rho_k = \mathcal{O}(K^{1/4})$; for MoSSP-R, we use $\beta = 1$, $\mu_k = \mathcal{O}(K^{-1/3})$, and $\rho_k = \mathcal{O}(K^{1/3})$, with the initial batch size chosen according to $b_0 = \mathcal{O}(K^{1/3})$. For SPDC and SALM, $\mu_k$ is tuned from $\{0.05, 0.2, 0.5, 1\}$; $\rho_k$ is tuned from $\{0.01, 0.1, 0.5, 1\}$ for SALM and from $\{1, 5, 10, 20\}$ for SPDC; the step sizes for the SPDC subgradient update and the inner-loop update are tuned from $\{0.001, 0.01, 0.05, 0.1\}$. For SALM, the dual update step size is set equal to $\rho_k$. The double-loop baselines use 5 inner iterations.

Each method is run five times with a shared constraint instance and initialization. For each momentum setting, all baselines are evaluated under the same computational budget as the corresponding MoSSP variant. We report the objective value and the aggregate constraint violation $\sum_{j=1}^{M} |c_j(\boldsymbol{x})|$. Figures 4 and 5 show convergence trajectories on both datasets (averaged over five runs), while Tables 3 and 4 report final values as mean $\pm$ std.

**Experimental Results**. As shown in Figures 4 and 5, the MoSSP variants achieve faster objective decrease than the baselines

*Table 3.* Mean $\pm$ std of objective value (Obj. Value) and constraint violation (Const. Viol.) for MoSSP-P and two baseline methods with Polyak momentum on the multiple quadratic equality experiment. Results are reported over five independent runs. Bold font denotes the best result.

| DATASET | METRIC | MoSSP-P | SPDC-P | SALM-P |
|---------|--------|---------|--------|--------|
| A9A | OBJ. VALUE | $\mathbf{0.5811 \pm 1.15 \times 10^{-5}}$ | $0.6017 \pm 2.40 \times 10^{-5}$ | $0.5914 \pm 2.51 \times 10^{-4}$ |
| | CONST. VIOL. | $\mathbf{9.93 \times 10^{-3} \pm 8.62 \times 10^{-5}}$ | $2.27 \times 10^{-2} \pm 5.56 \times 10^{-5}$ | $1.90 \times 10^{-2} \pm 2.46 \times 10^{-4}$ |
| PHISHING | OBJ. VALUE | $\mathbf{0.4951 \pm 9.20 \times 10^{-5}}$ | $0.5294 \pm 7.07 \times 10^{-5}$ | $0.5067 \pm 9.16 \times 10^{-5}$ |
| | CONST. VIOL. | $\mathbf{9.20 \times 10^{-3} \pm 2.14 \times 10^{-5}}$ | $2.40 \times 10^{-2} \pm 5.37 \times 10^{-5}$ | $1.22 \times 10^{-2} \pm 1.07 \times 10^{-4}$ |

*Table 4.* Mean $\pm$ std of objective value (Obj. Value) and constraint violation (Const. Viol.) for MoSSP-R and two baseline methods with recursive momentum on the multiple quadratic equality experiment. Results are reported over five independent runs. Bold font denotes the best result.

| DATASET | METRIC | MoSSP-R | SPDC-R | SALM-R |
|---------|--------|---------|--------|--------|
| A9A | OBJ. VALUE | $\mathbf{0.5911 \pm 1.11 \times 10^{-6}}$ | $0.5930 \pm 3.53 \times 10^{-5}$ | $0.7085 \pm 9.04 \times 10^{-6}$ |
| | CONST. VIOL. | $\mathbf{4.07 \times 10^{-3} \pm 3.72 \times 10^{-5}}$ | $1.79 \times 10^{-2} \pm 4.05 \times 10^{-5}$ | $1.70 \times 10^{-2} \pm 1.16 \times 10^{-4}$ |
| PHISHING | OBJ. VALUE | $\mathbf{0.5063 \pm 1.04 \times 10^{-6}}$ | $0.5085 \pm 1.27 \times 10^{-4}$ | $0.7030 \pm 3.36 \times 10^{-6}$ |
| | CONST. VIOL. | $9.93 \times 10^{-3} \pm 2.58 \times 10^{-5}$ | $1.94 \times 10^{-2} \pm 2.42 \times 10^{-5}$ | $\mathbf{3.27 \times 10^{-3} \pm 1.42 \times 10^{-5}}$ |

on both datasets. With Polyak momentum, MoSSP-P also yields lower constraint violation than SPDC-P and SALM-P. With recursive momentum, MoSSP-R maintains the fastest objective convergence and achieves competitive, often lower, constraint violation compared with SPDC-R and SALM-R. Tables 3 and 4 provide the corresponding final quantitative comparisons on the a9a and phishing datasets. Under Polyak momentum, MoSSP-P achieves both the lowest objective value and the smallest constraint violation on both datasets; in particular, it reaches an objective value of $0.5811$ and a violation of $9.93 \times 10^{-3}$ on the a9a dataset, and keeps the violation below $10^{-2}$ on the phishing dataset. With recursive momentum, MoSSP-R attains the best objective values on both datasets and the smallest constraint violation on the a9a dataset. On the phishing dataset, SALM-R achieves the smallest final violation, but at the cost of a substantially larger objective value than MoSSP-R and SPDC-R.

