# OpenReview forum: "MoSSP: A Momentum-Based Single-Loop Stochastic Penalty Method for Nonconvex Constrained DC-regularized Optimization"
_ICML.cc/2026/Conference — ICML 2026 regular_

### Official Review · Reviewer_wg6Y · 2026-03-09

**Soundness:** 3
**Presentation:** 3
**Significance:** 3
**Originality:** 2
**Overall Recommendation:** 4
**Confidence:** 3

**Summary:**

This paper studies nonconvex constrained stochastic difference-of-convex (DC) optimization problems, where the objective contains an expectation term $\mathbb{E}_\xi[\mathbf{f}(\mathbf{x},\xi)]$ and the constraints are $\mathbf{c}(\mathbf{x})=\mathbf{0}$ with $\mathbf{c}(\cdot)$ being continuously differentiable. The authors propose a Momentum-based Single-loop Stochastic Penalty (MoSSP) method that incorporates momentum into a single-loop stochastic penalty frameworkwithout solving nested inner-loop subproblems. This paper establishes oracle complexity guarantees for achieving a stochastic $\epsilon$-KKT point. The authors show that the MoSSP method achieves a complexity of  $\mathcal{O}(\epsilon^{-4})$ under standard boundedness and smoothness assumptions (Assumption 2.1-2.2), and improves to $\mathcal{O}(\epsilon^{-3})$ under an additional mean-squared smoothness assumption (Assumption 3.1). These complexities match the best-known lower bound for unconstrained stochastic optimiation (Arjevani et al., 2023).

**Compliance With Llm Reviewing Policy:**

Affirmed.

**Final Justification:**

Thanks the response from the authors. I maintain my score.

**Key Questions For Authors:**

1. On Assumption 2.3 (global LICQ-style condition). The main $\epsilon$-KKT guarantee appears to rely crucially on Assumption 2.3, which is the global condition $||\nabla \mathbf{c}(\mathbf{x}) \mathbf{c}(\mathbf{x})||\geq\delta||\mathbf{c}(\mathbf{x})||$, $\forall\mathbf{x}\in\mathbb{R}^n$. This is much stronger than a local LICQ or metric regularity condition near the feasible set. In the proof of Theorem 3.1, this assumption is used to convert a bound on $\mathbb{E}[||\nabla \mathbf{c}(\mathbf{x}^{R+1}) \mathbf{c}(\mathbf{x}^{R+1})||^2]$ into a bound on $\mathbb{E}[||\mathbf{c}(\mathbf{x}^{R+1})||^2]$, but the proof only states that these terms are "in the same order" without discussion on the dependence on $1/\delta^2$ (page 22-23). Could the authors clarify whether a weaker local condition would suffice, and how sensitive the final complexity guarantee is to the magnitude of $\delta$?

2. On the penalty parameter $\rho_k$. The theory chooses an increasing penalty parameter of the form $\rho_k\equiv\rho=\mathcal{O}(K^l)$ (Lemma 3.1), and in Theorem 3.1 specifically $\rho_k\equiv\rho_0 K^{1/4}$. Since Assumption 2.3 already imposes a strong global LICQ / nonsingularity-type condition, could the authors clarify why such an increasing penalty parameter is still necessary? In particular, is it possible to use an exact penalty formulation under this assumption, or is the increase of $\rho$ essential for the current proof technique?

**Limitations:**

yes.

**Strengths And Weaknesses:**

Strengths:

The paper studies nonconvex constrained stochastic DC problems with functional equality constraints. This problem class is relevant in machine learning settings where nonconvex regularizers or composite structures arise together with deterministic constraints. The proposed MoSSP framework provides a single-loop stochastic algorithm that avoids inner-loop subproblems commonly used in DC methods. This leads to a relatively simple algorithmic structure compared with double-loop methods.

The paper establishes oracle complexity guarantees for computing stochastic $\epsilon$-KKT points that match the best-known lower bounds for unconstrained stochastic optimization.

Weaknesses:

The theoretical guarantees rely on a relatively strong global regularity condition (Assumption 2.3), which assumes a global LICQ condition. Thus, the feasibility guarantees may degrade when $\delta$ is small, and the practical implications of this assumption are not clearly discussed.

The algorithm introduces momentum techniques within the single-loop stochastic penalty framework. However, the motivation for introducing momentum techniques in the constrained DC setting is not clearly articulated in the main body of the paper. In particular, it remains unclear why momentum stabilization is necessary to handle the constraint $\mathbf{c}(\mathbf{x})=\mathbf{0}$ or the penalty gradient $\nabla\mathbf{c}(\mathbf{x})\mathbf{c}(\mathbf{x})$, since the closely related single-loop stochastic DC algorithms based on Moreau-envelope smoothing for unconstrained DC problems (Hu et al., 2024) achieve similar complexity guarantees without momentum. The paper would benefit from a more clear explanation of whether momentum plays an essential algorithmic role for constrained problems or mainly serves as a practical stabilization device.

---

> ### Author Rebuttal · Authors · 2026-03-31
>
> We sincerely appreciate Reviewer wg6Y's positive assessments and insightful comments. Our point-by-point responses follow below.
> # [W2] On the motivation clarification of momentum in the constrained setting.
> We appreciate this insightful question, which allows us to distinguish our momentum-based algorithms from existing unconstrained methods. We emphasize that, under our penalty framework, momentum (Mom.) brings a genuine **theoretical complexity improvement** in the constrained setting (see Table 1).
>
> **Table 1: Comparison with unconstrained stochastic DC methods**
>
> | Set. | Method | $L$ | Mom. | Comp. |
> |---|---|---|---|---|
> | Unc. | Hu et al. (2024) | $O(1)$ | -- | $O(\epsilon^{-4})$ |
> | Con. | Jin and Wang [1] | $O(\epsilon^{-1})$ | -- | $\underline{O(\epsilon^{-5})}$ |
> | Con. | **MoSSP-P (Alg. 1)** | $O(\epsilon^{-1})$ | $\checkmark$ | $\underline{O(\epsilon^{-4})}$ |
> | Con. | **MoSSP-R (Alg. 2)** | $O(\epsilon^{-1})$ | $\checkmark$ | $\underline{O(\epsilon^{-3})}$ |
>
> Specifically, in the constrained setting, the penalty parameter $\rho$ induces an effective smoothness constant $L_\rho = \Theta(\rho)$, and the algorithm must simultaneously control both criticality (Crit.) and feasibility (Feas.). Since feasibility requires $\rho \to \infty$, $L_\rho$ grows unboundedly. Without momentum, one must adopt a smaller stepsize $\mu$ or a larger batch size to compensate, yielding oracle complexity **no better than** existing penalty-based stochastic algorithms [1]. With momentum $\alpha$, the criticality and feasibility residuals can be balanced via careful tuning of $\rho$, $\mu$, and $\alpha$ under $O(1)$ batch size, i.e.,
>
> $$ \text{Crit.}\lesssim \frac{\rho L\Delta}{\mu K}+\frac{\sigma^2\alpha}{\mu}, \quad \text{Feas.}\lesssim \frac{L\Delta}{\rho\mu K}+\frac{\sigma^2\alpha^2}{\rho^2\mu}+\frac{1}{\rho^2}$$
>
> *(Above: MoSSP-P)*, which enables the improved complexity results. We will revise the paper to make this important point clearer.
> # [W1&Q1] Assumption 2.3 and small $\delta$-dependence?
> We appreciate the constructive questions.
>
> **(1) A local condition can suffice.**  We clarify that our results only require the CQ along the iterate sequence and will revise the conditions used.
>
> **(2) Explicit $\delta$-dependence and practical implications.** While making $\delta$ explicit highlights the problem-structure-awareness of the bounds, this $\delta^{-2}$ dependence is theoretically unavoidable under our current analysis framework: comparable penalty-based analyses under similar CQ assumptions [1, 2] carry the same implicit dependence.
>
> Regarding practical impact, $\delta$ may remain a constant $\gg \varepsilon$ by algorithmic design. For $c(x) = \lVert x\rVert^2 - 1$ in our experiments, the penalty update drives iterates away from $x = 0$, the only point where $\delta$ could potentially become small, ensuring $\delta \geq O(1) \gg \varepsilon$. This guarantees small constraint violation at convergence, consistent with our numerical results. Moreover, for the CQ–complexity tradeoff, we find that [3] provides a quantitative characterization via the error bound $\lVert \nabla c(x) c(x) \rVert \geq \delta \lVert c(x) \rVert^\theta$ with $\theta \geq 1$. We believe that extending our analysis to this generalized case is a valuable future direction. We will add a corresponding discussion to the revision.
> # [Q2]  Why use increasing penalty parameter $\rho_k$?
> We clarify two key aspects of this insightful question below.
>
> **(1) The increasing schedule for $\rho_k$ is a necessary design choice to maintain feasibility**, which is inherent to quadratic penalty methods, including ALM for nonlinear constraints [1–3]. By increasing $\rho_k$, we aim to reduce the stationarity residual $\mathbb{E}[\lVert\nabla c(x) c(x)\rVert]$, and together with the CQ, to further guarantee constraint violation $\mathbb{E}[\lVert c(x)\rVert] \leq \varepsilon$.
>
> **(2) Simple implementation motivated our choice of quadratic penalty.** While exact penalty methods are theoretically attractive, the penalty threshold is typically unknown a priori, and the nonsmoothness of standard exact penalty functions (e.g., $\ell_1$) introduces additional practical difficulties. The quadratic penalty keeps our single-loop framework clean and practical, which is important for handling the nonsmooth DC term. We appreciate this constructive comment, which helps us identify important future directions. In particular, we may consider penalty methods with adaptive penalty update that can ensure exactness.
>
> ***References***
>
> [1] L. Jin and X. Wang, A stochastic primal-dual method for a class of nonconvex constrained optimization. COAP, 2022.
>
> [2] A. Alacaoglu and S. J. Wright, Complexity of single-loop algorithms for nonlinear programming with stochastic objective and constraints, AISTATS, 2024.
>
> [3] Z. Lu et al., Variance-reduced first-order methods for deterministically constrained stochastic nonconvex optimization with strong convergence guarantees, SIOPT, 2026.

---

> > ### Author Rebuttal · Reviewer_wg6Y · 2026-04-02
> >
> > Thanks for the response.

---

> > > ### Author Response · Authors · 2026-04-03
> > >
> > > Dear Reviewer wg6Y,
> > >
> > > We sincerely appreciate your response and updated assessment. We would be happy to address any follow-up questions and provide further clarification if helpful. Thank you again for your time and support.
> > >
> > > Best regards,
> > >
> > > The Authors

---

### Official Review · Reviewer_Ujzb · 2026-03-11

**Soundness:** 3
**Presentation:** 2
**Significance:** 2
**Originality:** 3
**Overall Recommendation:** 5
**Confidence:** 3

**Summary:**

The paper studies a class of nonconvex constrained optimization problems with DC regularizers and proposes a single-loop penalty method that uses the momentum of the smooth part. The paper uses both polyak momentum and momentum variance reduction and proves convergence rates matching known lower bounds.

**Compliance With Llm Reviewing Policy:**

Affirmed.

**Final Justification:**

The rebuttal convinced me of the merits of the paper. I recommend accepting the paper as I think it has nice ideas and techniques that will be of help to the community.

**Key Questions For Authors:**

1 - What are the challenges in dealing with inequality constraints in your approach compared to equality constraints? In other words, why limit the work to equality constraints?

2 - You already approximate the prox of $\psi_\rho$ why not do the same to g? This should be similar to what is done in Hu et al and should make your approach more general.

3 - Can you explain lines 321-322? What do you mean by "approximate feasible initialization"?

**Limitations:**

Some limitations were not discussed: h and g have to be prox-friendly, also f and c need to be both smooth and Lipschitz, which is strong. Also, only equality constraints are considered.

**Strengths And Weaknesses:**

Soundness: The results of the paper are sound, and the proofs look correct.

Presentation: The writing is overall clear, but some some paragraphs and statements are a bit misleading: the comparison with DC (line 43 to the right), the text is written in a way that implies the problem (P) covers the whole class of DC problems, which is not true (or at least, not trivial if it were to be true, which I doubt). The problem P has a special structure that prevents general concave parts (the concave part g is assumed to be prox-friendly, which is restrictive ). Also, the approach of using the difference of the Moreau envelope is not new and should be clearly credited to Hu et al. (2024). I also think the work of chayti et al. (2025) merits more comparison, as it also uses momentum in the context of general DC optimization problems. Note also that equations 3.4 and 3.8 have a closed form in terms of the prox of h, which is assumed to be simple.

Significance: Dealing with equality constraints and DC regularizers is interesting, although still somewhat restrictive (general constraints, g that is not necessarily prox-friendly), I don't know what challenges inequality constraints pose that the authors avoided them, if they are more complex, then this would be a first step towards generalizing to them. Another problem is the strong assumptions on f and c that are assumed to be smooth and Lipschitz, which is strong.

Originality: Applying the idea of the difference of Moreau smoothing from Hu et al. (2024) on top of the penalty method is a nice idea, it is original to the best of my knowledge.

---

> ### Author Rebuttal · Authors · 2026-03-30
>
> We sincerely thank Reviewer Ujzb for the thorough and constructive suggestions.  We are greatly encouraged by the characterization of our framework as "a nice idea", as well as the recognition of the meaningfulness of our problem. Below, we address each concern point by point.
> # [W1&Q2] Scope of Problem (P)? Why not also approximate $prox_g$?
> We greatly appreciate the valuable comments, which allow us to clarify the positioning of our work and elaborate on our treatment of the $g$-subproblem.
>
> (1) While Problem (P) is not intended to cover all DC programs, we focus on a structured subclass with nonsmooth regularizers, where $prox_h$ and $prox_g$ are typically available but $prox_{h-g}$ is generally not. We will refine the statement about "the general problem" in the revision.
>
> (2) While our current framework leverages closed-form $prox_g$, the analysis can indeed accommodate more general cases, such as $-\mathbb{E}\_{\xi}[g(x;\xi)]$, which arise in some applications, including PU learning and partial AUC. Specifically, our work focuses on the setting where $g$ is proximable so that the algorithmic and analytical efforts can be devoted to handling the nonconvex constraints. However, we fully agree that the general $g$ case is both important and timely, and approximating $prox_g$ as in Hu et al. (2024) offers an efficient and implementable solution. We thank the reviewer for helping us uncover the practical extension potential of our framework.
> # [W2(a)] Relation to prior unconstrained stochastic DC methods.
> We thank the reviewer for this constructive suggestion, which allows us to highlight our contribution better.
>
> Hu et al. (2024) is a fundamental work and, to our knowledge, the first single-loop method for stochastic difference-of-weakly-convex(DWC) optimization via the DME idea. Our contribution lies in showing that, in the **nonconvex constrained** setting, the DME idea can be incorporated into a penalty framework. This is done while retaining the optimal oracle complexity in our manuscript.
>
> Chayti & Jaggi (2025) is a solid work and, to our knowledge, the first to introduce momentum into stochastic DC optimization. Let us summarize **the core differences** below.
>
> **(i) Constraints:** Their work studies unconstrained DC optimization, whereas we consider the nonconvex constrained setting.
>
> **(ii) Problem structure:** Their framework focuses on smooth DC objectives, whereas ours also allows a nonsmooth concave part $-g$, which is motivated by many constrained regularized models where $prox_g$ is simple.
>
> **(iii) Role of momentum:** In our framework, momentum yields a genuine oracle complexity improvement in the stochastic setting from $O(\varepsilon^{-5})$ to $O(\varepsilon^{-3})$, and plays a theoretically distinct role from the unconstrained setting. Please see our response to Reviewer wg6Y [W2] for a more detailed discussion.
>
> We will revise the main body to reflect these clarifications and comparisons.
> # [W2(b)] Closed forms of (3.4) and (3.8).
> We will add the explicit form after each equation.
> # [W3] Strong assumptions on $f$ and $c$.
> We appreciate this valuable comment. We clarify that Assump. 2.1 is stated globally for brevity in our paper and is standard in constrained stochastic optimization. In fact, we **only require the boundedness along the iterate sequence**. A detailed discussion on this is provided in **our response to Reviewer Vqbz [Q1] (1) and (2)**. Extending to more general conditions, such as Hölder smoothness, is an interesting future direction that we believe our framework has the potential to accommodate.
>
> # [Q1] Why limit the work to equality constraints?
> Thanks for this constructive question. Equality constraints $c_E(x)$ are adopted for brevity, as inequality constraints $c_I(x) \le 0$ can be handled by replacing $Q_\rho(x)$ in (3.1), yielding
>
> $$Q_\rho(x) = f(x) + \frac{\rho}{2}\Bigl(\lVert c_E(x)\rVert^2 + \lVert [c_I(x)]_+\rVert^2\Bigr),$$
>
> where the smoothness conditions remain without additional assumptions, and the analysis carries over directly. We will explicitly state this in the revision.
> # [Q3] Explanation of “approximate feasible initialization”.
> We thank the reviewer for the careful reading. We clarify that "approximate feasible initialization" means $x^0$ has small but not necessarily zero constraint violation, e.g., $\lVert c(x^0)\rVert^2 = O(K^{-l}) (0<l<1)$. We present Corollary 3.1 to highlight how this initialization helps improve the complexity order in the constrained setting. Specifically, increasing $\rho_K = \rho K^l$ is necessary to reduce $\mathbb{E}[\lVert\nabla c(x^k) c(x^k)\rVert]$; however, this causes the penalty term $\rho_0/2 \lVert c(x^0)\rVert^2$ to scale as $O(K^{2l})$ when $\lVert c(x^0)\rVert^2 = O(1)$,  thereby dominating the bound in Eq. (3.11). **Approximate feasible initialization cancels this growth and keeps the term bounded by $O(1)$, thereby improving the complexity order**. We will refine the statement for greater rigor.

---

> > ### Author Rebuttal · Reviewer_Ujzb · 2026-04-02
> >
> > I want to thank the authors for their answer. Please incorporate the promised changes into your work. I will increase my score to ACCEPT. Best of luck.

---

> > > ### Author Response · Authors · 2026-04-03
> > >
> > > Dear Reviewer Ujzb,
> > >
> > > We sincerely appreciate your positive feedback and are greatly encouraged by your willingness to raise the score. We will carefully incorporate the promised revisions into the final version. Thank you again for your time and support.
> > >
> > > Best regards,
> > >
> > > The Authors

---

### Official Review · Reviewer_Vqbz · 2026-03-17

**Soundness:** 3
**Presentation:** 3
**Significance:** 3
**Originality:** 3
**Overall Recommendation:** 5
**Confidence:** 3

**Summary:**

This paper studies nonconvex constrained stochastic optimization with difference-of-convex (DC) regularization, where the constraints may themselves be nonconvex and the concave component of the DC term may be nonsmooth.
The authors propose MoSSP, a single-loop penalty framework that employs Moreau-envelope smoothing of each DC component.
They develop two variants: MoSSP-P, which uses Polyak momentum and attains an oracle complexity of $O(\epsilon^{-4})$, and MoSSP-R, which uses recursive momentum and attains an oracle complexity of $O(\epsilon^{-3})$, both for finding stochastic $\epsilon$-KKT points.

**Compliance With Llm Reviewing Policy:**

Affirmed.

**Final Justification:**

The rebuttal addressed my concern, clarify the weakness and improvement so far.

**Key Questions For Authors:**

Are Assumptions 2.1 and 2.3 too strong for the intended applications?
In particular, Assumption 2.1 requires $c(x) \le C$ for all $x$, which does not appear to be satisfied by the running example.

Have you considered experiments with constraints where projection is genuinely difficult, such as multiple nonlinear equality constraints? This would better motivate the penalty-based approach.

**Limitations:**

IMO, a limitation of the proposed algorithm is its reliance on the assumption that the initial point is nearly feasible in order to achieve the optimal rate.
This requirement, though understandable, could be articulated more explicitly.

**Strengths And Weaknesses:**

## Strenghts

The paper addresses an important problem, as single-loop stochastic methods for DC optimization under general nonconvex constraints were previously unavailable.
The combination of DC structure, nonconvex constraints, and stochastic gradients within a single-loop framework is a novel contribution.
The $O(\epsilon^{-3})$ rate for the recursive variant matches the best-known lower bound for unconstrained stochastic optimization.
The paper is generally well written, with a clear comparison to prior work (Table 1) and clean complexity analysis.


Weakness:
The experimental evaluation could be strengthened.
The constraint function used in the experiments is so simple that it seems unjustified to rule out projection-based methods.
For example, one could apply a proximal linearized DC algorithm combined with projection onto the Euclidean ball, which should have a complexity comparable to projected gradient methods and likely performs substantially better than the infeasible method here.

---

> ### Author Rebuttal · Authors · 2026-03-30
>
> We sincerely thank Reviewer Vqbz for the thorough and constructive suggestions. We are greatly encouraged by the characterization of our framework as "a novel contribution" as well as the recognition of the importance of our problem. We also thank you for your kind remarks on the clarity and organization of the paper. Below, we address each concern point by point.
> # [W1&Q2] Have you considered experiments with constraints where projection is genuinely difficult, such as multiple nonlinear equality constraints?
> We appreciate this valuable suggestion, which helps better highlight the motivation for our penalty-based approach. We have conducted additional experiments on Problem (4.1) in our paper with **$M=20$ quadratic equality constraints** of the form $g_j(x) = \tfrac{1}{2}x^\top Q_j x + a_j^\top x - b_j = 0$, where $Q_j$ is positive definite, and the exact projection onto the
> feasible set is **intractable**. The results on two LIBSVM datasets (*a9a* and *phishing*) demonstrate that both **MoSSP-P** and **MoSSP-R** consistently achieve **lower objective values and smaller constraint violations** than the competing methods. We will strengthen additional experiments with detailed settings in the revision. For the reviewer's convenience, training and test results are provided in
> [Fig. 1 (*a9a*)](https://anonymous.4open.science/r/anonymous279/a9a.png) and
> [Fig. 2 (*phishing*)](https://anonymous.4open.science/r/anonymous279/phishing.png).
> # [Q1] Are Assumptions 2.1 and 2.3 too strong for intended applications?
> We appreciate the valuable question regarding the theoretical assumptions, which allows us to clarify their roles and their practical reasonableness.
>
> ### (1) Boundedness.
> Assump. 2.1 is stated globally for brevity in our manuscript and is standard in constrained stochastic optimization [1–4]; as a matter of fact, our analysis **only requires the boundedness along the iterate sequence**, which is generally not hard to satisfy given the structure of the constraints and objective functions.
>
> The boundedness, $\lVert c(x^k) \rVert \leq C, \forall k \ge 1$, holds naturally when the feasible set is compact or when the objective is coercive. This ensures that the iterates remain bounded throughout, supporting this local condition. Such constraints commonly arise in intended applications with low-rank structure, such as $\lVert x\rVert_2^2 = 1$ or $W^\top W = I_r$ in nonnegative PCA or low-rank nonnegative matrix factorization.
>
> For $\nabla f$ and $\nabla c$, boundedness typically holds automatically on compact domains if the functions are continuously differentiable. In practice, common techniques such as bounded initialization, gradient clipping, and weight decay make these assumptions reasonable.
>
> ### (2) Smoothness.
> Our current analysis assumes smoothness; in practice, this assumption can cover several practical models, such as logistic regression and neural networks with smooth activation functions (e.g., ELU and $\tanh$). Extending to more general conditions, such as Hölder smoothness, is an interesting future direction that we believe our framework has the potential to accommodate.
>
> ### (3) Assumption 2.3 (CQ).
> We emphasize that this CQ condition only needs to hold along the iterate sequence [1-4]. Degeneracy occurs only when active constraint gradients become linearly dependent, which is relatively uncommon in structured or low-rank problems. For $\lVert x\rVert_2^2 = 1$ in our running examples, the only degenerate point is $x=0$, but our penalty mechanism keeps iterates away from zero, ensuring CQ holds at feasible points. Similarly, for QCQP-type constraints in signal processing (as additional experiments above), the gradients are derived from distinct quadratic forms and are not redundant, ensuring CQ holds at feasible points.
>
> # [L] Approximate feasible initialization could be articulated more explicitly.
> Thanks for this constructive suggestion, and we will elaborate on this point more explicitly in the revision. In fact, to remove the requirement of approximate feasible initialization, we can provide a two-phase algorithm [2]. The first phase is used to generate an approximate feasible point to initialize the second phase. Moreover, our analysis can be extended to settings without approximate feasible initialization by adopting an iteration-indexed penalty update [4].
>
> ***References***
>
> [1] Z. Lu et al., Variance-reduced first-order methods for deterministically constrained stochastic nonconvex optimization with strong convergence guarantees, SIOPT, 2026.
>
> [2] Y. Cui et al., A two-phase stochastic momentum-based algorithm for nonconvex expectation-constrained optimization. JSC, 2025.
>
> [3] F. E. Curtis et al., Worst-case complexity of an SQP method for nonlinear equality constrained stochastic optimization, MP, 2024.
>
> [4] A. Alacaoglu and S. J. Wright, Complexity of single-loop algorithms for nonlinear programming with stochastic objective and constraints, AISTATS, 2024.

---

> > ### Author Rebuttal · Reviewer_Vqbz · 2026-04-03
> >
> > I thank the authors for detailed feedback in both theory and strong experiment results. I will gladly raise the score.

---

> > > ### Author Response · Authors · 2026-04-03
> > >
> > > Dear Reviewer Vqbz,
> > >
> > > We sincerely appreciate your positive feedback and are greatly encouraged by your willingness to raise the score. We will carefully incorporate the relevant revisions into the final version. Thank you again for your time and support.
> > >
> > > Best regards,
> > >
> > > The Authors

---

### Official Review · Reviewer_o5nu · 2026-03-18

**Soundness:** 3
**Presentation:** 4
**Significance:** 3
**Originality:** 4
**Overall Recommendation:** 3
**Confidence:** 5

**Summary:**

This paper presents a Momentum-based Single-loop Stochastic Penalty (MoSSP) method, a new framework for solving stochastic optimization problem with a Difference-of-Convex (DC) objective function and non-convex constraint. It seems to me that this paper is the first work that studying DC optimization with non-convex constraint with theoretical guarantee. The proposed framework includes two variants, MoSSP-Polyak with Polyak momentum and MoSSP-Recursive using recursive momentum. The authors derive the oracle complexities for obtaining stochastic $\epsilon$-KKT points and stochastic  $\epsilon$-stationary points, with MoSSP-Polyak achieving $O(\epsilon^{-4})$ complexity and MoSSP-Recursive improving to $O(\epsilon^{-3})$ under stochastic smoothness assumption (Assumption 3.1), matching the best-known lower bounds for solving the considered problem. The authors also provide numerical experiments on an equality-constrained binary classification problem with DC regularization to verify the effectiveness of the proposed MoSSP.

**Compliance With Llm Reviewing Policy:**

Affirmed.

**Final Justification:**

Thanks for the authors' response. Since the authors have partially solved my concerns, I decided to change my overall score to 3.

**Key Questions For Authors:**

My key concern is the  practical relevance of the proposed method to the machine learning community, specifically the lack of numerical experiments. It would be better if the authors could provide more examples (as well as numerical experiments) with DC objectives with non-convex constraints.

**Limitations:**

Yes

**Strengths And Weaknesses:**

Soundness: While I have not verified all proofs in full detail, the convergence results seem plausible under the standard assumptions.  My concern regarding the numerical experiments is that the example (equation (4.1)) used in Section 4.1 does not fit the problem setting considered in this paper, as its constrained set is convex. It would be more convincing if the authors could provide numerical examples with non-convex constraints.

Presentation: The overall presentation of this paper is good to me. It is clearly written and well structured. However, the results summarized in Table 1 are obtained under different assumptions. For instance, Algorithm 2 requires Assumption 3.1, whereas the other methods do not rely on this assumption. It would be helpful if the authors could clearly state the assumptions used for each method to avoid misleading readers. Besides, the notation “unconst. = unconstrained” is redundant.

Significance: This paper addresses an important theoretical problem. My only concern is about the problem setting, DC objectives with non-convex constraints, is not practical in machine learning. I would like to see more examples.

Originality: While the tools employed in this paper, such as quadratic penalty, Moreau envelope, and momentum, are not new themselves, its originality comes from the creative integration of these techniques into a powerful new framework. In addition, the use of the Difference of Moreau Envelopes (DME) to smooth non-smooth DC function within a penalty framework is novel.

---

> ### Author Rebuttal · Authors · 2026-03-30
>
> We sincerely appreciate Reviewer o5nu's insightful comments. We are deeply grateful for your recognizing the novelty of our work. We also thank you for the kind remarks on the clarity and organization of the paper.  Below, we provide a detailed response to your feedback on practical relevance.
> # [W1&Q1] Could you provide more examples with DC objectives with non-convex constraints?
> We are glad to further demonstrate the practical potential of our work. In fact, Problem (P) captures several natural machine learning models that feature both task-motivated nonconvex constraints and DC objectives.
>
> (1) Nonconvex constraints arise whenever the feasible set encodes a structural requirement: low-rank constraints, $W^\top W = I_r$ coupled with $W \ge 0$ in matrix factorization; \$\lVert x\rVert_2^2 = 1$ coupled with $x \ge 0$ in nonnegative Sparse CCA; layer-wise sparsity and energy budget constraints in DNN training (see Example (i));  and safety-aware constraints, e.g., expected probability of failures [2] in safe reinforcement learning problem. Furthermore, it is well-known that inequality constraints can be converted to equality constraints via slack variables.
>
> (2) DC objectives arise in two canonical ways. (a) DC decomposition is an algebraic result of the minimax form in problems with a max-structured risk, such as positive-unlabeled (PU) learning [3], partial AUC optimization [4], or adversarial learning; (b) Sparsity-promoting regularizers. DC structure is deliberately introduced via structured regularizers [5], such as capped-$\ell_1$, which approximate $\ell_0$ while retaining a tractable convex-concave decomposition.
>
> Concrete examples combining both features are as follows.
>
> ***(i) DNN training under energy budgets [1].*** The DC-form capped group norm used for structured pruning, $R_{DC}(W) = \sum_g [\lVert W_g\rVert_2 - (\lVert W_g\rVert_2 - \tau)_+]$, together with hardware energy constraints, yields
>
> $\min_{W,S}\ \ell(W)+\lambda R_{DC}(W) \quad \text{s.t. }\ \phi(w_{(u)})\leq s_{(u)},\ \psi(S)\leq E_{\rm budget},\ \forall u,$
>
> where $W = \{w_{(u)}\}$ denotes the stacked layer weights and $S = \{s_{(u)}\}$ the stacked non-sparse weights of all layers.
>
> ***(ii) PU learning with error-control constraints [3].*** The DC-form unbiased risk estimator, incorporating false-positive-rate constraints (such as Neyman–Pearson type), yields
>
> $\min_{\theta}\ \pi_p \mathbb{E}\_{P_+}[\ell(\theta)] + \mathbb{E}\_{P_U}[\ell'(\theta)] - \pi_p \mathbb{E}\_{P_+}[\ell'(\theta)] \text{ s.t.}\ c_i(\theta)\leq 0,\ i=1,\dots,m.$
>
> ***(iii) Fair classification with demographic parity constraints [6].*** A concave adversarial fairness regularizer, together with a nonlinear demographic parity constraint on conditional acceptance rates, yields
>
> $\min_{\theta}\ \mathbb{E}[\ell(\theta)] - \lambda \mathbb{E}\_{x}[\log \sigma(g(\theta,x))] \text{ s.t. } \| \mathbb{E}\_{a=0}[\hat{y}] - \mathbb{E}\_{a=1}[\hat{y}] \| \leq \delta.$
>
> ***(iv) Nonnegative Sparse CCA [7].*** DC sparsity penalties combined with unit-norm and nonnegativity constraints yields
>
> $\min_{u,v} -u^\top \Sigma_{XY} v + R_{DC}(u)+R_{DC}(v) \text{ s.t. } u^\top u=1,\ v^\top v=1, u \geq 0, v \geq 0$.
>
> We have conducted additional experiments for our method on **Example (iv)** with synthetic data.
> Results in [Fig. 1(*SCCA*)](https://anonymous.4open.science/r/anonymous279/SCCA.png) show that MoSSP-P achieves rapid and stable convergence, while both variants achieve the lowest objective value with constraint violation around $10^{-2}$, confirming the practical effectiveness of our method. We will carefully include these examples as motivating applications and strengthen empirical validations in the revision.
>
> # [W2] Clearly state the assumptions used for each method in Table 1?
> We clarify that Assump. 3.1 (Algo. 2) is standard for variance reduction, also adopted by MLALM (the constrained baseline). Algo. 1 requires only the weaker expected smoothness, consistent with the unconstrained baselines in the first three rows of Table 1. We will refine accordingly in the revision.
>
> ***References***
>
> [1] H. Yang, et al., Ecc: Platform-independent energy-constrained deep neural network compression via
> a bilinear regression model, CVPR, 2019.
>
> [2] H. Bharadhwaj et al., Conservative safety critics for exploration, ICLR, 2021.
>
> [3] R. Kiryo et al., Positive-unlabeled learning with non-negative risk estimator, NeurIPS, 2017.
>
> [4] Q. Hu et al., Single-loop stochastic algorithms for difference of max-structured weakly convex functions, NeurIPS, 2024.
>
> [5] Y. Xu et al., Stochastic optimization for DC functions and nonsmooth non-convex regularizers with non-asymptotic convergence, ICML, 2019.
>
> [6] B. H. Zhang et al., Mitigating unwanted biases with adversarial learning, AAAI, 2018.
>
> [7] D. M. Witten et al., A penalized matrix decomposition, with applications to sparse principal components and canonical correlation analysis, Biostatistics, 2009.

---

> > ### Author Rebuttal · Reviewer_o5nu · 2026-04-03
> >
> > 1. It seems to me that Problem (P) only considers the equality constraints $c(x)=0$, while all four examples (i) (ii) (iii) (iv) that the authors provided have inequality constraints $h(x)\le 0$. More explanations are needed.
> >
> > 2. Since the authors claimed that $\|x\|_2^2=1$ coupled with $x\ge0$ in nonnegative Sparse CCA is non-convex constraints,  same reason for the problem (4.1) used in Section 4.1, why there is additional inequality constraints $x\ge 0$?
> >
> > 3. I don't think "Assump. 3.1 (Algo. 2) is standard for variance reduction" since Assumption 3.1 requires the smoothness for each INDIVIDUAL function $f(x,\xi)$ (NOT $f(x)$), which is not required in the baselines (the methods in the first six rows of Table 1). Given this, the theoretical contribution of this paper is weaker than original claim since it requires stronger assumptions, and thus the contribution of this paper should be revised.

---

> > > ### Author Response · Authors · 2026-04-04
> > >
> > > Dear Reviewer o5nu,
> > >
> > > We sincerely thank you for the constructive engagement and follow-up questions. We address the remaining concerns below.
> > > # [Q1] Coverage of inequality constraints in examples (i)-(iv).
> > > Thanks for the constructive question. Equality constraints $c_E(x)=0$ are adopted in Problem (P) merely for brevity.  Our framework and complexity results can be directly extended to inequality constraints $c_I(x)\leq 0$ by replacing $Q_\rho(x)$ in (3.1) with
> > > $$Q_\rho(x) = f(x) + \frac{\rho}{2}\Bigl(\lVert c_E(x)\rVert^2 + \lVert [c_I(x)]_+\rVert^2\Bigr).$$The smoothness conditions are preserved, and thus the analysis carries over without modification. Therefore, the inequality constraints appearing in examples (i)--(iv) are already covered by the same framework. We will make this extension explicit in the revision to improve clarity.
> > > # [Q2] Why does nonnegative Sparse CCA include $x \geq 0$ that is not in Problem (4.1)?
> > > Thanks for the constructive question. We clarify that the **nonconvex constraint** in Sparse CCA arises solely from **the sphere constraint** $|x|_2^2 = 1$. The additional $x \geq 0$ is convex, and is imposed purely for interpretability (nonnegative loadings allow each variable's contribution to be directly compared in magnitude without sign ambiguity), and does not introduce additional nonconvexity.
> > >
> > > We note that while the unit ball $|x|_2^2 \leq 1$ is convex, the unit sphere $|x|_2^2 = 1$ is nonconvex (e.g., $(1,0,\ldots,0) \in \mathbb{R}^d$ and $(-1,0,\ldots,0) \in \mathbb{R}^d$ are feasible, but their midpoint $\mathbf{0}$ is not feasible). We acknowledge that our original phrasing, such as "coupled with $x \geq 0$", may have obscured this distinction, and we will state the nonconvexity more explicitly in the revision.
> > >
> > > # [Q3] On the necessity of Assumption 3.1 for the improved rate.
> > > We sincerely appreciate the valuable question, which helped us state our contributions more precisely and rigorously. While we agree that the improved $\mathcal{O}(\varepsilon^{-3})$  complexity result of Algo. 2 relies on the slightly stronger Assump. 3.1, **this is precisely the structure exploited by variance reduction (VR) methods [2–4] to achieve improved complexity in nonconvex stochastic optimization.** Without Assump. 3.1, [1] establishes an $\Omega(\varepsilon^{-4})$ lower bound for any stochastic first-order method, explaining why VR methods [2–4] generally require this condition to attain lower complexity results. For nonconvex constrained stochastic optimization, proximal point methods [6] achieve $\mathcal{O}(\varepsilon^{-4})$ complexity without Assump. 3.1, under a slightly weaker Lipschitz smoothness assumption on the expected function.
> > >
> > > We have updated Table 1 to clearly indicate which algorithms use VR and which stochastic assumptions they rely on.
> > >
> > > **Table 1. Updated in the paper.**
> > > | Algo | Stoch. Assump. | V.R. |
> > > |---|---|---|
> > > | SPD, SSDC-SPG, SMAG$^*$ | L‑sm (or none) | --|
> > > | CLCDC-ALM, iMBAdc | deterministic | -- |
> > > | MLALM [5] | Assmp. 3.1 | ✓ |
> > > | **Algo. 1** | L-sm | -- |
> > > | **Algo. 2** | Assmp. 3.1 | ✓ |
> > >
> > > *Rm: L-sm denotes smoothness of the expected function; SMAG studies stochastic difference-of-weakly-convex optimization and does not require the smoothness assumption.*
> > >
> > > Specifically, in Table 1, Assump. 3.1 is only required by VR-based stochastic methods. SPD, SSDC-SPG, and SMAG use no VR and do not rely on it; CLCDC-ALM and iMBAdc are deterministic. Notably, MLALM, the constrained stochastic baseline using VR, is analyzed under the same assumption **(Assumption 3 in [5])** to achieve the $O(\varepsilon^{-3})$ rate, making the comparison with Algo. 2 consistent.
> > >
> > > Our refined theoretical contribution is now stated as follows:
> > >
> > > >  Algo. 2 attains an improved $\mathcal{O}(\varepsilon^{-3})$ rate **under the mean-square smoothness condition (Assump. 3.1)**, matching both the best-known lower bound for unconstrained stochastic optimization [1] and the best-known complexity for nonconvex constrained stochastic optimization [5] **under the same assumption**.
> > >
> > > We will revise the Introduction and the discussion surrounding Assump. 3.1 and Theorem 3.2 to explicitly compare the assumptions across all methods and better contextualize our contributions.
> > >
> > > Thank you again for your time and support!
> > >
> > > ***References***
> > >
> > > [1] Y. Arjevani et al., Lower bounds for nonconvex stochastic optimization, MP, 2023.
> > >
> > > [2] Z. Wang et al., SpiderBoost and momentum: faster variance reduction algorithms, NeurIPS, 2019.
> > >
> > > [3] Q. Tran-Dinh et al., A hybrid stochastic optimization framework for composite nonconvex optimization, MP, 2022.
> > >
> > > [4] A. Cutkosky and F. Orabona, Momentum-based variance reduction in nonconvex SGD, NeurIPS, 2019.
> > >
> > > [5] Q. Shi et al., A momentum-based linearized augmented Lagrangian method for nonconvex constrained stochastic optimization, MOR, 2025.
> > >
> > > [6] D. Boob et al., Stochastic first-order methods for convex and nonconvex functional constrained optimization, MP, 2023.

---

### Decision · Program_Chairs · 2026-04-30

**Decision:**

Accept (regular)

**Comment:**

This paper considers stochastic optimization with dc objectives and non-convex equality constraints. It proposes two algorithms for finding $\epsilon$ stationary solution with provable convergence guarantee.

The reviewers agree the contributions are solid and give high ratings. Hence, I recommend acceptance of this paper.